# Convergence Guarantees for RMSProp and Adam in Generalized-smooth Non-convex Optimization with Affine Noise Variance

**Qi Zhang**                                                                                 *qzhan261@asu.edu*
*School of Electrical, Computer and Energy Engineering*
*Arizona State University*

**Yi Zhou**                                                                                  *yi.zhou@tamu.edu*
*Department of Computer Science and Engineerin*
*Texas A&M University*

**Shaofeng Zou**                                                                             *zou@asu.edu*
*School of Electrical, Computer and Energy Engineering*
*Arizona State University*

**Reviewed on OpenReview:** *https://openreview.net/forum?id=QIzRdjIWnS*

## Abstract

This paper provides the first tight convergence analyses for RMSProp and Adam for non-convex optimization under the most relaxed assumptions of coordinate-wise generalized smoothness and affine noise variance. RMSProp is firstly analyzed, which is a special case of Adam with adaptive learning rates but without first-order momentum. Specifically, to solve the challenges due to the dependence among adaptive update, unbounded gradient estimate and Lipschitz constant, we demonstrate that the first-order term in the descent lemma converges and its denominator is upper bounded by a function of gradient norm. Based on this result, we show that RMSProp with proper hyperparameters converges to an $\epsilon$-stationary point with an iteration complexity of $\mathcal{O}(\epsilon^{-4})$. We then generalize our analysis to Adam, where the additional challenge is due to a mismatch between the gradient and the first-order momentum. We develop a new upper bound on the first-order term in the descent lemma, which is also a function of the gradient norm. We show that Adam with proper hyperparameters converges to an $\epsilon$-stationary point with an iteration complexity of $\mathcal{O}(\epsilon^{-4})$. Our complexity results for both RMSProp and Adam match with the complexity lower bound established in Arjevani et al. (2023).

## 1 Introduction

RMSProp (Hinton et al., 2012) and Adam (Kingma & Ba, 2014) are among the most popular and powerful adaptive optimizers in training state-of-the-art machine learning models (Brock et al., 2018; Brown et al., 2020; Cutkosky & Mehta, 2020; Dosovitskiy et al., 2020). RMSProp and Adam only require first-order gradients with little memory requirement, and thus are efficient to use in practice. RMSProp is based on the idea of adaptive learning rates for each individual parameter, and Adam combines the benefits of RMSprop (Hinton et al., 2012) and AdaGrad (Duchi et al., 2011), which consists of two key components of adaptive learning rates and momentum. Despite their empirical success, theoretical understandings on the convergence and complexity, especially when optimizing non-convex loss functions, e.g., neural networks, still remain underdeveloped until very recently.

Recently, there have been a series of works in examining the convergence and complexity of RMSProp and Adam for non-convex loss functions (see Table 1 for a detailed review). However, these works do

not completely explain the performance of RMSProp and Adam in training neural networks, as they rely on assumptions that may not necessarily hold. For example, Zhang et al. (2019) pointed out that neural networks are not $L$-smooth, and instead satisfy the generalized $(L_0, L_1)$-smoothness, where the gradient Lipschitz constant increases linearly with the gradient norm. Furthermore, many of these works assumed that the stochastic gradient has a bounded norm/variance, which however does not even hold for linear regression (Wang et al., 2023b), and instead a relaxed affine noise variance condition shall be used.

In this paper, we derive the convergence guarantee and iteration complexity for RMSProp and Adam with coordinate-wise generalized $(L_0, L_1)$-smooth loss function and affine noise variance. To the best of our knowledge, this is one of the most relaxed assumption sets in the convergence analyses of RMSProp and Adam that best describe the training of some neural networks. We prove that RMSProp and Adam with proper hyperparameters converge to an $\epsilon$-stationary point with a complexity of $\mathcal{O}(\epsilon^{-4})$, which matches with the lower bound for first-order optimization in Arjevani et al. (2023).

## 1.1 Related work

### 1.1.1 Relaxed Assumptions

**Affine Noise Variance:** In most of the studies on stochastic optimization, access to an unbiased estimate of the gradient with uniformly bounded variance is assumed (Nemirovskij & Yudin, 1983; Ghadimi & Lan, 2013; Bubeck et al., 2015; Foster et al., 2019). Ghadimi & Lan (2013) first showed that for $L$-smooth objectives, the SGD algorithm converges to a first-order $\epsilon$-stationary point with an iteration complexity of $\mathcal{O}(\epsilon^{-4})$ if the stochastic gradient has uniformly bounded variance. Furthermore, Arjevani et al. (2023) proved that for any first-order algorithm with uniformly bounded gradient variance, the iteration complexity of $\mathcal{O}(\epsilon^{-4})$ is optimal. For overparameterized neural networks, Vaswani et al. (2019) considered another gradient noise assumption: the strong growth condition, where the upper bound on the second-order moment of the norm of gradient estimate scales with the gradient square norm. Both the uniformly bounded variance and strong growth condition are special cases of the affine noise variance. It was demonstrated in Bottou et al. (2018) that for non-adaptive algorithms with affine noise variance, the optimal iteration complexity of $\mathcal{O}(\epsilon^{-4})$ can be achieved. The extension of affine noise variance assumption to adaptive algorithms is not straightforward and was studied in Jin et al. (2021); Chen et al. (2023); Wang et al. (2022; 2023b); Shi et al. (2020); Faw et al. (2022; 2023); Hong & Lin (2023). In this paper, we study two adaptive optimizers: RMSProp and Adam with affine noise variance.

**Generalized Smoothness:** In stochastic optimization, the $L$-smooth objectives are widely assumed (Ghadimi & Lan, 2013; Ghadimi et al., 2016). However, it was demonstrated in Zhang et al. (2019) that the $L$-smoothness does not hold for some neural networks and polynomial functions with degree larger than 2. Then, extensive experiments were conducted to verify that these functions satisfy the generalized $(L_0, L_1)$-smoothness condition, where the gradient Lipschitz constant increases linearly with the gradient norm. Compared with $L$-smoothness, $(L_0, L_1)$-smoothness introduces extra second-order error terms, thus making the optimization problem hard to solve. The clipping algorithms for generalized smooth function were studied in Zhang et al. (2019; 2020). However, they require the gradient norm to be bounded. A relaxed assumption on bounded gradient variance was studied in Reisizadeh et al. (2023), where the SPIDER algorithm was applied. Furthermore, Chen et al. (2023) showed that for generalized smooth objectives with affine noise variance, the SPIDER algorithm still finds a stationary point. Under the same assumption, Jin et al. (2021) provided the convergence rate for a normalized momentum algorithm. With extensive experiments, Crawshaw et al. (2022) showed that in the training of Transformer, the $(L_0, L_1)$-smoothness holds coordinate-wisely. This condition is widely used in coordinate-wise type optimizers like generalized SignSGD and Adam. Note that for the original Adam, proving the expectation of gradient norm converges with $(L_0, L_1)$-smoothness remains an unresolved issue. In this paper, we consider functions that are coordinate-wise $(L_0, L_1)$-smooth.

### 1.1.2 Adaptive Optimizers

Adaptive optimizers are widely used in deep learning due to their ability to adapt to changing data and conditions. Adagrad (Duchi et al., 2011) is the first adaptive algorithm, which calculates the accumulated sum of the past gradient norms and uses the reciprocal of its square root to scale the current gradient.

Recently, Wang et al. (2023b); Faw et al. (2023) studied Adagrad under generalized smoothness and affine noise variance conditions. However, the training of the above Adagrad algorithm may stop in advance since the accumulated sum does not shrink, and thus the learning rate can be extremely close to 0. To overcome this problem, RMSProp (Hinton et al., 2012) was proposed, where a momentum method is employed to replace the accumulated sum. Thus, the adaptive learning rate can increase or decrease. There is a rich literature on the convergence analyses of RMSProp (Zaheer et al., 2018; De et al., 2018; Shi et al., 2020). However, all of them focus on $L$-smooth objectives, and only Shi et al. (2020) considered the affine noise variance. RMSProp is a special case of Adam, which only includes the second-order momentum, and is widely studied e.g., Défossez et al. (2020); Zou et al. (2019); Chen et al. (2018); Zhang et al. (2022); Wang et al. (2022); Guo et al. (2021); Hong & Lin (2023); Li et al. (2023); Wang et al. (2023a). There are also two recent works (Hong & Lin, 2023; Wang et al., 2023a) that studied Adam for $L$-smooth objectives with affine noise variance. However, their methods can not be generalized to $(L_0, L_1)$ smooth objectives due to the additional terms invalidating the key inequalities or requirements of bounded Lipchitz constant in their key Lemma. In Li et al. (2023), Adam for $(L_0, L_1)$-smooth objectives with sub-Gaussian norm was studied, where the gradient estimate bias follows a sub-Gaussian distribution. Under this assumption, based on the gradient estimate, the real gradient belongs to a bounded set with high probability, which converts the unbounded Lipschitz constant to a bounded one. However, the bounded Lipschitz constant is quite large, which leads to small step sizes and slow practical convergence. Adam on $(L_0, L_1)$-smoothness with affine noise variance (for the special case of finite sum problems) were in Wang et al. (2022). However, they only showed that Adam converges to the neighborhood of a stationary point with a constant learning rate. More details can be found in Table 1.

| Method | Smoothness[1] | Algorithm | Convergence[2] | Assumption[3] | Batch size | Complexity |
|---|---|---|---|---|---|---|
| De et al. (2018) | (LS) | RMSProp | ✓ | (BN)[4] | $\mathcal{O}(1)$ | $\mathcal{O}(\epsilon^{-4})$ |
| Zaheer et al. (2018) | (LS) | RMSProp | ✓ | (BN) | $\mathcal{O}(\epsilon^{-2})$ | $\mathcal{O}(\epsilon^{-4})$ |
| Shi et al. (2020) | (LS) | RMSProp | ✓ | (FSAN) | - | - |
| Défossez et al. (2020) | (LS) | Adam | ✓ | (BN) | $\mathcal{O}(1)$ | $\mathcal{O}(\epsilon^{-4})$ |
| Zou et al. (2019) | (LS) | Adam | ✓ | (BSM) | $\mathcal{O}(1)$ | $\mathcal{O}(\epsilon^{-4})$ |
| Chen et al. (2018) | (LS) | Adam | ✗ | (BN) | - | - |
| Zhang et al. (2022) | (LS) | Adam | ✗ | (FSAN) | - | - |
| Wang et al. (2022) | (FSGS) | Adam | ✗ | (FSAN) | - | - |
| Guo et al. (2021) | (LS) | Adam | ✓ | (AN)[5] | $\mathcal{O}(1)$ | $\mathcal{O}(\epsilon^{-4})$ |
| Hong & Lin (2023) | (LS) | Adam | ✓ | (CAN) | $\mathcal{O}(1)$ | $\tilde{\mathcal{O}}(\epsilon^{-4})$ |
| Wang et al. (2023a) | (LS) | Adam | ✓ | (CAN) | $\mathcal{O}(1)$ | $\mathcal{O}(\epsilon^{-4})$ |
| Li et al. (2023) | (GS) | Adam | ✓ | (SGN) | $\mathcal{O}(1)$ | $\mathcal{O}(\epsilon^{-4})$[6] |
| Wang et al. (2024) | (GS) | Scalar Adam | ✓ | (AN) | $\mathcal{O}(1)$ | $\mathcal{O}(\epsilon^{-4})$ |
| Our method | (CWGS) | Adam | ✓ | (CAN) | $\mathcal{O}(1)$ | $\mathcal{O}(\epsilon^{-4})$ |

Table 1: Comparison for existing RMSProp and Adam analyses. For $\nabla f(\boldsymbol{x})$ with its estimate $\boldsymbol{g}$, the bounded norm assumption is $\|\boldsymbol{g}\| \leq G$ (almost surely), where $G$ is some positive constant. The bounded second-order moment assumption is that $\mathbb{E}[\|\boldsymbol{g}\|^2] \leq G^2$. The bounded sub-Gaussian norm assumption is that $\|\boldsymbol{g} - \nabla f(\boldsymbol{x})\|$ follows a sub-Gaussion distribution, which is weaker than the bounded norm assumption but stronger than the bounded variance assumption. The batch size refers to the number of samples necessary to compute the gradient estimate $\boldsymbol{g}$ and complexity denotes the total computational effort required to achieve an $\epsilon$-stationary point. Explanation on the upper footmarks: 1 : (LS) indicates the standard $L$-smoothness, (GS) denotes the generalized $(L_0, L_1)$-smoothness, (FSGS) denotes the finite sum $(L_0, L_1)$-smoothness and (CWGS) indicates the coordinate-wise $(L_0, L_1)$-smoothness. 2 : ✗ indicates the algorithm only converges to the neighborhood of a stationary point, whose radius can not be made small. 3 : (BN) indicates Bounded Norm, (FSAN) indicates Finite Sum Affine Noise, (BSM) indicates Bounded Second-order Moment, (AN) indicates Affine Noise, (CGN) indicates Sub-Gaussian Norm. 4 : De et al. (2018) also requires the signs of the gradients to be the same across batches. 5 : Guo et al. (2021) also requires the upper bound on the gradient norm. 6 : A variance-reduced method is also investigated in Li et al. (2023), and the complexity is $\mathcal{O}(\epsilon^{-3})$.

When preparing this work, we have observed a concurrent work by Wang et al. (2024), which studies a scalar—or "norm"—version of Adam. In this paper, we study the per-coordinate version of Adam with the practical and challenging coordinate-wise $(L_0, L_1)$-smooth objectives (see Algorithm 1 for more details).

## 2 Preliminaries

Let $f : \mathbb{R}^d \to \mathbb{R}$ be a differentiable non-convex loss function. For a positive integer $d$, let $[d]$ denote the set $\{1, 2, ..., d\}$. Let $\boldsymbol{x} \in \mathbb{R}^d$ be an optimization variable. Our goal is to minimize the objective function $f(x)$:

$$\min_{\boldsymbol{x}} f(\boldsymbol{x}).$$

For a differentiable function $f$, a point $\boldsymbol{x} \in \mathbb{R}^d$ is called a first-order $\epsilon$-stationary point if $\|\nabla f(\boldsymbol{x})\| \leq \epsilon$. Denote by $\boldsymbol{x}_t \in \mathbb{R}^d$ the optimization variable at the $t$-th iteration and we have access to an estimate $\boldsymbol{g}_t$ of $\nabla f(\boldsymbol{x}_t)$. Define $\mathcal{F}_t := \sigma(\boldsymbol{g}_1, ..., \boldsymbol{g}_{t-1})$ is the sigma field of the stochastic gradients up to $t-1$. We focus on the Adam algorithm shown in Algorithm 1, where $\odot$ denotes the Hadamard product. For any $i \in [d]$, $\partial_i f(\boldsymbol{x}_t), \boldsymbol{g}_{t,i}, \boldsymbol{m}_{t,i}$ and $\boldsymbol{v}_{t,i}$ are the $i$-th element of $\nabla f(\boldsymbol{x}_t), \boldsymbol{g}_t, \boldsymbol{m}_t$ and $\boldsymbol{v}_t$, respectively.

The Adam algorithm is provided in Algorithm 1. Compared with the original Adam, we make a minor change in the adaptive stepsize from $\boldsymbol{\eta}_t = \frac{\eta}{\sqrt{\boldsymbol{v}_t} + \zeta}$ to $\frac{\eta}{\sqrt{\boldsymbol{v}_t + \zeta}}$. This minor change does not influence the adaptivity of the algorithm but makes the analysis much easier.

### 2.1 Technical Assumptions

Throughout this paper, we make the following assumptions.

**Assumption 1.** $f(\boldsymbol{x})$ is bounded from below such that $\inf_{\boldsymbol{x}} f(\boldsymbol{x}) > -\infty$.

**Assumption 2** (Coordinate-wise affine noise variance (Wang et al., 2023a; Hong & Lin, 2023)). *We have access to an unbiased gradient estimate $\boldsymbol{g}_t$ such that $\mathbb{E}[\boldsymbol{g}_t | \mathcal{F}_t] = \nabla f(\boldsymbol{x}_t)$ and for any $i \in [d]$, $\mathbb{E}[\boldsymbol{g}_{t,i}^2 | \mathcal{F}_t] \leq D_0 + D_1(\partial_i f(\boldsymbol{x}_t))^2$, where $D_0, D_1 \geq 0$ are some constants.*

As discussed in Hong & Lin (2023), this assumption allows the magnitude of noise to scale with the corresponding gradient coordinate. Many widely used noise assumptions are special cases of this affine noise variance assumption. For example, when $D_1 = 0$, it is the bounded second-order moment assumption in Zou et al. (2019) and when $D_1 = 1$, it is equivalent to coordinate-wise bounded gradient variance. However, as pointed out in Wang et al. (2023b), these two assumptions of bounded second-order moment and bounded gradient variance do not even hold for linear regression problems. For example, let $f(\omega) = \mathbb{E}_{z \sim \mathcal{D}}(\langle z, \omega \rangle)^2 = \omega^2$, where $z$ is a sample and $\mathcal{D}$ is a standard Gaussian distribution over $\mathbb{R}$. It can be shown that $g = 2z^2\omega$ is an unbiased estimate of $\nabla f(\omega)$. However, both the variance and second-order moment of $g$ is in the order of $\mathcal{O}(\omega^2)$ which are unbounded when $\omega \to \infty$. When $D_0 = 0$, it is called the "strong growth condition" (Vaswani et al., 2019), which is shown to be reasonable for overparameterized neural networks that can interpolate all data points (Vaswani et al., 2019). Under Assumption 2, the norm of the gradient increases with the norm of the true gradient. This is important for model parameters that are multiplicatively perturbed by noise, e.g., multilayer network (Faw et al., 2022). In this paper, we study the coordinate-wise affine noise variance assumption, which was also used in Hong & Lin (2023); Wang et al. (2023a).

Though the $L$-smoothness assumption is widely used in optimization studies, recently it has been observed that in the training of neural networks, such as LSTMs (Zhang et al., 2019), ResNets(Zhang et al., 2019) and Transformers (Crawshaw et al., 2022), this assumption does not hold. Instead, it is numerically verified that the following generalized smoothness assumption better models the training of neural networks (Zhang et al., 2019): $\|\nabla f(\boldsymbol{x}) - \nabla f(\boldsymbol{y})\| \leq (L_0 + L_1\|\nabla f(\boldsymbol{x})\|)\|\boldsymbol{x} - \boldsymbol{y}\|$ for some positive $L_0$ and $L_1$. This assumption is widely studied in the literature, e.g., Jin et al. (2021); Chen et al. (2023); Li et al. (2023); Wang et al. (2022; 2023b); Faw et al. (2023). Compared with $L$-smooth functions, for the generalized smooth functions, the Lipschitz constant scales with the true gradient norm thus may not be bounded. For the training of Transformer models, Crawshaw et al. (2022) finds the following coordinate-wise $(L_0, L_1)$-smoothness, which

provides a more accurate characterization of the objective. It is generalized from the coordinate-wise $L$-smoothness (Richtárik & Takáč, 2014; Khaled & Richtárik, 2020; Bernstein et al., 2018). In this paper, we focus on coordinate-wise $(L_0, L_1)$-smooth functions as defined below:

**Assumption 3** (Coordinate-wise $(L_0, L_1)$-smoothness). *A function $f$ is coordinate-wise $(L_0, L_1)$-smooth if for any $\boldsymbol{x}, \boldsymbol{y} \in \mathbb{R}^d$ and $i \in [d]$,*

$$|\partial_i f(\boldsymbol{x}) - \partial_i f(\boldsymbol{y})| \leq \left( \frac{L_0}{\sqrt{d}} + L_1 |\partial_i f(\boldsymbol{x})| \right) \|\boldsymbol{x} - \boldsymbol{y}\|. \tag{1}$$

The training of Adam enjoys adaptive learning rates for each parameter individually due to the coordinate-wise update process. Moreover, extensive experiments in Crawshaw et al. (2022) show that the $L_0$ and $L_1$ for each coordinate vary a lot. Therefore, it is more accurate to leverage the coordinate-wise $(L_0, L_1)$-smoothness. In this paper, for the sake of simplicity and coherence with the coordinate-wise affine noise variance assumption, we assume the $L_0$ and $L_1$ to be identical for each coordinate. Our results can be easily adapted to the case with distinct $L_0$ and $L_1$ for different coordinates.

## 2.2 Challenges and Insights

Our theoretical analyses address several major challenges: (i) dependence between stepsize and gradient, (ii) potential unbounded gradients, (iii) mismatch between gradient and first-order momentum, and (iv) additional bias terms due to affine variance and coordinate-wise $(L_0, L_1)$-smoothness. Prior research circumvented most of these challenges by introducing extra assumptions, whereas we provide several new insights and show that these assumptions may not be needed.

---

**Algorithm 1** Adam

---

    Initialize parameters: $\boldsymbol{x}_1$, learning rates $\eta$, $\beta_1, \beta_2$, $\zeta$, Iteration $T$
    Initialize first and second moment estimates: $\boldsymbol{v}_0 \in \mathbb{R}^+, \boldsymbol{m}_0 = 0$
    Initialize time step: $t = 1$
    **while** $t \leq T$ **do**
        Generate stochastic gradient: $\boldsymbol{g}_t$
        Update first-order momentum estimate: $\boldsymbol{m}_t \leftarrow \beta_1 \boldsymbol{m}_{t-1} + (1 - \beta_1)\boldsymbol{g}_t$
        Update second-order momentum estimate: $\boldsymbol{v}_t \leftarrow \beta_2 \boldsymbol{v}_{t-1} + (1 - \beta_2)\boldsymbol{g}_t \odot \boldsymbol{g}_t$
        Update parameters: $\boldsymbol{x}_{t+1} \leftarrow \boldsymbol{x}_t - \eta \frac{1}{\sqrt{\boldsymbol{v}_t + \zeta}} \odot \boldsymbol{m}_t$
        $t \leftarrow t + 1$
    **end while**

---

In this paper, we have an access to an unbiased estimate $\boldsymbol{g}$ of $\nabla f(\boldsymbol{x})$ such that $\mathbb{E}[\boldsymbol{g}|\boldsymbol{x}] = \nabla f(\boldsymbol{x})$. Consider the Adam algorithm in Algorithm 1, which reduces to RMSProp if $\beta_1 = 0$. For coordinate-wise $(L_0, L_1)$-smooth objective functions, we have the following descent inequality (Lemma 1 in Crawshaw et al. (2022)):

$$\underbrace{\mathbb{E}\left[\langle \nabla f(\boldsymbol{x}_t), \boldsymbol{x}_t - \boldsymbol{x}_{t+1} \rangle | \mathcal{F}_t\right]}_{\text{first-order}} \leq f(\boldsymbol{x}_t) - \mathbb{E}[f(\boldsymbol{x}_{t+1})|\mathcal{F}_t]$$

$$+ \underbrace{\sum_{i=1}^d \frac{L_0}{2\sqrt{d}} \mathbb{E}[\|\boldsymbol{x}_{t+1} - \boldsymbol{x}_t\| |\boldsymbol{x}_{t+1,i} - \boldsymbol{x}_{t,i}| |\mathcal{F}_t]}_{\text{second-order}} + \underbrace{\sum_{i=1}^d \frac{L_1 |\partial_i f(\boldsymbol{x}_t)|}{2} \mathbb{E}[\|\boldsymbol{x}_{t+1} - \boldsymbol{x}_t\| |\boldsymbol{x}_{t+1,i} - \boldsymbol{x}_{t,i}| |\mathcal{F}_t]}_{\text{additional term}}, \tag{2}$$

where the last term is the additional term due to the $(L_0, L_1)$-smooth assumption.

**Challenge 1: dependence between stepsize and gradient.** We use the RMSProp optimizer to explain our technical novelty. The challenge is the same for Adam. For RMSProp the optimized parameter $\boldsymbol{x}$ is updated: $\boldsymbol{x}_{t+1} = \boldsymbol{x}_t - \frac{\eta \boldsymbol{g}_t}{\sqrt{\boldsymbol{v}_t} + \zeta}$, where the adaptive stepsize $\boldsymbol{\eta}_t = \frac{\eta}{\sqrt{\boldsymbol{v}_t} + \zeta}$ depends on the current gradient estimate $\boldsymbol{g}_t$, which makes it hard to bound the conditional expectation of the first-order term in equation 2. To address this challenge, studies on Adagrad (Ward et al., 2020; Défossez et al., 2020; Faw et al., 2022)

and studies on RMSProp (Zaheer et al., 2018) propose a surrogate $\tilde{\boldsymbol{v}}_t$ of $\boldsymbol{v}_t$ which is independent of $\boldsymbol{g}_t$, and then the first-order term is divided into two parts: the first-order.a term $\mathbb{E}\left[\left\langle \nabla f(\boldsymbol{x}_t), \frac{-\eta \boldsymbol{g}_t}{\sqrt{\tilde{\boldsymbol{v}}_t}+\zeta}\right\rangle \Big| \mathcal{F}_t\right]$ and first-order.b term $\mathbb{E}\left[\left\langle \nabla f(\boldsymbol{x}_t), \frac{-\eta \boldsymbol{g}_t}{\sqrt{\boldsymbol{v}_t}+\zeta} + \frac{\eta \boldsymbol{g}_t}{\sqrt{\tilde{\boldsymbol{v}}_t}+\zeta}\right\rangle \Big| \mathcal{F}_t\right]$. The main challenge lies in the first-order.b term (surrogate error). In Zaheer et al. (2018), this term is bounded based on the assumption of bounded gradient norm, which does not hold in this paper.

*Insight 1: reduce surrogate error.* We choose the same surrogate $\tilde{\boldsymbol{v}}_t = \beta_2 \boldsymbol{v}_{t-1}$ as the one in Zaheer et al. (2018), which requires that the gradient norm is bounded. To remove this assumption, we change the adaptive stepsize from $\frac{\eta}{\sqrt{\boldsymbol{v}_t}+\zeta}$ to $\frac{\eta}{\sqrt{\boldsymbol{v}_t}+\zeta}$ (details seen in Remark 1). For Adagrag (Wang et al., 2023b), due to the non-increasing stepsize, in the $t$-th iteration, the error term can be written as $\mathbb{E}[\phi(\boldsymbol{v}_{t-1}) - \phi(\boldsymbol{v}_t)|\mathcal{F}_t]$ where $\phi$ is some function. Thus, these terms cancel out with each other by taking the sum from $t = 1$ to $T$. However, this method does not work for Adam due to the employment of second-order momentum. With our modified adaptive stepsize, we can also show the sum of error terms is bounded (details can be found in Lemma 1). We notice that a different surrogate is selected in Wang et al. (2023a), which aims to bound the surrogate error for $L$-smooth objectives while our surrogate is tailored to the specific requirements of our generalized smooth objectives.

**Challenge 2: unbounded gradient.** Previous works, e.g., De et al. (2018); Défossez et al. (2020); Zaheer et al. (2018) assumed that the gradient norm is bounded, based on which, they proved the convergence. However, with affine gradient noise, the gradient may be infinite, and thus those approaches do not apply.

*Insight 2: recursive bounding technique.* For RMSProp, with bounded surrogate error in Lemma 1, we first show that $\mathbb{E}\left[\frac{1}{T}\sum_{t=1}^{T}\frac{\|\nabla f(\boldsymbol{x}_t)\|^2}{\sqrt{\|\tilde{\boldsymbol{v}}_t\|+\zeta}}\right]$ is of the order of $\mathcal{O}(\epsilon^2)$. If the gradient norm is upper bounded, then $\tilde{\boldsymbol{v}}_t$ is bounded, and the convergence result directly follows. However, under affine gradient noise, the gradient norm may not be bounded. For generalized smooth objectives, we develop a novel approach to bound $\mathbb{E}\left[\frac{1}{T}\sum_{t=1}^{T}\sqrt{\|\tilde{\boldsymbol{v}}_t\|+\zeta}\right]$ using $\mathbb{E}\left[\sum_{t=1}^{T}\|\nabla f(\boldsymbol{x}_t)\|\right]$ instead of a constant (see Lemma 3 for details). Applying Hölder's inequality (Hardy et al., 1952) we will obtain the convergence result. The complexity result matches with the lower bound in Arjevani et al. (2023). A similar method is applied in Wang et al. (2023a), which focuses on the $L$-smooth objectives and bound $\mathbb{E}\left[\sum_{t=1}^{T}\|\nabla f(\boldsymbol{x}_t)\|\right]$ by a constant.

**Challenge 3: mismatch between gradient and first-order momentum.** Compared with RMSProp, Adam employs the first-order momentum $\boldsymbol{m}_t$. The momentum $\boldsymbol{m}_t$ is dependent on the surrogate stepsize $\frac{\eta}{\sqrt{\tilde{\boldsymbol{v}}_t}+\zeta}$. Moreover, the momentum $\boldsymbol{m}_t$ is a biased estimate of the current true gradient. Both the above challenges make it hard to theoretically characterize the convergence rate of Adam. These mismatch challenges also accur in the analysis for SGDM (Liu et al., 2020) and Adam (Wang et al., 2023a), where a potential function of $f(\boldsymbol{u}_t)$ with $\boldsymbol{u}_t = \frac{\boldsymbol{x}_t - \boldsymbol{x}_{t-1}\beta_1/\sqrt{\beta_2}}{1-\beta_1/\sqrt{\beta_2}}$ is studied. It can be shown that $\boldsymbol{u}_{t+1} - \boldsymbol{u}_t$ is close to a function of $\frac{\boldsymbol{g}_t}{\sqrt{\tilde{\boldsymbol{v}}_t}+\zeta}$, which is much easier to analyze compared with $\frac{\boldsymbol{m}_t}{\sqrt{\tilde{\boldsymbol{v}}_t}+\zeta}$. However, both of them are limited to $L$-smooth objectives.

*Insight 3: bounding first-order term using $\mathbb{E}\left[\frac{1}{T}\sum_{t=1}^{T}\|\nabla f(\boldsymbol{x}_t)\|\right]$.* In this paper, we choose the same potential function but different surrogate in Wang et al. (2023a). Using the descent lemma of $f(\boldsymbol{u}_t)$, we show that the first-order term is also bounded by a $\epsilon^2$-level constant plus a function of $\mathbb{E}\left[\frac{1}{T}\sum_{t=1}^{T}\|\nabla f(\boldsymbol{x}_t)\|\right]$. Compared to RMSProp, this additional function is introduced due to the bias of $\boldsymbol{m}_t$. Then via Hölder's inequality, we show Adam converges as fast as RMSProp.

**Challenge 4: additional error terms due to affine variance and $(L_0, L_1)$-smoothness.** Compared with $L$-smooth objectives, in the analysis for RMSProp with $(L_0, L_1)$-smooth objectives, there is an additional second-order error term: $\sum_{i=1}^{d}\frac{L_1|\partial_i f(\boldsymbol{x}_t)|}{2}[\|\boldsymbol{x}_{t+1} - \boldsymbol{x}_t\|\|\boldsymbol{x}_{t+1,i} - \boldsymbol{x}_{t,i}\|]$, which is hard to bound since $|\partial_i f(\boldsymbol{x}_t)|$ may be unbounded. Moreover, for RMSProp, since $\mathbb{E}[\|\boldsymbol{x}_{t+1,i} - \boldsymbol{x}_{t,i}\||\mathcal{F}_t] \leq \mathbb{E}\left[\frac{\eta|\boldsymbol{g}_{t,i}|}{\sqrt{\tilde{\boldsymbol{v}}_{t,i}}+\zeta}\Big|\mathcal{F}_t\right]$ and $\boldsymbol{g}_{t,i}$ is independent of $\tilde{\boldsymbol{v}}_{t,i}$ given $\mathcal{F}_t$, the affine noise variance assumption can be leveraged to bound the second-

order error term directly. Nevertheless, for Adam due to the dependence between $\boldsymbol{m}_{t,i}$ and $\tilde{\boldsymbol{v}}_{t,i}$, the above approach cannot be applied directly.

*Insight 4: bounding additional term by function of first-order term.* For RMSProp, we can show that $\|\boldsymbol{x}_{t+1} - \boldsymbol{x}_t\| \leq \frac{\eta\sqrt{d}}{\sqrt{1-\beta_2}}$ and $|\partial_i f(\boldsymbol{x}_t)||\boldsymbol{x}_{t+1,i} - \boldsymbol{x}_{t,i}| \leq \frac{|\partial_i f(\boldsymbol{x}_t)|^2 + \eta^2 \boldsymbol{g}_{t,i}^2}{2\sqrt{\tilde{\boldsymbol{v}}_{t,i} + \zeta}}$. According to the affine noise assumption, we have that $\mathbb{E}[\boldsymbol{g}_{t,i}^2|\mathcal{F}_t]$ is upper bounded by a linear function of $(\partial_i f(\boldsymbol{x}_t))^2$, thus we can bound the additional error term. However, we cannot directly apply this method to Adam since $\mathbb{E}[\boldsymbol{m}_{t,i}^2|\mathcal{F}_t]$ is hard to bound. Instead, we provide a new upper bound on $\sum_{t=1}^{T} \frac{\boldsymbol{m}_{t,i}^2}{\sqrt{\boldsymbol{v}_{t,i}}}$ using the gradient norm (details can be found in Lemma 8).

## 3 Convergence Analysis of RMSProp

To provide a fundamental understanding of Adam, in this section, we focus on RMSProp which consists of the design of adaptive learning rates for each individual parameter in Adam.

For RMSProp, the main challenges come from the dependence between stepsize and gradient, potential unbounded gradients due to the affine noise variance, and additional error terms due to $(L_0, L_1)$-smoothness. The analysis can be extended to general Adam, which requires additional efforts to handle the first-order momentum. Define $c = \sqrt{\zeta} + d\sqrt{D_0 + \|\boldsymbol{v}_0\|}$. We then present our results of RMSProp in the following theorem.

**Theorem 1** (Informal). *Let Assumptions 1, 2 and 3 hold. Let $1 - \beta_2 = \mathcal{O}(\epsilon^2)$, $\eta = \mathcal{O}(\epsilon^2)$, and $T = \mathcal{O}(\epsilon^{-4})$. For $\epsilon$ such that $\epsilon \leq \frac{\sqrt{5dD_0}}{\sqrt{D_1}\sqrt[4]{\zeta}}$, we have that*

$$\frac{1}{T} \sum_{t=1}^{T} \mathbb{E}[\|\nabla f(\boldsymbol{x}_t)\|] \leq \left( \frac{2d\sqrt{35D_0 D_1}}{\sqrt[4]{\zeta}} + \sqrt{c} \right) \epsilon. \tag{3}$$

To the best of our knowledge, this paper provides the first convergence analysis of RMSProp for $(L_0, L_1)$-smooth functions with affine noise variances. Existing studies mostly assume bounded gradient norm or variance (De et al., 2018; Zaheer et al., 2018) or only show the algorithm converges asymptotically (Shi et al., 2020). More importantly, our result matches the lower bound in Arjevani et al. (2023), and thus is optimal.

The formal version of the theorem and the detailed proof can be found in Appendix D.

Below, we provide a proof sketch to highlight our major technical novelties.

*Proof sketch.* Our proof can be divided into three stages: Stage I: develop an upper bound of $\mathbb{E}\left[ \frac{\|\nabla f(\boldsymbol{x}_t)\|^2}{\sqrt{\beta_2 \|\boldsymbol{v}_{t-1}\| + \zeta}} \right]$; Stage II: develop an upper bound on $\mathbb{E}[\sqrt{\beta_2 \|\boldsymbol{v}_{t-1}\| + \zeta}]$; and Stage III: show $\mathbb{E}[\|\nabla f(\boldsymbol{x}_t)\|]$ converges using results from Stages I, II and Hölder's inequality.

**Stage I:** upper bound of $\mathbb{E}\left[ \frac{\|\nabla f(\boldsymbol{x}_t)\|^2}{\sqrt{\beta_2 \|\boldsymbol{v}_{t-1}\| + \zeta}} \right]$. As discussed in Section 1, for coordinate-wise $(L_0, L_1)$-smooth functions, following the descent inequality (Lemma 1 in Crawshaw et al. (2022)) we can get (2). We first obtain $\frac{\|\nabla f(\boldsymbol{x}_t)\|^2}{\sqrt{\beta_2 \|\boldsymbol{v}_{t-1}\| + \zeta}} \leq \sum_{i=1}^{d} \frac{\eta(\partial_i f(\boldsymbol{x}_t))^2}{\sqrt{\beta_2 \boldsymbol{v}_{t-1,i} + \zeta}}$. Therefore, in the following we will bound $\mathbb{E}\left[ \sum_{i=1}^{d} \frac{\eta(\partial_i f(\boldsymbol{x}_t))^2}{\sqrt{\beta_2 \boldsymbol{v}_{t-1,i} + \zeta}} \right]$. Towards this goal, in Step 1.1, we will show the LHS of equation 2 is lower bounded by a function of $\sum_{i=1}^{d} \frac{\eta(\partial_i f(\boldsymbol{x}_t))^2}{\sqrt{\beta_2 \boldsymbol{v}_{t-1,i} + \zeta}}$; and in Step 1.2 we will show the RHS of equation 2 is upper bounded by a function of $\sum_{i=1}^{d} \frac{\eta(\partial_i f(\boldsymbol{x}_t))^2}{\sqrt{\beta_2 \boldsymbol{v}_{t-1,i} + \zeta}}$. Combining the two steps, we will obtain an upper bound on $\sum_{i=1}^{d} \frac{\eta(\partial_i f(\boldsymbol{x}_t))^2}{\sqrt{\beta_2 \boldsymbol{v}_{t-1,i} + \zeta}}$.

*Step 1.1: lower bound on the first-order term in equation 2.* Since the adaptive stepsize and the gradient estimate are dependent, we design a surrogate $\tilde{\boldsymbol{v}}_t = \beta_2 \boldsymbol{v}_{t-1}$ to decompose the first-order term in equation 2

into two parts:

$$\underbrace{\mathbb{E}[\langle \nabla f(\boldsymbol{x}_t), \boldsymbol{x}_t - \boldsymbol{x}_{t+1}\rangle | \mathcal{F}_t]}_{\text{first-order}}$$

$$= \underbrace{\mathbb{E}\left[\left\langle \nabla f(\boldsymbol{x}_t), \frac{\eta \boldsymbol{g}_t}{\sqrt{\tilde{\boldsymbol{v}}_t + \zeta}}\right\rangle \Big| \mathcal{F}_t\right]}_{\text{first-order.a}} + \underbrace{\mathbb{E}\left[\left\langle \nabla f(\boldsymbol{x}_t), \frac{\eta \boldsymbol{g}_t}{\sqrt{\boldsymbol{v}_t + \zeta}} - \frac{\eta \boldsymbol{g}_t}{\sqrt{\tilde{\boldsymbol{v}}_t + \zeta}}\right\rangle \Big| \mathcal{F}_t\right]}_{\text{first-order.b}}. \tag{4}$$

For the first-order.a term in equation 4, we can show that

$$\mathbb{E}\left[\left\langle \nabla f(\boldsymbol{x}_t), \frac{\eta \boldsymbol{g}_t}{\sqrt{\tilde{\boldsymbol{v}}_t + \zeta}}\right\rangle \Big| \mathcal{F}_t\right] = \sum_{i=1}^d \frac{\eta (\partial_i f(\boldsymbol{x}_t))^2}{\sqrt{\beta_2 \boldsymbol{v}_{t-1,i} + \zeta}}.$$

We then bound the first-order.b term in equation 4.

**Remark 1** (Importance of modified adaptive stepsize $\boldsymbol{\eta}$). *For the original RMSProp, Zaheer et al. (2018) chose the same surrogate $\tilde{\boldsymbol{v}}_t = \beta_2 \boldsymbol{v}_{t-1}$ as in this paper. Then the first-order.b term was lower bounded by a function of $\sum_{i=1}^d \mathbb{E}\left[\frac{-\boldsymbol{g}_{t,i}^2}{\sqrt{\beta_2 \boldsymbol{v}_{t-1,i}+\zeta}}\Big|\mathcal{F}_t\right] \times \mathbb{E}\left[\frac{(1-\beta_2)\boldsymbol{g}_{t,i}^2}{(\sqrt{\boldsymbol{v}_{t,i}}+\sqrt{\beta_2 \boldsymbol{v}_{t-1,i}})^2}\Big|\mathcal{F}_t\right]$. Then they developed an upper bound on the second term $\mathbb{E}\left[\frac{(1-\beta_2)\boldsymbol{g}_{t,i}^2}{(\sqrt{\boldsymbol{v}_{t,i}}+\sqrt{\beta_2 \boldsymbol{v}_{t-1,i}})^2}\Big|\mathcal{F}_t\right] \leq 1$ which is quite loose, thus they introduced an additional assumption on the upper bound of $|\boldsymbol{g}_{t,i}|$ (Zaheer et al., 2018).*

*In contrast, using our adaptive stepsize $\eta \frac{\boldsymbol{g}_t}{\sqrt{\boldsymbol{v}_t + \zeta}}$, we can show that the first-order.b term can be lower bounded by a function of $\sum_{i=1}^d \mathbb{E}\left[\frac{-\boldsymbol{g}_{t,i}^2}{\sqrt{\beta_2 \boldsymbol{v}_{t-1,i}+\zeta}}\Big|\mathcal{F}_t\right] \times \mathbb{E}\left[\frac{(1-\beta_2)\boldsymbol{g}_{t,i}^2}{(\sqrt{\boldsymbol{v}_{t,i}+\zeta}+\sqrt{\beta_2 \boldsymbol{v}_{t-1,i}+\zeta})^2}\Big|\mathcal{F}_t\right]$, which can be further bounded by $\sum_{i=1}^d \mathbb{E}[-\boldsymbol{g}_{t,i}^2|\mathcal{F}_t]\mathbb{E}\left[\frac{1}{\sqrt{\beta_2 \boldsymbol{v}_{t-1,i}+\zeta}} - \frac{1}{\sqrt{\boldsymbol{v}_{t,i}+\zeta}}\Big|\mathcal{F}_t\right]$. Applying the affine noise variance assumption in Assumption 2, we obtain a lower bound on the first-order.b term in the following lemma.*

**Lemma 1** (Informal). *Under Assumptions 2 and 3, we have that*

$$\mathbb{E}\left[\left\langle \nabla f(\boldsymbol{x}_t), \frac{\eta \boldsymbol{g}_t}{\sqrt{\boldsymbol{v}_t + \zeta}} - \frac{\eta \boldsymbol{g}_t}{\sqrt{\beta_2 \boldsymbol{v}_{t-1} + \zeta}}\right\rangle \Big| \mathcal{F}_t\right]$$

$$\geq -\sum_{i=1}^d \left(\mathcal{O}\left(\frac{\eta (\partial_i f(\boldsymbol{x}_t))^2}{\sqrt{\beta_2 \boldsymbol{v}_{t-1,i} + \zeta}}\right) + \mathcal{O}\left(\frac{\eta^2}{\sqrt{1-\beta_2}} \frac{(\partial_i f(\boldsymbol{x}_t))^2}{\sqrt{\beta_2 \boldsymbol{v}_{t-1,i} + \zeta}}\right)\right.$$

$$+ \mathcal{O}\left(\frac{\eta}{\sqrt{\beta_2 \boldsymbol{v}_{t-1,i} + \zeta}} - \mathbb{E}\left[\frac{\eta}{\sqrt{\boldsymbol{v}_{t,i} + \zeta}}\Big|\mathcal{F}_t\right]\right) + \mathcal{O}\left(\frac{\eta (\partial_i f(\boldsymbol{x}_{t-1}))^2}{\sqrt{\beta_2 \boldsymbol{v}_{t-1,i} + \zeta}} - \mathbb{E}\left[\frac{\eta (\partial_i f(\boldsymbol{x}_t))^2}{\sqrt{\boldsymbol{v}_{t,i} + \zeta}}\Big|\mathcal{F}_t\right]\right)\right)$$

$$- \text{small error.} \tag{5}$$

The formal version and detailed proof of Lemma 1 can be found in the Appendix A, which is based on our modified update process on $\boldsymbol{v}_t$, Hölder's inequality and Assumption 3.

In the RHS of equation 5, consider the term $\mathbb{E}\left[\frac{(\partial_i f(\boldsymbol{x}_{t-1}))^2}{\sqrt{\beta_2 \boldsymbol{v}_{t-1,i}+\zeta}} - \frac{(\partial_i f(\boldsymbol{x}_t))^2}{\sqrt{\boldsymbol{v}_{t,i}+\zeta}}\Big|\mathcal{F}_t\right]$. Taking sum from $t = 1$ to $T$, the terms $\frac{(\partial_i f(\boldsymbol{x}_{t-1}))^2}{\sqrt{\beta_2 \boldsymbol{v}_{t-1,i}+\zeta}}$ and $\frac{(\partial_i f(\boldsymbol{x}_{t-1}))^2}{\sqrt{\boldsymbol{v}_{t-1,i}+\zeta}}$ shall be close to each other as $\beta_2 \to 1$. Similarly, $\frac{\eta}{\sqrt{\beta_2 \boldsymbol{v}_{t-1,i}+\zeta}} - \frac{\eta}{\sqrt{\boldsymbol{v}_{t-1,i}+\zeta}}$ can also be bounded.

*Step 1.2: upper bound on second-order and additional terms in equation 2.* We first focus on the second-order term. Based on the update process of $\boldsymbol{v}_t$ and Assumption 2, we get that

$$\sum_{i=1}^d \frac{L_0}{2\sqrt{d}}\mathbb{E}[\|\boldsymbol{x}_{t+1} - \boldsymbol{x}_t\|\|\boldsymbol{x}_{t+1,i} - \boldsymbol{x}_{t,i}\||\mathcal{F}_t] \leq \sum_{i=1}^d \frac{L_0 \eta^2}{2\sqrt{\zeta}} \frac{D_0 + D_1(\partial_i f(\boldsymbol{x}_t))^2}{\sqrt{\beta_2 \boldsymbol{v}_{t-1,i} + \zeta}}. \tag{6}$$

We then focus on the additional term in equation 2 and we provide a new upper bound using $\sum_{i=1}^{d} \frac{(\partial_i f(\boldsymbol{x}_t))^2}{\sqrt{\beta_2 \boldsymbol{v}_{t-1,i}+\zeta}}$ with some small errors: for any $\alpha_2 > 0$, we have that

$$
\begin{aligned}
&\frac{L_1 |\partial_i f(\boldsymbol{x}_t)|}{2} \mathbb{E}[\|\boldsymbol{x}_{t+1} - \boldsymbol{x}_t\| \|\boldsymbol{x}_{t+1,i} - \boldsymbol{x}_{t,i}\| | \mathcal{F}_t] \\
&\leq \frac{\eta^2 \sqrt{d}}{2\sqrt{1-\beta_2}} \left( L_1 \sqrt{D_1} + \frac{1}{\alpha_2 \sqrt{1-\beta_2}} \right) \frac{(\partial_i f(\boldsymbol{x}_t))^2}{\sqrt{\beta_2 \boldsymbol{v}_{t-1,i} + \zeta}} + \frac{\eta^2 \sqrt{d} \alpha_2 L_1^2 D_0}{2\sqrt{\beta_2 \boldsymbol{v}_{t-1,i} + \zeta}}.
\end{aligned}
\tag{7}
$$

Plugging Lemma 1, (6) and (7) in (2) we get the following lemma.

**Lemma 2.** *Let Assumptions 1, 2 and 3 hold. With the same parameters in Theorem 1, we have that*

$$
\frac{1}{T} \sum_{t=1}^{T} \mathbb{E} \left[ \frac{\|\nabla f(\boldsymbol{x}_t)\|^2}{\sqrt{\beta_2 \|\boldsymbol{v}_{t-1}\| + \zeta}} \right] \leq \epsilon^2.
\tag{8}
$$

The details of the proof can be found in the Appendix B. The major idea in the proof of Lemma 2 is to bound the first-order.b, the second-order term and the additional term using $\sum_{i=1}^{d} \mathbb{E} \left[ \frac{(\partial_i f(\boldsymbol{x}_t))^2}{\sqrt{\beta_2 \boldsymbol{v}_{t-1,i} + \zeta}} \right]$.

**Stage II:** upper bound of $\mathbb{E}[\sqrt{\beta_2 \|\boldsymbol{v}_{t-1}\| + \zeta}]$. With the bounded gradient norm assumption, $\frac{1}{T} \sum_{t=1}^{T} \mathbb{E}[\|\nabla f(\boldsymbol{x}_t)\|^2] = \mathcal{O}(\epsilon^2)$ follows directly from Lemma 2. However, under the affine gradient noise assumption in this paper, the gradient norm may not be bounded. Here, we establish a key observation that $\frac{1}{T} \sum_{t=1}^{T} \mathbb{E}[\sqrt{\beta_2 \|\boldsymbol{v}_{t-1}\| + \zeta}]$ can be bounded using $\frac{1}{T} \sum_{t=1}^{T} \mathbb{E}[\|\nabla f(\boldsymbol{x}_t)\|]$. By Hölder's inequality, we have

$$
\left( \frac{1}{T} \sum_{t=1}^{T} \mathbb{E}[\|\nabla f(\boldsymbol{x}_t)\|] \right)^2 \leq \left( \frac{1}{T} \sum_{t=1}^{T} \mathbb{E}[\sqrt{\beta_2 \|\boldsymbol{v}_{t-1}\| + \zeta}] \right) \times \left( \frac{1}{T} \sum_{t=1}^{T} \mathbb{E} \left[ \frac{\|\nabla f(\boldsymbol{x}_t)\|^2}{\sqrt{\beta_2 \|\boldsymbol{v}_{t-1}\| + \zeta}} \right] \right).
\tag{9}
$$

It is worth noting that in the RHS of (9), the second term is bound in Lemma 2. If the first term is upper bounded using $\frac{1}{T} \sum_{t=1}^{T} \mathbb{E}[\|\nabla f(\boldsymbol{x}_t)\|]$, we then can prove an upper bound on $\frac{1}{T} \sum_{t=1}^{T} \mathbb{E}[\|\nabla f(\boldsymbol{x}_t)\|]$, and show the algorithm converges to a stationary point. In the following lemma, we show a novel bound on $\frac{1}{T} \sum_{t=1}^{T} \mathbb{E} \left[ \sqrt{\beta_2 \|\boldsymbol{v}_{t-1}\| + \zeta} \right]$.

**Lemma 3.** *Let Assumption 2 hold. Then, we have*

$$
\frac{1}{T} \sum_{t=1}^{T} \mathbb{E} \left[ \sqrt{\beta_2 \|\boldsymbol{v}_{t-1}\| + \zeta} \right] \leq c + \frac{2\sqrt{dD_1}}{\sqrt{(1-\beta_2)}} \frac{\sum_{t=1}^{T} \mathbb{E}[\|\nabla f(\boldsymbol{x}_t)\|]}{T}.
\tag{10}
$$

The detailed proof can be found in the Appendix C, where we recursively apply Jensen's inequality (Jensen, 1906). The proof only depends on the affine noise variance and the update process on $v_t$. Thus, it works for both RMSProp and Adam with $(L_0, L_1)$-smooth objectives.

**Stage III:** upper bound of $\mathbb{E}[\|\nabla f(\boldsymbol{x}_t)\|]$. Now we show that the algorithm converges to a stationary point. Define $e = \frac{1}{T} \sum_{t=1}^{T} \mathbb{E}[\|\nabla f(\boldsymbol{x}_t)\|]$. By (9), Lemma 2 and Lemma 3 we have that

$$
e^2 \leq \epsilon^2 \left( c + \frac{2\sqrt{dD_1}}{\sqrt{1-\beta_2}} e \right).
\tag{11}
$$

Thus if $\epsilon \leq \frac{\sqrt{5dD_0}}{\sqrt{D_1} \sqrt[4]{\zeta}}$, we have

$$
e \leq \left( \frac{2d\sqrt{35D_0 D_1}}{\sqrt[4]{\zeta}} + \sqrt{c} \right) \epsilon,
$$

which indicates the algorithm converges to a stationary point. $\qquad\square$

## 4 Convergence Analysis of Adam

In this section, we extend our convergence analysis of RMSProp to Adam. Such a result is attractive since empirical results on complicated tasks show Adam may perform better, e.g., the mean reward is improved by 88% via RMSProp and 110% via Adam for Atari games (Agarwal et al., 2020).

We present the convergence result for Adam as follows.

**Theorem 2** (Informal). *Let Assumptions 1, 2 and 3 hold. Let* $1-\beta_2 = \mathcal{O}(\epsilon^2)$, $0 < \beta_1 \leq \sqrt{\beta_2} < 1$, $\eta = \mathcal{O}(\epsilon^2)$, *and* $T = \mathcal{O}(\epsilon^{-4})$. *For small* $\epsilon$ *such that* $\epsilon \leq \frac{\sqrt{2C_2}}{\sqrt{7\alpha_0 C_6 D_1}}$, *we have that*

$$\frac{1}{T} \sum_{t=1}^{T} \mathbb{E}[\|\nabla f(\boldsymbol{x}_t)\|] \leq \left( 2c + \sqrt{2c} + \frac{4\sqrt{dD_1}}{\sqrt{C_6}} \right) \epsilon, \tag{12}$$

*where* $C_6$ *is a positive constant defined in Appendix H.*

To the best of our knowledge, Theorem 2 is the first convergence result of Adam to a stationary point under some of the most relaxed assumptions of $(L_0, L_1)$-smoothness and affine gradient noise. In Li et al. (2023); Wang et al. (2023a), the authors show that the Adam converges to a stationary point with affine noise variance. However, their methods only work for $L$-smooth objectives, while in this paper we focus on the $(L_0, L_1)$-smooth functions. Moreover, we show that for Adam, the overall computational complexity matches with the lower bound in Arjevani et al. (2023), while there is an additional logarithmic term in Li et al. (2023). The normalized momentum algorithm (Jin et al., 2021) can also be viewed as a special case of the Adam family, which applies the momentum on the first-order gradient. However, in their algorithm, a mini-batch of samples is required in each training iteration, while we do not require such a mini-batch. Thus, in the distributed setting with heterogeneous data, where the data distributions under each computational node are different, Algorithm 1 can be used directly. However, the normalized momentum in Jin et al. (2021) may require gradient information from many computational nodes, making the problem degrade to the centralized setting.

The formal version of the theorem and detailed proof can be found in Appendix H. Below, we provide a proof sketch to underscore our key technical novelties.

*Proof Sketch.* Similar to the proof of RMSProp, we divided our proof into three stages. The key difference lies in Stage I, which is because of the dependence between $\boldsymbol{m}_t$ and $\tilde{\boldsymbol{v}}_t$.

**Stage I:** upper bound of $\mathbb{E}\left[ \frac{\|\nabla f(\boldsymbol{x}_t)\|^2}{\sqrt{\beta_2 \|\boldsymbol{v}_{t-1}\| + \zeta}} \right]$. For the Adam optimizer, the first-order.a term $\mathbb{E}\left[ \left\langle \nabla f(\boldsymbol{x}_t), \frac{-\eta \boldsymbol{m}_t}{\sqrt{\tilde{\boldsymbol{v}}_t} + \zeta} \right\rangle \Big| \mathcal{F}_t \right]$ is challenging to bound due to the dependence between $\boldsymbol{m}_t$ and $\tilde{\boldsymbol{v}}_t$. Following the recent analyses of SGDM (Liu et al., 2020) and Adam (Wang et al., 2023a), we study a potential function $f(\boldsymbol{u}_t)$ with $\boldsymbol{u}_t = \frac{\boldsymbol{x}_t - \frac{\beta_1}{\sqrt{\beta_2}} \boldsymbol{x}_{t-1}}{1 - \frac{\beta_1}{\sqrt{\beta_2}}}$. The benefit of analyzing $\boldsymbol{u}_t$ is that we have $\mathbb{E}\left[ \langle \nabla f(\boldsymbol{x}_t), \boldsymbol{u}_t - \boldsymbol{u}_{t+1} \rangle | \mathcal{F}_t \right] \approx \frac{(1-\beta_1)}{1 - \frac{\beta_1}{\sqrt{\beta_2}}} \mathbb{E}\left[ \left\langle \nabla f(\boldsymbol{x}_t), \frac{\eta \boldsymbol{g}_t}{\sqrt{\tilde{\boldsymbol{v}}_t} + \zeta} \right\rangle \Big| \mathcal{F}_t \right]$, where the numerator and denominator in the last term are independent.

In Step 1.1, we will show that the LHS of the descent lemma for $f(\boldsymbol{u}_t)$ (shown in equation 79) is lower bounded by a function of $\sum_{i=1}^{d} \frac{\eta(\partial_i f(\boldsymbol{x}_t))^2}{\sqrt{\beta_2 \boldsymbol{v}_{t-1,i}} + \zeta}$; and in Step 1.2 we will show the RHS of the descent lemma of $f(\boldsymbol{u}_t)$ is upper bounded by a function of $\sum_{i=1}^{d} \frac{\eta(\partial_i f(\boldsymbol{x}_t))^2}{\sqrt{\beta_2 \boldsymbol{v}_{t-1,i}} + \zeta}$. Combining the two steps, we will obtain an upper bound on $\sum_{i=1}^{d} \frac{\eta(\partial_i f(\boldsymbol{x}_t))^2}{\sqrt{\beta_2 \boldsymbol{v}_{t-1,i}} + \zeta}$.

*Step 1.1: lower bound of the first-order term* $\mathbb{E}[\langle \nabla f(\boldsymbol{u}_t), \boldsymbol{u}_t - \boldsymbol{u}_{t+1} \rangle | \mathcal{F}_t]$. We first divide the first-order term into two parts:

$$\mathbb{E}[\langle \nabla f(\boldsymbol{u}_t), \boldsymbol{u}_t - \boldsymbol{u}_{t+1} \rangle | \mathcal{F}_t] = \mathbb{E}[\langle \nabla f(\boldsymbol{x}_t), \boldsymbol{u}_t - \boldsymbol{u}_{t+1} \rangle | \mathcal{F}_t] + \mathbb{E}\left[ \langle \nabla f(\boldsymbol{u}_t) - \nabla f(x_t), \boldsymbol{u}_t - \boldsymbol{u}_{t+1} \rangle | \mathcal{F}_t \right].$$

The first part mimics the first-order term in the proof of RMSProp and the second one is due to a mismatch between $\boldsymbol{u}_t$ and $\boldsymbol{x}_t$. By bounding these two parts separately, we have a lemma similar to Lemma 1 but with two additional terms: $\mathbb{E}\left[\frac{\boldsymbol{m}_{t,i}^2}{\boldsymbol{v}_{t,i}+\zeta}\Big|\mathcal{F}_t\right]$ and $\mathbb{E}\left[\frac{\boldsymbol{m}_{t,i}^2}{\sqrt{\boldsymbol{v}_{t,i}}+\zeta}\Big|\mathcal{F}_t\right]$ due to the employment of the first-order momentum (see details in Lemma 9). For the additional terms, Wang et al. (2023a) showed that $\sum_{t=1}^T \frac{\boldsymbol{m}_{t,i}^2}{\boldsymbol{v}_{t,i}}$ can be bounded by a function of $\boldsymbol{v}_{T,i}$ (shown in Lemma 7). It is worth noting that due to the $(L_0, L_1)$-smoothness our objective is harder to bound and only Lemma 7 is not enough. Here, we bound the $\sum_{t=1}^T \frac{\boldsymbol{m}_{t,i}^2}{\sqrt{\boldsymbol{v}_{t,i}}}$ term by a function of $\frac{\sqrt{\boldsymbol{v}_{T,i}}-\sqrt{\boldsymbol{v}_{0,i}}}{1-\beta_2} + \sum_{t=1}^T \sqrt{\boldsymbol{v}_{t-1,i}}$ (the details can be found in Lemma 8).

*Step 1.2: upper bound on the second-order and additional terms.* Based on the update process of $\boldsymbol{u}_t$ and $\boldsymbol{x}_t$, we bound the second-order and additional terms similarly to those in equation 6 and equation 7, but with $\boldsymbol{g}_t$ replaced by $\boldsymbol{m}_t$ (see details in equation 80 and equation 81). Unlike in the proof of RMSProp, where $\mathbb{E}\left[\frac{\boldsymbol{g}_{t,i}^2}{\sqrt{\beta_2\boldsymbol{v}_{t,i}}+\zeta}\Big|\mathcal{F}_t\right]$ can be bounded by $\frac{D_0}{\sqrt{\zeta}} + \frac{D_1(\partial_i f(\boldsymbol{x}_t))^2}{\sqrt{\boldsymbol{v}_{t,i}}+\zeta}$, Assumption 2 does not hold for $\boldsymbol{m}_t$ and this is the reason why two terms $\mathbb{E}\left[\frac{\boldsymbol{m}_{t,i}^2}{\boldsymbol{v}_{t,i}+\zeta}\Big|\mathcal{F}_t\right]$ and $\mathbb{E}\left[\frac{\boldsymbol{m}_{t,i}^2}{\sqrt{\boldsymbol{v}_{t,i}}+\zeta}\Big|\mathcal{F}_t\right]$ are kept in the upper bound of the second-order and additional terms. Then based on the descent lemma of $f(\boldsymbol{u}_t)$, we can show $\mathbb{E}\left[\sum_{i=1}^d \frac{\eta(\partial_i f(\boldsymbol{x}_t))^2}{\sqrt{\beta_2\boldsymbol{v}_{t-1,i}}+\zeta}\right]$ can be upper bounded by a function of $\sum_{t=1}^T \frac{\boldsymbol{m}_{t,i}^2}{\boldsymbol{v}_{t,i}}$ and $\sum_{t=1}^T \frac{\boldsymbol{m}_{t,i}^2}{\sqrt{\boldsymbol{v}_{t,i}}}$, which further can be bounded by a function of $\mathbb{E}[\nabla f(\boldsymbol{x}_t)]$. We then get the following lemma

**Lemma 4.** *Let Assumptions 1, 2 and 3 hold. With the same parameters as in Theorem 2, we have that*

$$\frac{1}{T}\sum_{t=1}^T \mathbb{E}[\frac{\|\nabla f(\boldsymbol{x}_t)\|^2}{\sqrt{\beta_2\|\boldsymbol{v}_{t-1}\|}+\zeta}] \leq \epsilon^2 + \frac{\epsilon}{T}\sum_{t=1}^T \mathbb{E}[\|\nabla f(\boldsymbol{x}_t)\|].$$

The details of the proof can be found in the Appendix G.

**Stage II** is the same as the proof of RMSProp and thus is omitted here.

**Stage III:** upper bound of $\mathbb{E}[\|\nabla f(\boldsymbol{x}_t)\|]$. As we mentioned before, Lemma 3 and equation 9 hold for Adam. It is worth noting that in the RHS of (9), the first term is bounded in Lemma 4, which is more complicated than Lemma 2 and has an additional term. Let $e = \frac{1}{T}\sum_{t=1}^T \mathbb{E}[\|\nabla f(\boldsymbol{x}_t)\|]$. By (9), Lemma 4 and Lemma 3 we have that

$$e^2 \leq c\epsilon^2 + \frac{2\sqrt{dD_1}}{\sqrt{C_6}\epsilon}e\epsilon^2 + ce\epsilon + \frac{e^2}{2}. \tag{13}$$

Thus, $e \sim \mathcal{O}(\epsilon)$, which shows the algorithm converges to a stationary point. □

# 5 Comparison with Existing Works

We observe that there are two recent works (Li et al., 2023; Wang et al., 2024) on Adam with $(L_0, L_1)$-smooth objectives. In this section, we provide detailed comparisons with them.

Li et al. (2023) studies the original Adam where the adaptive learning rate is $\frac{\eta}{\sqrt{\boldsymbol{v}_t}+\lambda}$ and in this paper we study the modified one where the adaptive learning rate is $\frac{\eta}{\sqrt{\boldsymbol{v}_t+\zeta}}$. Fig. 1 and Fig. 2 demonstrates that this modification has little influence to the model performance. Moreover, though computational complexities of the method in Li et al. (2023) and our method are dependent on $\zeta$ (e.g. $\mathcal{O}(\zeta^{-2})$ for our paper and $\mathcal{O}(\zeta^{-4})$ for Li et al. (2023)), in practice, the selection of $\zeta$ makes minor differences. The analysis of Li et al. (2023) is fundamentally different from this paper, which relies on a stopping time. Thus the authors in Li et al. (2023) only show Adam converges with high probability. Their proof relies on the fact that there exists a large constant $G$ such that $\|\nabla f(\boldsymbol{x}_t)\| \leq G$ for any $t$ before their stopping time, which requires a stronger assumption on gradient noise. Thus the generalized smooth problem is converted to a standard $L$-smooth

problem with a large smoothness constant, leading to very small step sizes which make the convergence slow in practice. In this paper, we do not need this stopping time and we show that the expectation of gradient norm converges, which is stronger than the convergence with high probability.

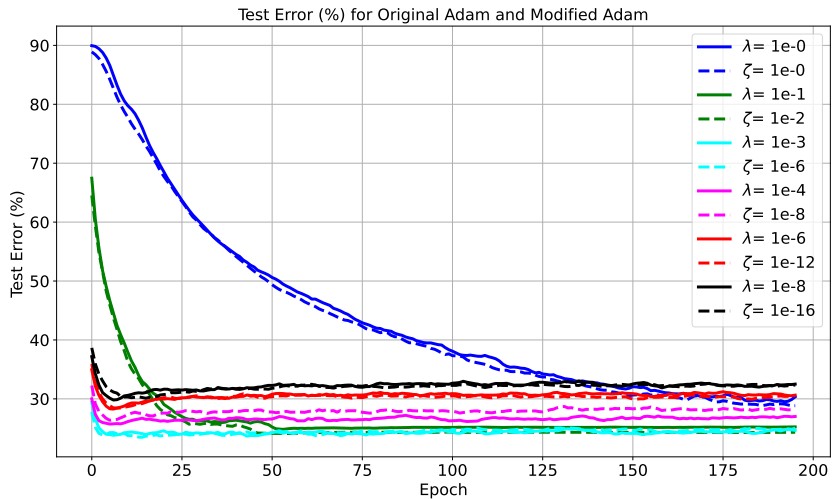

Figure 1: Test Error for Original Adam and Modified Adam. The stepsize in the original Adam is set to $\frac{\eta}{\sqrt{\boldsymbol{v}_t}+\lambda}$ and our stepsize is set to $\frac{\eta}{\sqrt{\boldsymbol{v}_t+\zeta}}$. The parameters are the same as CNN task in Fig. 1 of Li et al. (2023), where $\eta = 0.001, \beta_1 = 0.9, \beta_2 = 0.999$ and we build a six layers CNN for CIFAR 10.

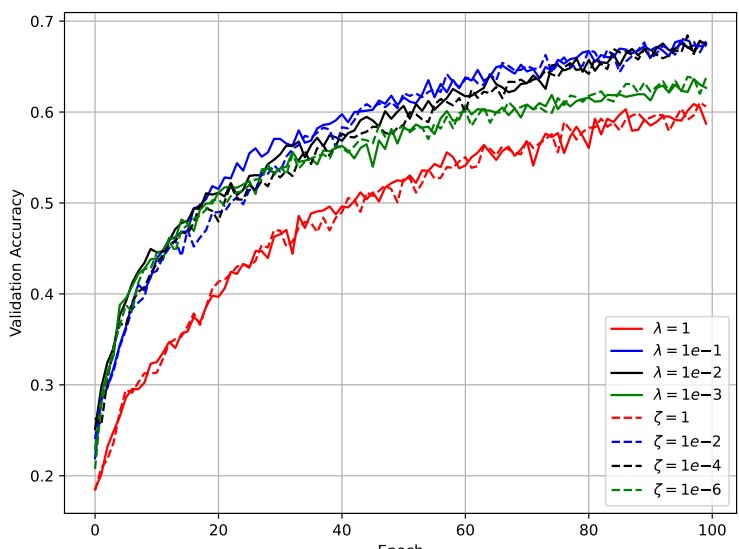

Figure 2: Test accuracy for Original Adam and Modified Adam. The stepsize in the original Adam is set to $\frac{\eta}{\sqrt{\boldsymbol{v}_t}+\lambda}$ and our stepsize is set to $\frac{\eta}{\sqrt{\boldsymbol{v}_t+\zeta}}$. We follow the setting in Yoshioka (2024) to build a vision-transformers for CIFAR 10. The stepsize is set to $\eta = 0.001, \beta_1 = 0.9, \beta_2 = 0.999$.

Both this paper and Wang et al. (2023a) extend the work of Wang et al. (2023b), which focused on the Adagrad algorithm. Our RMSProp analysis was developed concurrently with Wang et al. (2023a), but there are significant differences between their proofs and ours. Specifically, we consider the generalized smoothness objectives while Wang et al. (2023a) only considers the $L$-smooth objectives. Thus we choose a different surrogate, modify the Adam and bound the first-order term and its denominator by different

functions. When we extend our findings from RMSProp to Adam, we study a potential function $f(\boldsymbol{u}_t)$ with $\boldsymbol{u}_t = \frac{\boldsymbol{x}_t - \beta \boldsymbol{x}_{t-1}}{1-\beta}$, which was introduced by Liu et al. (2020) and has been extensively employed in the analysis of momentum-based algorithms. The only thing inspired by Wang et al. (2023a) in this paper is to set this $\beta = \frac{\beta_1}{\sqrt{\beta_2}}$. Wang et al. (2024) extends the work of Wang et al. (2023a) and is a concurrent work of this paper, which focuses on the scalar version of Adam with $(L_0, L_1)$-smooth objectives. This paper focuses on the per-coordinate version of Adam and our proof is different from Wang et al. (2023a; 2024).

## 6 Conclusion

In this paper, we provide tight convergence analyses for RMSProp and Adam for non-convex objectives under some of the most relaxed assumptions of generalized smoothness and affine gradient noise. The complexity to achieve an $\epsilon$-stationary point for both algorithms is shown to be $\mathcal{O}(\epsilon^{-4})$, which matches with the lower bound for first-order algorithms established in Arjevani et al. (2023). In the future, we will explore the convergence of the original Adam with the challenging $(L_0, L_1)$-smoothness condition and affine gradient noise variance.

## 7 Acknowledgment

The work of Q. Zhang and S. Zou was partially supported by the National Science Foundation under Grants ECCS-2438392 and ECCS-2501649. Y. Zhou's work was supported by the National Science Foundation under grants DMS-2134223, ECCS-2237830.

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

## A    Formal Version of Lemma 1 and Its Proof

**Lemma 5.** *Under Assumptions 2 and 3, for any $\alpha_0, \alpha_1 > 0, \boldsymbol{x}_0 = \boldsymbol{x}_1$ and $t > 0$, we have that*

$$
\mathbb{E}\left[\left\langle \nabla f(\boldsymbol{x}_t), \frac{\eta \boldsymbol{g}_t}{\sqrt{\boldsymbol{v}_t + \zeta}} - \frac{\eta \boldsymbol{g}_t}{\sqrt{\beta_2 \boldsymbol{v}_{t-1} + \zeta}} \right\rangle \Big| \mathcal{F}_t \right]
$$

$$
\geq - \left( \sum_{i=1}^{d} \frac{\eta(\partial_i f(\boldsymbol{x}_t))^2}{2\alpha_0 \sqrt{\beta_2 \boldsymbol{v}_{t-1,i} + \zeta}} + \sum_{i=1}^{d} \frac{\eta \alpha_0 D_0}{2} \mathbb{E}\left[ \frac{1}{\sqrt{\beta_2 \boldsymbol{v}_{t-1,i} + \zeta}} - \frac{1}{\sqrt{\boldsymbol{v}_{t,i} + \zeta}} \Big| \mathcal{F}_t \right] \right.
$$

$$
+ \sum_{i=1}^{d} \frac{\eta \alpha_0 D_1}{2} \mathbb{E}\left[ \frac{(\partial_i f(\boldsymbol{x}_{t-1}))^2}{\sqrt{\beta_2 \boldsymbol{v}_{t-1,i} + \zeta}} - \frac{(\partial_i f(\boldsymbol{x}_t))^2}{\sqrt{\boldsymbol{v}_{t,i} + \zeta}} \Big| \mathcal{F}_t \right]
$$

$$
+ \left. \sum_{i=1}^{d} \frac{\eta \alpha_0 D_1}{2\sqrt{\beta_2 \boldsymbol{v}_{t-1,i} + \zeta}} \left( \frac{(\partial_i f(\boldsymbol{x}_t))^2}{\alpha_1 D_1} + \frac{\alpha_1 d D_1 L_0^2 \eta^2}{1 - \beta_2} + \frac{2\sqrt{d}\eta L_1(\partial_i f(\boldsymbol{x}_t))^2}{\sqrt{1 - \beta_2}} \right) \right)
\tag{14}
$$

*Proof.* Since $\boldsymbol{v}_t = \beta_2 \boldsymbol{v}_{t-1} + (1 - \beta_2)\boldsymbol{g}_t \odot \boldsymbol{g}_t$, for any $i \in [d]$, we have that

$$
-\frac{1}{\sqrt{\boldsymbol{v}_{t,i} + \zeta}} + \frac{1}{\sqrt{\beta_2 \boldsymbol{v}_{t-1,i} + \zeta}}
$$

$$
= \frac{\boldsymbol{v}_{t,i} - \beta_2 \boldsymbol{v}_{t-1,i}}{\sqrt{\boldsymbol{v}_{t,i} + \zeta}\sqrt{\beta_2 \boldsymbol{v}_{t-1,i} + \zeta}(\sqrt{\boldsymbol{v}_{t,i} + \zeta} + \sqrt{\beta_2 \boldsymbol{v}_{t-1,i} + \zeta})}
$$

$$
= \frac{(1 - \beta_2)\boldsymbol{g}_{t,i}^2}{\sqrt{\boldsymbol{v}_{t,i} + \zeta}\sqrt{\beta_2 \boldsymbol{v}_{t-1,i} + \zeta}(\sqrt{\boldsymbol{v}_{t,i} + \zeta} + \sqrt{\beta_2 \boldsymbol{v}_{t-1,i} + \zeta})}.
\tag{15}
$$

Thus, the first-order.b term can be bounded as

$$
\mathbb{E}\left[\left\langle \nabla f(\boldsymbol{x}_t), \left[ \frac{-\eta}{\sqrt{\boldsymbol{v}_t + \zeta}} + \frac{\eta}{\sqrt{\beta_2 \boldsymbol{v}_{t-1} + \zeta}} \right] \odot \boldsymbol{g}_t \right\rangle \Big| \mathcal{F}_t \right]
$$

$$
\leq \sum_{i=1}^{d} \mathbb{E}\left[ \eta |\partial_i f(\boldsymbol{x}_t)| |\boldsymbol{g}_{t,i}| \left| \frac{-1}{\sqrt{\boldsymbol{v}_{t,i} + \zeta}} + \frac{1}{\sqrt{\beta_2 \boldsymbol{v}_{t-1,i} + \zeta}} \right| \Big| \mathcal{F}_t \right]
$$

$$
= \sum_{i=1}^{d} \mathbb{E}\left[ \eta |\partial_i f(\boldsymbol{x}_t)| |\boldsymbol{g}_{t,i}| \frac{(1 - \beta_2)\boldsymbol{g}_{t,i}^2}{\sqrt{\boldsymbol{v}_{t,i} + \zeta}\sqrt{\beta_2 \boldsymbol{v}_{t-1,i} + \zeta}(\sqrt{\boldsymbol{v}_{t,i} + \zeta} + \sqrt{\beta_2 \boldsymbol{v}_{t-1,i} + \zeta})} \Big| \mathcal{F}_t \right]
$$

$$
\leq \sum_{i=1}^{d} \frac{\eta |\partial_i f(\boldsymbol{x}_t)|}{\sqrt{\beta_2 \boldsymbol{v}_{t-1,i} + \zeta}} \mathbb{E}\left[ \frac{\sqrt{1 - \beta_2}\boldsymbol{g}_{t,i}^2}{(\sqrt{\boldsymbol{v}_{t,i} + \zeta} + \sqrt{\beta_2 \boldsymbol{v}_{t-1,i} + \zeta})} \Big| \mathcal{F}_t \right],
\tag{16}
$$

where the second equality is due to equation 15 and the last inequality is because $\frac{|\boldsymbol{g}_{t,i}|}{\sqrt{\boldsymbol{v}_{t,i} + \zeta}} \leq \frac{1}{\sqrt{1 - \beta_2}}$. For any $a, b \in \mathbb{R}$, we have $ab \leq \frac{a^2 + b^2}{2}$. Thus for any $\alpha_0 > 0$, the RHS of equation 16 can be further bounded as

$$
\frac{\eta |\partial_i f(\boldsymbol{x}_t)|}{\sqrt{\beta_2 \boldsymbol{v}_{t-1,i} + \zeta}} \mathbb{E}\left[ \frac{\sqrt{1 - \beta_2}\boldsymbol{g}_{t,i}^2}{(\sqrt{\boldsymbol{v}_{t,i} + \zeta} + \sqrt{\beta_2 \boldsymbol{v}_{t-1,i} + \zeta})} \Big| \mathcal{F}_t \right]
$$

$$
\leq \frac{\eta(\partial_i f(\boldsymbol{x}_t))^2}{2\alpha_0 \sqrt{\beta_2 \boldsymbol{v}_{t-1,i} + \zeta}} + \frac{\eta \alpha_0}{2\sqrt{\beta_2 \boldsymbol{v}_{t-1,i} + \zeta}} \left( \mathbb{E}\left[ \frac{\sqrt{1 - \beta_2}\boldsymbol{g}_{t,i}^2}{\sqrt{\boldsymbol{v}_{t,i} + \zeta} + \sqrt{\beta_2 \boldsymbol{v}_{t-1,i} + \zeta}} \Big| \mathcal{F}_t \right] \right)^2.
\tag{17}
$$

For the last term of equation 17, due to Hölder's inequality, we have that

$$
\frac{\eta}{2\sqrt{\beta_2 \boldsymbol{v}_{t-1,i} + \zeta}} \left( \mathbb{E}\left[ \frac{\sqrt{1 - \beta_2}\boldsymbol{g}_{t,i}^2}{\sqrt{\boldsymbol{v}_{t,i} + \zeta} + \sqrt{\beta_2 \boldsymbol{v}_{t-1,i} + \zeta}} \Big| \mathcal{F}_t \right] \right)^2
$$

$$\leq \frac{(1-\beta_2)\eta}{2\sqrt{\beta_2 \boldsymbol{v}_{t-1,i}+\zeta}} \mathbb{E}[\boldsymbol{g}_{t,i}^2|\mathcal{F}_t] \mathbb{E}\left[\frac{\boldsymbol{g}_{t,i}^2}{(\sqrt{\boldsymbol{v}_{t,i}+\zeta}+\sqrt{\beta_2 \boldsymbol{v}_{t-1,i}+\zeta})^2}\bigg|\mathcal{F}_t\right]$$

$$\leq \frac{\eta}{2}(D_0+D_1(\partial_i f(\boldsymbol{x}_t))^2) \mathbb{E}\left[\frac{(1-\beta_2)\boldsymbol{g}_{t,i}^2}{\sqrt{\boldsymbol{v}_{t,i}+\zeta}\sqrt{\beta_2 \boldsymbol{v}_{t-1,i}+\zeta}(\sqrt{\boldsymbol{v}_{t,i}+\zeta}+\sqrt{\beta_2 \boldsymbol{v}_{t-1,i}+\zeta})}\bigg|\mathcal{F}_t\right]$$

$$= \frac{\eta}{2}(D_0+D_1(\partial_i f(\boldsymbol{x}_t))^2) \mathbb{E}\left[\frac{\boldsymbol{v}_{t,i}+\zeta-(\beta_2 \boldsymbol{v}_{t-1,i}+\zeta)}{\sqrt{\boldsymbol{v}_{t,i}+\zeta}\sqrt{\beta_2 \boldsymbol{v}_{t-1,i}+\zeta}(\sqrt{\boldsymbol{v}_{t,i}+\zeta}+\sqrt{\beta_2 \boldsymbol{v}_{t-1,i}+\zeta})}\bigg|\mathcal{F}_t\right]$$

$$= \frac{\eta}{2}(D_0+D_1(\partial_i f(\boldsymbol{x}_t))^2) \mathbb{E}\left[\frac{1}{\sqrt{\beta_2 \boldsymbol{v}_{t-1,i}+\zeta}}-\frac{1}{\sqrt{\boldsymbol{v}_{t,i}+\zeta}}\bigg|\mathcal{F}_t\right], \tag{18}$$

where the second inequality is due to Assumption 2, and the fact that $\sqrt{v_{t,i}+\zeta}+\sqrt{\beta_2 v_{t-1,i}+\zeta} > \sqrt{v_{t,i}+\zeta}$. Thus for any $t \geq 1$, combining equation 16, equation 17 and equation 18, we have that

$$\mathbb{E}\left[\left\langle \nabla f(\boldsymbol{x}_t), \left[\frac{-\eta}{\sqrt{\boldsymbol{v}_t+\zeta}}+\frac{\eta}{\sqrt{\beta_2 \boldsymbol{v}_{t-1}+\zeta}}\right]\odot \boldsymbol{g}_t\right\rangle\bigg|\mathcal{F}_t\right]$$

$$\leq \sum_{i=1}^d \frac{\eta(\partial_i f(\boldsymbol{x}_t))^2}{2\alpha_0\sqrt{\beta_2 \boldsymbol{v}_{t-1,i}+\zeta}}+\sum_{i=1}^d \frac{\eta\alpha_0}{2}(D_0+D_1(\partial_i f(\boldsymbol{x}_t))^2) \mathbb{E}\left[\frac{1}{\sqrt{\beta_2 \boldsymbol{v}_{t-1,i}+\zeta}}-\frac{1}{\sqrt{\boldsymbol{v}_{t,i}+\zeta}}\bigg|\mathcal{F}_t\right]. \tag{19}$$

For $t > 1$, equation 18 can be rewritten as

$$\frac{(1-\beta_2)\eta}{2\sqrt{\beta_2 \boldsymbol{v}_{t-1,i}+\zeta}} \mathbb{E}[\boldsymbol{g}_{t,i}^2|\mathcal{F}_t] \mathbb{E}\left[\frac{\boldsymbol{g}_{t,i}^2}{(\sqrt{\boldsymbol{v}_{t,i}+\zeta}+\sqrt{\beta_2 \boldsymbol{v}_{t-1,i}+\zeta})^2}\bigg|\mathcal{F}_t\right]$$

$$\leq \frac{\eta D_0}{2}\mathbb{E}\left[\frac{1}{\sqrt{\beta_2 \boldsymbol{v}_{t-1,i}+\zeta}}-\frac{1}{\sqrt{\boldsymbol{v}_{t,i}+\zeta}}\bigg|\mathcal{F}_t\right]+\frac{\eta D_1}{2}\mathbb{E}\left[\frac{(\partial_i f(\boldsymbol{x}_{t-1}))^2}{\sqrt{\beta_2 \boldsymbol{v}_{t-1,i}+\zeta}}-\frac{(\partial_i f(\boldsymbol{x}_t))^2}{\sqrt{\boldsymbol{v}_{t,i}+\zeta}}\bigg|\mathcal{F}_t\right]$$

$$+\frac{\eta D_1}{2}\frac{(\partial_i f(\boldsymbol{x}_t))^2-(\partial_i f(\boldsymbol{x}_{t-1}))^2}{\sqrt{\beta_2 \boldsymbol{v}_{t-1,i}+\zeta}}. \tag{20}$$

For the last term in (20), by the fact that $f$ is $(L_0,L_1)$-smooth and for any $\alpha_1 > 0$, we have that

$$\frac{\eta D_1}{2}\frac{(\partial_i f(\boldsymbol{x}_t))^2-(\partial_i f(\boldsymbol{x}_{t-1}))^2}{\sqrt{\beta_2 \boldsymbol{v}_{t-1,i}+\zeta}}$$

$$\leq \frac{\eta D_1}{2}\frac{2|\partial_i f(\boldsymbol{x}_t)||\partial_i f(\boldsymbol{x}_t)-\partial_i f(\boldsymbol{x}_{t-1})|}{\sqrt{\beta_2 \boldsymbol{v}_{t-1,i}+\zeta}}$$

$$\leq \eta D_1 \frac{|\partial_i f(\boldsymbol{x}_t)|(L_0+L_1|\partial_i f(\boldsymbol{x}_t)|)\eta\left\|\frac{1}{\sqrt{\boldsymbol{v}_{t-1}+\zeta}}\odot \boldsymbol{g}_{t-1}\right\|}{\sqrt{\beta_2 \boldsymbol{v}_{t-1,i}+\zeta}}$$

$$\leq \frac{\eta D_1|\partial_i f(\boldsymbol{x}_t)|}{\sqrt{\beta_2 \boldsymbol{v}_{t-1,i}+\zeta}}\left(L_0\eta\left\|\frac{1}{\sqrt{\boldsymbol{v}_{t-1}+\zeta}}\odot \boldsymbol{g}_{t-1}\right\|+L_1|\partial_i f(\boldsymbol{x}_t)|\eta\left\|\frac{1}{\sqrt{\boldsymbol{v}_{t-1}+\zeta}}\odot \boldsymbol{g}_{t-1}\right\|\right)$$

$$\leq \frac{\eta D_1}{2\sqrt{\beta_2 \boldsymbol{v}_{t-1,i}+\zeta}}\left(\frac{(\partial_i f(\boldsymbol{x}_t))^2}{\alpha_1 D_1}+\alpha_1 D_1 L_0^2\eta^2\sum_{i=1}^d \frac{(\boldsymbol{g}_{t-1,i})^2}{\boldsymbol{v}_{t-1,i}+\zeta}+\frac{2\sqrt{d}\eta L_1(\partial_i f(\boldsymbol{x}_t))^2}{\sqrt{1-\beta_2}}\right)$$

$$\leq \frac{\eta D_1}{2\sqrt{\beta_2 \boldsymbol{v}_{t-1,i}+\zeta}}\left(\frac{(\partial_i f(\boldsymbol{x}_t))^2}{\alpha_1 D_1}+\frac{\alpha_1 d D_1 L_0^2\eta^2}{1-\beta_2}+\frac{2\sqrt{d}\eta L_1(\partial_i f(\boldsymbol{x}_t))^2}{\sqrt{1-\beta_2}}\right). \tag{21}$$

where the first inequality is due to the fact that for any $a,b$, we have $a^2-b^2 \leq 2|a||a-b|$, the second inequality is by the $(L_0,L_1)$-smoothness and the third one is because that $ab \leq \frac{a^2+b^2}{2}$ for any $a,b$ and

$\frac{|\boldsymbol{g}_{t,i}|}{\sqrt{\boldsymbol{v}_{t,i}+\zeta}} \leq \frac{1}{\sqrt{1-\beta_2}}$. Based on (16), (17), (20) and (21), for $t > 1$ we can get that

$$
\mathbb{E}\left[\left\langle \nabla f(\boldsymbol{x}_t), \left[\frac{-\eta \boldsymbol{g}_t}{\sqrt{\boldsymbol{v}_t+\zeta}} + \frac{\eta \boldsymbol{g}_t}{\sqrt{\beta_2 \boldsymbol{v}_{t-1}+\zeta}}\right]\right\rangle \middle| \mathcal{F}_t\right]
$$

$$
\leq \sum_{i=1}^{d} \frac{\eta(\partial_i f(\boldsymbol{x}_t))^2}{2\alpha_0 \sqrt{\beta_2 \boldsymbol{v}_{t-1,i}+\zeta}} + \sum_{i=1}^{d} \frac{\eta \alpha_0 D_0}{2} \mathbb{E}\left[\left.\frac{1}{\sqrt{\beta_2 \boldsymbol{v}_{t-1,i}+\zeta}} - \frac{1}{\sqrt{\boldsymbol{v}_{t,i}+\zeta}}\right| \mathcal{F}_t\right]
$$

$$
+ \sum_{i=1}^{d} \frac{\eta \alpha_0 D_1}{2} \mathbb{E}\left[\left.\frac{(\partial_i f(\boldsymbol{x}_{t-1}))^2}{\sqrt{\beta_2 \boldsymbol{v}_{t-1,i}+\zeta}} - \frac{(\partial_i f(\boldsymbol{x}_t))^2}{\sqrt{\boldsymbol{v}_{t,i}+\zeta}}\right| \mathcal{F}_t\right]
$$

$$
+ \sum_{i=1}^{d} \frac{\eta \alpha_0 D_1}{2\sqrt{\beta_2 \boldsymbol{v}_{t-1,i}+\zeta}} \left(\frac{(\partial_i f(\boldsymbol{x}_t))^2}{\alpha_1 D_1} + \frac{\alpha_1 d D_1 L_0^2 \eta^2}{1-\beta_2} + \frac{2\sqrt{d}\eta L_1(\partial_i f(\boldsymbol{x}_t))^2}{\sqrt{1-\beta_2}}\right). \tag{22}
$$

Since we set $\boldsymbol{x}_0 = \boldsymbol{x}_1$, the RHS of equation 22 is an upper bound on the RHS of equation 19. As a result, equation 22 also holds for $t = 1$. We then complete the proof. $\qquad\square$

## B Proof of Lemma 2

For $(L_0, L_1)$-smooth objective functions, we have the following descent inequality (Lemma 1 in Crawshaw et al. (2022)):

$$
\underbrace{\mathbb{E}\left[\left\langle \nabla f(\boldsymbol{x}_t), \frac{\eta}{\sqrt{\tilde{\boldsymbol{v}}_t+\zeta}} \odot \boldsymbol{g}_t\right\rangle \middle| \mathcal{F}_t\right]}_{\text{first-order.a}}
$$

$$
\leq f(\boldsymbol{x}_t) - \mathbb{E}[f(\boldsymbol{x}_{t+1})|\mathcal{F}_t] + \underbrace{\sum_{i=1}^{d} \frac{L_0}{2\sqrt{d}} \mathbb{E}[\|\boldsymbol{x}_{t+1} - \boldsymbol{x}_t\| |\boldsymbol{x}_{t+1,i} - \boldsymbol{x}_{t,i}| |\mathcal{F}_t]}_{\text{second-order}}
$$

$$
+ \underbrace{\sum_{i=1}^{d} \frac{L_1|\partial_i f(\boldsymbol{x}_t)|}{2} \mathbb{E}[\|\boldsymbol{x}_{t+1} - \boldsymbol{x}_t\| |\boldsymbol{x}_{t+1,i} - \boldsymbol{x}_{t,i}| |\mathcal{F}_t]}_{\text{additional term}}
$$

$$
- \underbrace{\mathbb{E}\left[\left\langle \nabla f(\boldsymbol{x}_t), \left[\frac{\eta}{\sqrt{\boldsymbol{v}_t+\zeta}} + \frac{-\eta}{\sqrt{\tilde{\boldsymbol{v}}_t+\zeta}}\right] \odot \boldsymbol{g}_t\right\rangle \middle| \mathcal{F}_t\right]}_{\text{first-order.b}}. \tag{23}
$$

For the first-order.a item, given $\mathcal{F}_t$, we have $\boldsymbol{g}_t$ independent of $\tilde{\boldsymbol{v}}_t$ and $\boldsymbol{x}_t$. It then follows that

$$
\mathbb{E}\left[\left\langle \nabla f(\boldsymbol{x}_t), \frac{\eta}{\sqrt{\tilde{\boldsymbol{v}}_t+\zeta}} \odot \boldsymbol{g}_t\right\rangle \middle| \mathcal{F}_t\right] = \sum_{i=1}^{d} \frac{\eta(\partial_i f(\boldsymbol{x}_t))^2}{\sqrt{\beta_2 \boldsymbol{v}_{t-1,i}+\zeta}}.
$$

Based on Lemma 1, we provide a lower bound on the first-order.b term. Plugging Lemma 1 to equation 23, we have the following inequality for any $\alpha_0, \alpha_1 > 0$ and $t > 1$:

$$
\sum_{i=1}^{d} \frac{\eta(\partial_i f(\boldsymbol{x}_t))^2}{\sqrt{\beta_2 \boldsymbol{v}_{t-1,i}+\zeta}}
$$

$$
\leq f(\boldsymbol{x}_t) - \mathbb{E}[f(\boldsymbol{x}_{t+1})|\mathcal{F}_t] + \underbrace{\sum_{i=1}^{d} \frac{L_0}{2\sqrt{d}} \mathbb{E}[\|\boldsymbol{x}_{t+1} - \boldsymbol{x}_t\| |\boldsymbol{x}_{t+1,i} - \boldsymbol{x}_{t,i}| |\mathcal{F}_t]}_{\text{second-order}}
$$

$$+ \underbrace{\sum_{i=1}^{d} \frac{L_1 |\partial_i f(\boldsymbol{x}_t)|}{2} \mathbb{E}[\|\boldsymbol{x}_{t+1} - \boldsymbol{x}_t\| \|\boldsymbol{x}_{t+1,i} - \boldsymbol{x}_{t,i}\| | \mathcal{F}_t]}_{\text{additional term}}$$

$$+ \sum_{i=1}^{d} \frac{\eta (\partial_i f(\boldsymbol{x}_t))^2}{2\alpha_0 \sqrt{\beta_2 \boldsymbol{v}_{t-1,i} + \zeta}} + \sum_{i=1}^{d} \frac{\eta \alpha_0 D_0}{2} \mathbb{E}\left[ \left| \frac{1}{\sqrt{\beta_2 \boldsymbol{v}_{t-1,i} + \zeta}} - \frac{1}{\sqrt{\boldsymbol{v}_{t,i} + \zeta}} \right| \Big| \mathcal{F}_t \right]$$

$$+ \sum_{i=1}^{d} \frac{\eta \alpha_0 D_1}{2} \mathbb{E}\left[ \frac{(\partial_i f(\boldsymbol{x}_{t-1}))^2}{\sqrt{\beta_2 \boldsymbol{v}_{t-1,i} + \zeta}} - \frac{(\partial_i f(\boldsymbol{x}_t))^2}{\sqrt{\boldsymbol{v}_{t,i} + \zeta}} \Big| \mathcal{F}_t \right]$$

$$+ \sum_{i=1}^{d} \frac{\eta \alpha_0 D_1}{2\sqrt{\beta_2 \boldsymbol{v}_{t-1,i} + \zeta}} \left( \frac{(\partial_i f(\boldsymbol{x}_t))^2}{\alpha_1 D_1} + \frac{\alpha_1 d D_1 L_0^2 \eta^2}{1 - \beta_2} + \frac{2\sqrt{d} \eta L_1 (\partial_i f(\boldsymbol{x}_t))^2}{\sqrt{1 - \beta_2}} \right). \tag{24}$$

Now we focus on the second-order term, which can be bounded as follows:

$$\sum_{i=1}^{d} \frac{L_0}{2\sqrt{d}} \mathbb{E}\left[ \|\boldsymbol{x}_{t+1} - \boldsymbol{x}_t\| \|\boldsymbol{x}_{t+1,i} - \boldsymbol{x}_{t,i}\| | \mathcal{F}_t \right]$$

$$\leq \sum_{i=1}^{d} \frac{L_0}{2\sqrt{d}} \mathbb{E}\left[ \frac{\|\boldsymbol{x}_{t+1} - \boldsymbol{x}_t\|^2}{2\sqrt{d}} + \frac{\sqrt{d}}{2} (\boldsymbol{x}_{t+1,i} - \boldsymbol{x}_{t,i})^2 \Big| \mathcal{F}_t \right]$$

$$= \sum_{i=1}^{d} \frac{L_0}{2} \mathbb{E}\left[ \eta^2 \frac{\boldsymbol{g}_{t,i}^2}{\boldsymbol{v}_{t,i} + \zeta} \Big| \mathcal{F}_t \right]$$

$$\leq \sum_{i=1}^{d} \frac{L_0}{2} \mathbb{E}\left[ \eta^2 \frac{\boldsymbol{g}_{t,i}^2}{\beta_2 \boldsymbol{v}_{t-1,i} + \zeta} \Big| \mathcal{F}_t \right]$$

$$\leq \sum_{i=1}^{d} \frac{L_0 \eta^2}{2} \frac{D_0 + D_1 (\partial_i f(\boldsymbol{x}_t))^2}{\beta_2 \boldsymbol{v}_{t-1,i} + \zeta}$$

$$\leq \sum_{i=1}^{d} \frac{L_0 \eta^2}{2\sqrt{\zeta}} \frac{D_0 + D_1 (\partial_i f(\boldsymbol{x}_t))^2}{\sqrt{\beta_2 \boldsymbol{v}_{t-1,i} + \zeta}}, \tag{25}$$

where the fourth inequality is due to Assumption 2. Based on Assumption 2, for any $\alpha_2 > 0$, the addition term can be bounded as follows

$$\frac{L_1 |\partial_i f(\boldsymbol{x}_t)|}{2} \mathbb{E}[\|\boldsymbol{x}_{t+1} - \boldsymbol{x}_t\| \|\boldsymbol{x}_{t+1,i} - \boldsymbol{x}_{t,i}\| | \mathcal{F}_t]$$

$$\leq \frac{\sqrt{d} L_1 |\partial_i f(\boldsymbol{x}_t)|}{2\sqrt{1 - \beta_2}} \mathbb{E}\left[ \eta^2 \frac{|\boldsymbol{g}_{t,i}|}{\sqrt{\beta_2 \boldsymbol{v}_{t-1,i} + \zeta}} \Big| \mathcal{F}_t \right]$$

$$= \frac{\eta^2 \sqrt{d} L_1 |\partial_i f(\boldsymbol{x}_t)|}{2\sqrt{1 - \beta_2} \sqrt{\beta_2 \boldsymbol{v}_{t-1,i} + \zeta}} \sqrt{\mathbb{E}\left[ \boldsymbol{g}_{t,i}^2 | \mathcal{F}_t \right]}$$

$$\leq \frac{\eta^2 \sqrt{d} L_1 |\partial_i f(\boldsymbol{x}_t)| (\sqrt{D_0} + \sqrt{D_1} |\partial_i f(\boldsymbol{x}_t)|)}{2\sqrt{1 - \beta_2} \sqrt{\beta_2 \boldsymbol{v}_{t-1,i} + \zeta}}$$

$$\leq \frac{\eta^2 \sqrt{d} L_1 \sqrt{D_1} (\partial_i f(\boldsymbol{x}_t))^2}{2\sqrt{1 - \beta_2} \sqrt{\beta_2 \boldsymbol{v}_{t-1,i} + \zeta}} + \frac{\eta^2 \sqrt{d} (\partial_i f(\boldsymbol{x}_t))^2}{2(1 - \beta_2) \alpha_2 \sqrt{\beta_2 \boldsymbol{v}_{t-1,i} + \zeta}} + \frac{\eta^2 \sqrt{d} \alpha_2 L_1^2 D_0}{2\sqrt{\beta_2 \boldsymbol{v}_{t-1,i} + \zeta}}, \tag{26}$$

where the first inequality is due to the fact that $\|\boldsymbol{x}_{t+1} - \boldsymbol{x}_t\| \leq \frac{\sqrt{d}\eta}{\sqrt{1-\beta_2}}$, the second inequality is because that $\mathbb{E}[\boldsymbol{g}_{t,i} | \mathcal{F}_t] \leq \sqrt{\mathbb{E}[\boldsymbol{g}_{t,i}^2 | \mathcal{F}_t]}$, and the third inequality is due to Assumption 3 and the fact that for any $a, b \geq 0$, we have $\sqrt{a + b} \leq \sqrt{a} + \sqrt{b}$. Plug (25) and (26) into (24), and it follows that

$$\sum_{i=1}^{d} \frac{\eta (\partial_i f(\boldsymbol{x}_t))^2}{\sqrt{\beta_2 \boldsymbol{v}_{t-1,i} + \zeta}}$$

$$\leq f(\boldsymbol{x}_t) - \mathbb{E}[f(\boldsymbol{x}_{t+1})|\mathcal{F}_t] + \sum_{i=1}^{d} \frac{L_0 \eta^2}{2\sqrt{\zeta}} \frac{D_0 + D_1(\partial_i f(\boldsymbol{x}_t))^2}{\sqrt{\beta_2 \boldsymbol{v}_{t-1,i} + \zeta}}$$

$$+ \sum_{i=1}^{d} \left( \frac{\eta^2 \sqrt{d} L_1 \sqrt{D_1} (\partial_i f(\boldsymbol{x}_t))^2}{2\sqrt{1-\beta_2}\sqrt{\beta_2 \boldsymbol{v}_{t-1,i} + \zeta}} + \frac{\eta^2 \sqrt{d} (\partial_i f(\boldsymbol{x}_t))^2}{2(1-\beta_2)\alpha_2 \sqrt{\beta_2 \boldsymbol{v}_{t-1,i} + \zeta}} + \frac{\eta^2 \sqrt{d} \alpha_2 L_1^2 D_0}{2\sqrt{\beta_2 \boldsymbol{v}_{t-1,i} + \zeta}} \right)$$

$$+ \sum_{i=1}^{d} \frac{\eta (\partial_i f(\boldsymbol{x}_t))^2}{2\alpha_0 \sqrt{\beta_2 \boldsymbol{v}_{t-1,i} + \zeta}} + \sum_{i=1}^{d} \frac{\eta \alpha_0 D_0}{2} \mathbb{E}\left[ \frac{1}{\sqrt{\beta_2 \boldsymbol{v}_{t-1,i} + \zeta}} - \frac{1}{\sqrt{\boldsymbol{v}_{t,i} + \zeta}} \middle| \mathcal{F}_t \right]$$

$$+ \sum_{i=1}^{d} \frac{\eta \alpha_0 D_1}{2} \mathbb{E}\left[ \frac{(\partial_i f(\boldsymbol{x}_{t-1}))^2}{\sqrt{\beta_2 \boldsymbol{v}_{t-1,i} + \zeta}} - \frac{(\partial_i f(\boldsymbol{x}_t))^2}{\sqrt{\boldsymbol{v}_{t,i} + \zeta}} \middle| \mathcal{F}_t \right]$$

$$+ \sum_{i=1}^{d} \frac{\eta \alpha_0 D_1}{2\sqrt{\beta_2 \boldsymbol{v}_{t-1,i} + \zeta}} \left( \frac{(\partial_i f(\boldsymbol{x}_t))^2}{\alpha_1 D_1} + \frac{\alpha_1 d D_1 L_0^2 \eta^2}{1-\beta_2} + \frac{2\sqrt{d}\eta L_1(\partial_i f(\boldsymbol{x}_t))^2}{\sqrt{1-\beta_2}} \right), \tag{27}$$

which can be written as

$$\sum_{i=1}^{d} \left( \eta - \frac{\eta}{2\alpha_0} - \frac{\eta \alpha_0}{2\alpha_1} - \frac{L_0 \eta^2 D_1}{2\sqrt{\zeta}} - \frac{\eta^2 L_1 \sqrt{d D_1}}{2\sqrt{1-\beta_2}} - \frac{\eta^2 \sqrt{d}}{2(1-\beta_2)\alpha_2} - \frac{\eta^2 \sqrt{d} \alpha_0 L_1 D_1}{\sqrt{1-\beta_2}} \right) \frac{(\partial_i f(\boldsymbol{x}_t))^2}{\sqrt{\beta_2 \boldsymbol{v}_{t-1,i} + \zeta}}$$

$$\leq f(\boldsymbol{x}_t) - \mathbb{E}[f(\boldsymbol{x}_{t+1})|\mathcal{F}_t] + \frac{d L_0 \eta^2 D_0}{2\zeta} + \frac{\alpha_0 \alpha_1 d^2 L_0^2 \eta^3 D_1^2}{2(1-\beta_2)} \frac{1}{\sqrt{\zeta}} + \frac{\eta^2 \alpha_2 d^{1.5} L_1^2 D_0}{2\sqrt{\zeta}}$$

$$+ \sum_{i=1}^{d} \frac{\eta \alpha_0 D_0}{2} \mathbb{E}\left[ \frac{1}{\sqrt{\beta_2 \boldsymbol{v}_{t-1,i} + \zeta}} - \frac{1}{\sqrt{\boldsymbol{v}_{t,i} + \zeta}} \middle| \mathcal{F}_t \right]$$

$$+ \sum_{i=1}^{d} \frac{\eta \alpha_0 D_1}{2} \mathbb{E}\left[ \frac{(\partial_i f(\boldsymbol{x}_{t-1}))^2}{\sqrt{\beta_2 \boldsymbol{v}_{t-1,i} + \zeta}} - \frac{(\partial_i f(\boldsymbol{x}_t))^2}{\sqrt{\boldsymbol{v}_{t,i} + \zeta}} \middle| \mathcal{F}_t \right]. \tag{28}$$

It is worth noting that the sum of the last two terms in equation 28 from $t = 1$ to $T$ can be further bounded. Specifically, for any $i \in [d]$, taking the expectation with respect to $\mathcal{F}_t$, and the sum from $t = 1$ to $T$, we have that

$$\sum_{t=1}^{T} \mathbb{E}\left[ \frac{1}{\sqrt{\beta_2 \boldsymbol{v}_{t-1,i} + \zeta}} - \frac{1}{\sqrt{\boldsymbol{v}_{t,i} + \zeta}} \right]$$

$$= \mathbb{E}\left[ \frac{1}{\sqrt{\beta_2 \boldsymbol{v}_{0,i} + \zeta}} \right] + \sum_{t=1}^{T-1} \mathbb{E}\left[ \frac{1}{\sqrt{\beta_2 \boldsymbol{v}_{t,i} + \zeta}} - \frac{1}{\sqrt{\boldsymbol{v}_{t,i} + \zeta}} \right] - \mathbb{E}\left[ \frac{1}{\sqrt{\boldsymbol{v}_{T,i} + \zeta}} \right]$$

$$\leq \frac{1}{\sqrt{\zeta}} + \sum_{t=1}^{T-1} \mathbb{E}\left[ \frac{1}{\sqrt{\beta_2 \boldsymbol{v}_{t,i} + \zeta}} - \frac{\sqrt{\beta_2}}{\sqrt{\beta_2 \boldsymbol{v}_{t,i} + \zeta}} \right]$$

$$= \frac{1}{\sqrt{\zeta}} + \sum_{t=1}^{T-1} \mathbb{E}\left[ \frac{1 - \sqrt{\beta_2}}{\sqrt{\beta_2 \boldsymbol{v}_{t,i} + \zeta}} \right]$$

$$\leq \frac{1}{\sqrt{\zeta}} + T \frac{1 - \sqrt{\beta_2}}{\sqrt{\zeta}}. \tag{29}$$

Similarly, for the last term in equation 28, the sum from $t = 1$ to $T$ can be bounded as follows

$$\sum_{t=1}^{T} \mathbb{E}\left[ \frac{(\partial_i f(\boldsymbol{x}_{t-1}))^2}{\sqrt{\beta_2 \boldsymbol{v}_{t-1,i} + \zeta}} - \frac{(\partial_i f(\boldsymbol{x}_t))^2}{\sqrt{\boldsymbol{v}_{t,i} + \zeta}} \right]$$

$$= \mathbb{E}\left[ \frac{(\partial_i f(\boldsymbol{x}_0))^2}{\sqrt{\beta_2 \boldsymbol{v}_{0,i} + \zeta}} - \frac{(\partial_i f(\boldsymbol{x}_1))^2}{\sqrt{\boldsymbol{v}_{1,i} + \zeta}} \right] + \sum_{t=2}^{T} \mathbb{E}\left[ \frac{(\partial_i f(\boldsymbol{x}_{t-1}))^2}{\sqrt{\beta_2 \boldsymbol{v}_{t-1,i} + \zeta}} - \frac{(\partial_i f(\boldsymbol{x}_t))^2}{\sqrt{\boldsymbol{v}_{t,i} + \zeta}} \right]$$

$$= \mathbb{E}\left[ \frac{(\partial_i f(\boldsymbol{x}_1))^2}{\sqrt{\beta_2 \boldsymbol{v}_{0,i} + \zeta}} \right] + \sum_{t=1}^{T-1} \mathbb{E}\left[ \frac{(\partial_i f(\boldsymbol{x}_t))^2}{\sqrt{\beta_2 \boldsymbol{v}_{t,i} + \zeta}} - \frac{(\partial_i f(\boldsymbol{x}_t))^2}{\sqrt{\boldsymbol{v}_{t,i} + \zeta}} \right] - \mathbb{E}\left[ \frac{(\partial_i f(\boldsymbol{x}_T))^2}{\sqrt{\boldsymbol{v}_{T,i} + \zeta}} \right]$$

$$
\begin{aligned}
\leq & \mathbb{E}\left[\frac{(\partial_i f(\boldsymbol{x}_1))^2}{\sqrt{\zeta}}\right] + \sum_{t=1}^{T-1} \mathbb{E}\left[\left(\frac{1}{\sqrt{\beta_2 \boldsymbol{v}_{t,i}} + \zeta} - \frac{\sqrt{\beta_2}}{\sqrt{\beta_2 \boldsymbol{v}_{t,i}} + \zeta}\right)(\partial_i f(\boldsymbol{x}_t))^2\right] \\
= & \frac{(\partial_i f(\boldsymbol{x}_1))^2}{\sqrt{\zeta}} + \sum_{t=1}^{T-1}(1 - \sqrt{\beta_2})\mathbb{E}\left[\frac{(\partial_i f(\boldsymbol{x}_t))^2}{\sqrt{\beta_2 \boldsymbol{v}_{t,i}} + \zeta}\right] \\
\leq & \frac{(\partial_i f(\boldsymbol{x}_1))^2}{\sqrt{\zeta}} + \sum_{t=1}^{T-1}\frac{1 - \sqrt{\beta_2}}{\sqrt{\beta_2}}\mathbb{E}\left[\frac{(\partial_i f(\boldsymbol{x}_t))^2}{\sqrt{\boldsymbol{v}_{t,i}} + \zeta}\right] \\
\leq & \frac{(\partial_i f(\boldsymbol{x}_1))^2}{\sqrt{\zeta}} + \sum_{t=1}^{T-1}(1 - \beta_2)\mathbb{E}\left[\frac{(\partial_i f(\boldsymbol{x}_t))^2}{\sqrt{\beta_2 \boldsymbol{v}_{t-1,i}} + \zeta}\right],
\end{aligned}
\tag{30}
$$

since $\boldsymbol{x}_0 = \boldsymbol{x}_1$, $\boldsymbol{g}_0 = 0$ and the last inequality is due to the fact that $\frac{1}{\sqrt{\beta_2}} \leq 1 + \sqrt{\beta_2}$. By taking expectations and sums of (28) from $t = 1$ to $T$, based on (29) and (30), we have that

$$
\begin{aligned}
& \sum_{t=1}^{T}\sum_{i=1}^{d}\left(\eta - \frac{\eta}{2\alpha_0} - \frac{\eta\alpha_0}{2\alpha_1} - \frac{L_0\eta^2 D_1}{2\sqrt{\zeta}} - \frac{\eta^2 L_1\sqrt{dD_1}}{2\sqrt{1-\beta_2}} - \frac{\eta^2\sqrt{d}}{2(1-\beta_2)\alpha_2} - \frac{\eta^2\sqrt{d}\alpha_0 L_1 D_1}{\sqrt{1-\beta_2}}\right)\frac{(\partial_i f(\boldsymbol{x}_t))^2}{\sqrt{\beta_2 \boldsymbol{v}_{t-1,i}} + \zeta} \\
\leq & f(\boldsymbol{x}_1) - \mathbb{E}[f(\boldsymbol{x}_{T+1})|\mathcal{F}_t] + \frac{\eta\alpha_0 D_1\|\nabla f(\boldsymbol{x}_1)\|^2}{2\sqrt{\zeta}} + \frac{\eta\alpha_0 dD_0}{2\sqrt{\zeta}} + T\frac{dL_0\eta^2 D_0}{2\zeta} + T\frac{\alpha_0\alpha_1 d^2 L_0^2\eta^3 D_1^2}{2(1-\beta_2)}\frac{1}{\sqrt{\zeta}} \\
& + T\frac{\eta^2\alpha_2 d^{1.5}L_1^2 D_0}{2\sqrt{\zeta}} + T\frac{\eta\alpha_0 dD_0(1-\sqrt{\beta_2})}{2\sqrt{\zeta}} + \frac{\eta\alpha_0 D_1(1-\beta_2)}{2}\sum_{i=1}^{d}\sum_{t=1}^{T}\mathbb{E}\left[\frac{\|\partial_i f(\boldsymbol{x}_t)\|^2}{\sqrt{\beta_2 \boldsymbol{v}_{t-1,i}} + \zeta}\right].
\end{aligned}
\tag{31}
$$

Define $\Delta = f(\boldsymbol{x}_1) - f(\boldsymbol{x}^*) + \frac{\eta\alpha_0 dD_0}{2\sqrt{\zeta}} + \frac{\eta\alpha_0 D_1\|\nabla f(\boldsymbol{x}_1)\|^2}{2\sqrt{\zeta}}$, where $f(\boldsymbol{x}^*) = \inf_{\boldsymbol{x}} f(\boldsymbol{x})$. If we set $\alpha_0 = 1, \alpha_1 = 7$ and $\alpha_2 = 1$, obviously we can find some $1 - \beta_2 = \min\left(\frac{1}{7D_1}, \frac{\sqrt{\zeta}\epsilon^2}{35dD_0}\right) = \mathcal{O}(\epsilon^2)$, and

$$
\eta \leq \min\left(\frac{\sqrt{\zeta}}{7L_0 D_1}, \frac{\sqrt{1-\beta_2}}{\max\left(14L_1\sqrt{d}D_1, 7L_1\sqrt{dD_1}\right)}, \frac{1-\beta_2}{7\sqrt{d}}, \frac{\zeta\epsilon^2}{35L_0 dD_0}, \frac{\epsilon\sqrt{1-\beta_2}\sqrt[4]{\zeta}}{7D_1 L_0 d\sqrt{5}}, \frac{\sqrt{\zeta}\epsilon^2}{35L_1^2 d^{1.5}D_0}\right) = \mathcal{O}(\epsilon^2)
$$

such that

$$
\frac{\eta}{14}\sum_{i=1}^{d}\sum_{t=1}^{T}\mathbb{E}\left[\frac{(\partial_i f(\boldsymbol{x}_t))^2}{\sqrt{\beta_2 \boldsymbol{v}_{t-1,i}} + \zeta}\right] \leq \Delta + T\frac{\eta}{70}\epsilon^2 + T\frac{\eta}{70}\epsilon^2 + T\frac{\eta}{70}\epsilon^2 + T\frac{\eta}{70}\epsilon^2,
\tag{32}
$$

Set $T \geq \frac{70\Delta}{\eta\epsilon^2} = \mathcal{O}(\epsilon^{-4})$, and we have that

$$
\frac{1}{T}\sum_{t=1}^{T}\mathbb{E}\left[\frac{\|\nabla f(\boldsymbol{x}_t)\|^2}{\sqrt{\beta_2\|\boldsymbol{v}_{t-1}\|} + \zeta}\right] \leq \sum_{i=1}^{d}\frac{1}{T}\sum_{t=1}^{T}\mathbb{E}\left[\frac{(\partial_i f(\boldsymbol{x}_t))^2}{\sqrt{\beta_2 \boldsymbol{v}_{t-1,i}} + \zeta}\right] \leq \epsilon^2.
\tag{33}
$$

This completes the proof.

## C   Proof of Lemma 3

For any $a, b \geq 0$, we have that $\sqrt{a + b} \leq \sqrt{a} + \sqrt{b}$. It then follows that

$$
\mathbb{E}\left[\sqrt{\beta_2\|\boldsymbol{v}_{t-1}\|} + \zeta\right] \leq \sum_{i=1}^{d}\mathbb{E}\left[\sqrt{\beta_2 \boldsymbol{v}_{t-1,i}}\right] + \sqrt{\zeta}.
\tag{34}
$$

For the first term $\mathbb{E}[\sqrt{\beta_2 \boldsymbol{v}_{t-1,i}}]$ and $t > 1$ we have that

$$
\mathbb{E}[\sqrt{\beta_2 \boldsymbol{v}_{t-1,i}}] = \mathbb{E}\left[\mathbb{E}\left[\sqrt{\beta_2 \boldsymbol{v}_{t-1,i}}\Big|\mathcal{F}_{t-1}\right]\right] = \mathbb{E}\left[\mathbb{E}\left[\sqrt{\beta_2^2 \boldsymbol{v}_{t-2,i} + \beta_2(1-\beta_2)(\boldsymbol{g}_{t-1,i})^2}\Big|\mathcal{F}_{t-1}\right]\right].
\tag{35}
$$

Given $\mathcal{F}_{t-1}$, $\boldsymbol{v}_{t-2}$ is deterministic. By Jensen's inequality, we have that

$$\mathbb{E}\left[\mathbb{E}\left[\sqrt{\beta_2^2 \boldsymbol{v}_{t-2,i} + \beta_2(1-\beta_2)(\boldsymbol{g}_{t-1,i})^2}\Big|\mathcal{F}_{t-1}\right]\right]$$

$$\leq \mathbb{E}\left[\sqrt{\mathbb{E}\left[\beta_2^2 \boldsymbol{v}_{t-2,i} + \beta_2(1-\beta_2)(\boldsymbol{g}_{t-1,i})^2\big|\mathcal{F}_{t-1}\right]}\right]$$

$$\leq \mathbb{E}\left[\sqrt{\beta_2^2 \boldsymbol{v}_{t-2,i} + \beta_2(1-\beta_2)(D_0 + D_1(\partial_i f(\boldsymbol{x}_{t-1}))^2)}\right]$$

$$\leq \mathbb{E}\left[\sqrt{\beta_2^2 \boldsymbol{v}_{t-2,i} + \beta_2(1-\beta_2)D_0}\right] + \mathbb{E}\left[\sqrt{\beta_2(1-\beta_2)D_1}|\partial_i f(\boldsymbol{x}_{t-1,i})|\right], \tag{36}$$

where the second inequality is according to Assumption 2. By recursively applying (36) we have that

$$\mathbb{E}\left[\sqrt{\beta_2^2 \boldsymbol{v}_{t-2} + \beta_2(1-\beta_2)D_0}\right]$$

$$\leq \mathbb{E}\left[\sqrt{\beta_2^3 \boldsymbol{v}_{t-3,i} + (\beta_2 + \beta_2^2)(1-\beta_2)D_0}\right] + \mathbb{E}\left[\sqrt{\beta_2^2(1-\beta_2)D_1}|\partial_i f(\boldsymbol{x}_{t-2})|\right]. \tag{37}$$

Specifically, we can get that

$$\mathbb{E}\left[\sqrt{\beta_2 \boldsymbol{v}_{t-1,i}}\right] \leq \mathbb{E}\left[\sqrt{\beta_2^2 \boldsymbol{v}_{t-2,i} + \beta_2(1-\beta_2)D_0}\right] + \mathbb{E}\left[\sqrt{\beta_2(1-\beta_2)D_1}|\partial_i f(\boldsymbol{x}_{t-1})|\right]$$

$$\leq \mathbb{E}\left[\sqrt{\beta_2^3 \boldsymbol{v}_{t-3,i} + (\beta_2 + \beta_2^2)(1-\beta_2)D_0}\right] + \mathbb{E}\left[\sqrt{\beta_2^2(1-\beta_2)D_1}|\partial_i f(\boldsymbol{x}_{t-t})|\right]$$

$$\qquad + \mathbb{E}\left[\sqrt{\beta_2(1-\beta_2)D_1}|\partial_i f(\boldsymbol{x}_{t-1})|\right]$$

$$\leq \ldots$$

$$\leq \mathbb{E}\left[\sqrt{(\beta_2 + \beta_2^2 + \beta_2^3 + \ldots + \beta_2^t)(1-\beta_2)D_0 + \boldsymbol{v}_{0,i}}\right]$$

$$\qquad + \sum_{i=1}^{t-1} \mathbb{E}\left[\sqrt{\beta_2^j(1-\beta_2)D_1}|\partial_i f(\boldsymbol{x}_{t-j})|\right]$$

$$\leq \sqrt{D_0 + \|\boldsymbol{v}_0\|} + \sum_{j=1}^{t-1}\mathbb{E}\left[\sqrt{\beta_2^j(1-\beta_2)D_1}|\partial_i f(\boldsymbol{x}_{t-j})|\right]. \tag{38}$$

As a result, we have that

$$\frac{1}{T}\sum_{t=1}^{T}\mathbb{E}\left[\sqrt{\beta_2 \|\boldsymbol{v}_{t-1}\| + \zeta}\right]$$

$$\leq \sqrt{\zeta} + d\sqrt{D_0 + \|\boldsymbol{v}_0\|} + \frac{1}{T}\sum_{i=1}^{d}\sum_{t=1}^{T}\sum_{j=1}^{t-1}\mathbb{E}\left[\sqrt{\beta_2^j(1-\beta_2)D_1}|\partial_i f(\boldsymbol{x}_{t-j})|\right]$$

$$= c + \frac{1}{T}\sum_{i=1}^{d}\sum_{t=1}^{T-1}\left(\sum_{j=1}^{T-t}\sqrt{\beta_2^j}\right)\mathbb{E}\left[\sqrt{(1-\beta_2)D_1}|\partial_i f(\boldsymbol{x}_t)|\right]$$

$$\leq c + \sum_{i=1}^{d}\frac{\sum_{t=1}^{T}\mathbb{E}[|\partial_i f(\boldsymbol{x}_t)|]}{T}\sqrt{(1-\beta_2)D_1}\left(\sum_{j=1}^{T}\sqrt{\beta_2^j}\right)$$

$$\leq c + \frac{2\sqrt{D_1}}{\sqrt{(1-\beta_2)}}\sum_{i=1}^{d}\frac{\sum_{t=1}^{T}\mathbb{E}[|\partial_i f(\boldsymbol{x}_t)|]}{T}$$

$$\leq c + \frac{2\sqrt{dD_1}}{\sqrt{(1-\beta_2)}}\frac{\sum_{t=1}^{T}\mathbb{E}[\|\nabla f(\boldsymbol{x}_t)\|]}{T}, \tag{39}$$

where the last inequity is due to $\sum_{i=1}^{d} \|\partial_i f(\boldsymbol{x})\| \leq \sqrt{d} \|f(\boldsymbol{x})\|$. We then complete the proof.

## D   Formal Version of Theorem 1 and Its Proof

Recall that $c = \sqrt{\zeta} + d\sqrt{D_0 + \|\boldsymbol{v}_0\|}$, $\Delta = f(\boldsymbol{x}_1) - f(\boldsymbol{x}^*) + \frac{\eta \alpha_0 d D_0}{2\sqrt{\zeta}} + \frac{\eta \alpha_0 D_1 \|\nabla f(\boldsymbol{x}_1)\|^2}{2\sqrt{\zeta}}$. Define $\Lambda_1 = \frac{1}{\max\left(14 L_1 \sqrt{d D_1}, 7 L_1 \sqrt{d D_1}\right)}$, $\Lambda_2 = \min\left(\frac{\zeta}{35 L_0 d D_0}, \frac{\sqrt{\zeta}}{35 L_1^2 d^{1.5} D_0}\right)$ and $\Lambda_3 = \frac{\sqrt[4]{\zeta}}{7 D_1 L_0 d \sqrt{5}}$.

**Theorem 3.** *Let Assumptions 1, 2 and 3 hold. Let* $1 - \beta_2 = \min\left(\frac{1}{7 D_1}, \frac{\sqrt{\zeta} \epsilon^2}{35 d D_0}\right) = \mathcal{O}(\epsilon^2)$, $\eta \leq \min\left(\frac{\sqrt{\zeta}}{7 L_0 D_1}, \Lambda_1 \sqrt{1 - \beta_2}, \frac{1 - \beta_2}{7\sqrt{d}}, \Lambda_2 \epsilon^2, \Lambda_3 \epsilon \sqrt{1 - \beta_2}\right) = \mathcal{O}(\epsilon^2)$, *and* $T \geq \frac{70 \Delta}{\eta \epsilon^2} = \mathcal{O}(\epsilon^{-4})$. *For small* $\epsilon$ *such that* $\epsilon \leq \frac{\sqrt{5 d D_0}}{\sqrt{D_1} \sqrt[4]{\zeta}}$, *we have that*

$$\frac{1}{T} \sum_{t=1}^{T} \mathbb{E}[\|\nabla f(\boldsymbol{x}_t)\|] \leq \left(\frac{2 d \sqrt{35 D_0 D_1}}{\sqrt[4]{\zeta}} + \sqrt{c}\right) \epsilon. \tag{40}$$

*Proof.* According to Lemma 2, we have that

$$\frac{1}{T} \sum_{t=1}^{T} \mathbb{E}\left[\frac{\|\nabla f(\boldsymbol{x}_t)\|^2}{\sqrt{\beta_2 \|\boldsymbol{v}_{t-1}\| + \zeta}}\right] \leq \epsilon^2. \tag{41}$$

According to Lemma 3, we have that

$$\frac{1}{T} \sum_{t=1}^{T} \mathbb{E}[\sqrt{\beta_2 \|\boldsymbol{v}_{t-1}\| + \zeta}]$$
$$\leq c + \frac{2\sqrt{d D_1}}{\sqrt{(1 - \beta_2)}} \frac{\sum_{t=1}^{T} \mathbb{E}[\|\nabla f(\boldsymbol{x}_t)\|]}{T}. \tag{42}$$

Define $e = \frac{1}{T} \sum_{t=1}^{T} \mathbb{E}[\|\nabla f(\boldsymbol{x}_t)\|]$. By Hölder's inequality, we have that

$$\left(\frac{1}{T} \sum_{t=1}^{T} \mathbb{E}[\|\nabla f(\boldsymbol{x}_t)\|]\right)^2 \leq \left(\frac{1}{T} \sum_{t=1}^{T} \mathbb{E}\left[\frac{\|\nabla f(\boldsymbol{x}_t)\|^2}{\sqrt{\beta_2 \|\boldsymbol{v}_{t-1}\| + \zeta}}\right]\right) \left(\frac{1}{T} \sum_{t=1}^{T} \mathbb{E}[\sqrt{\beta_2 \|\boldsymbol{v}_{t-1}\| + \zeta}]\right). \tag{43}$$

By Lemma 3 and Lemma 2, this can be written as

$$e^2 \leq \epsilon^2 \left(c + \frac{2\sqrt{d D_1}}{\sqrt{1 - \beta_2}} e\right). \tag{44}$$

Thus we have that

$$\frac{1}{T} \sum_{t=1}^{T} \mathbb{E}[\|\nabla f(x_t)\|] = e \leq \frac{\sqrt{d D_1} \epsilon^2}{\sqrt{1 - \beta_2}} + \frac{1}{2} \sqrt{\left(\frac{2\sqrt{d D_1} \epsilon^2}{\sqrt{1 - \beta_2}}\right)^2 + 4 c \epsilon^2}.$$

Since $1 - \beta_2 = \min\left(\frac{1}{7 D_1}, \frac{\sqrt{\zeta} \epsilon^2}{35 d D_0}\right) = \mathcal{O}(\epsilon^2)$, if $\epsilon \leq \frac{\sqrt{5 d D_0}}{\sqrt{D_1} \sqrt[4]{\zeta}}$, we have $\frac{\epsilon^2}{\sqrt{1 - \beta_2}} \leq \frac{\sqrt{35 d D_0}}{\sqrt[4]{\zeta}} \epsilon$. It demonstrates that

$$\frac{1}{T} \sum_{t=1}^{T} \mathbb{E}[\|\nabla f(x_t)\|] \leq \left(\frac{2 d \sqrt{35 D_0 D_1}}{\sqrt[4]{\zeta}} + \sqrt{c}\right) \epsilon, \tag{45}$$

which completes the proof.  □

# E Lemmas for Theorem 2

Here are some lemmas we need for the proof of Theorem 2. Lemma 6 and Lemma 7 are from Wang et al. (2023a). We include their proof for completeness.

**Lemma 6.** *(Lemma 6 in Wang et al. (2023a)) For any $\{c_t\}_{t=0}^{\infty} \geq 0$ and $a_t = \beta_2 a_{t-1} + (1 - \beta_2)c_t^2$ and $b_t = \beta_1 b_{t-1} + (1 - \beta_1)c_t$, if $0 < \beta_1^2 < \beta_2 < 1$, we have that*

$$\frac{b_t}{\sqrt{a_t + \zeta}} \leq \frac{1 - \beta_1}{\sqrt{1 - \beta_2}\sqrt{1 - \frac{\beta_1^2}{\beta_2}}}. \tag{46}$$

*Proof.* We can show that

$$\frac{b_t}{\sqrt{a_t + \zeta}} \leq \frac{b_t}{\sqrt{a_t}} \leq \frac{\sum_{i=0}^{t-1}(1 - \beta_1)\beta_1^i c_{t-i}}{\sqrt{\sum_{i=0}^{t-1}(1 - \beta_2)\beta_2^i c_{t-i}^2 + a_0}}$$

$$\leq \frac{1 - \beta_1}{\sqrt{1 - \beta_2}} \frac{\sqrt{\sum_{i=0}^{t-1}\beta_2^i c_{t-i}^2}\sqrt{\sum_{i=0}^{t-1}\frac{\beta_1^{2i}}{\beta_2^i}}}{\sqrt{\sum_{i=0}^{t-1}\beta_2^i c_{t-i}^2}} \leq \frac{1 - \beta_1}{\sqrt{1 - \beta_2}\sqrt{1 - \frac{\beta_1^2}{\beta_2}}}, \tag{47}$$

where the third inequality is due to the Cauchy-Schwarz inequality. $\qquad\square$

**Lemma 7.** *(Lemma 5 in Wang et al. (2023a)) For any $\{c_t\}_{t=0}^{\infty} \geq 0$ and $a_t = \beta_2 a_{t-1} + (1 - \beta_2)c_t^2$ and $b_t = \beta_1 b_{t-1} + (1 - \beta_1)c_t$, if $0 < \beta_1^2 < \beta_2 < 1$ and $a_0 > 0$, we have that*

$$\sum_{t=1}^{T} \frac{b_t^2}{a_t} \leq \frac{(1 - \beta_1)^2}{(1 - \frac{\beta_1}{\sqrt{\beta_2}})^2(1 - \beta_2)}\left(\ln\left(\frac{a_T}{a_0}\right) - T\ln(\beta_2)\right). \tag{48}$$

*Proof.* Due to the monotonicity of the $\frac{1}{a}$ function and Lemma 5.2 in Défossez et al. (2020), we have that

$$\frac{(1 - \beta_2)c_t^2}{a_t} \leq \int_{a=a_t-(1-\beta_2)c_t^2}^{a_t} \frac{1}{a}da = \ln\left(\frac{a_t}{a_t - (1 - \beta_2)c_t^2}\right) = \ln\left(\frac{a_t}{a_{t-1}}\right) - \ln(\beta_2). \tag{49}$$

By telescoping, we have $\sum_{t=1}^{T} \frac{c_t^2}{a_t} \leq \frac{1}{1-\beta_2}\ln\left(\frac{a_T}{a_0}\right) - T\frac{\ln(\beta_2)}{(1-\beta_2)}$. Moreover, for $b_t$ we can show that

$$\frac{b_t}{\sqrt{a_t}} \leq (1 - \beta_1)\sum_{i=1}^{t}\frac{\beta_1^{t-i}c_i}{\sqrt{a_t}} \leq (1 - \beta_1)\sum_{i=1}^{t}\left(\frac{\beta_1}{\sqrt{\beta_2}}\right)^{t-i}\frac{c_i}{\sqrt{a_i}}. \tag{50}$$

Further applying the Cauchy-Schwarz inequality, we get that

$$\frac{b_t^2}{a_t} \leq (1 - \beta_1)^2\left(\sum_{i=1}^{t}\left(\frac{\beta_1}{\sqrt{\beta_2}}\right)^{t-i}\frac{c_i}{\sqrt{a_i}}\right)^2$$

$$\leq (1 - \beta_1)^2\left(\sum_{i=1}^{t}\left(\frac{\beta_1}{\sqrt{\beta_2}}\right)^{t-i}\right)\left(\sum_{i=1}^{t}\left(\frac{\beta_1}{\sqrt{\beta_2}}\right)^{t-i}\frac{c_i^2}{a_i}\right)$$

$$\leq \frac{(1 - \beta_1)^2}{\left(1 - \frac{\beta_1}{\sqrt{\beta_2}}\right)}\left(\sum_{i=1}^{t}\left(\frac{\beta_1}{\sqrt{\beta_2}}\right)^{t-i}\frac{c_i^2}{a_i}\right). \tag{51}$$

This further implies that

$$\sum_{t=1}^{T} \frac{b_t^2}{a_t} \leq \frac{(1 - \beta_1)^2}{(1 - \frac{\beta_1}{\sqrt{\beta_2}})^2(1 - \beta_2)}\left(\ln\left(\frac{a_T}{a_0}\right) - T\ln(\beta_2)\right). \tag{52}$$

$\qquad\square$

**Lemma 8.** *For any $\{c_t\}_{t=0}^{\infty} \geq 0$ and $a_t = \beta_2 a_{t-1} + (1-\beta_2)c_t^2$ and $b_t = \beta_1 b_{t-1} + (1-\beta_1)c_t$, if $0 < \beta_1^4 < \beta_2 < 1$ and $a_0 > 0$, we have that*

$$\sum_{t=1}^{T} \frac{b_t^2}{\sqrt{a_t}} \leq \frac{(1-\beta_1)^2}{(1 - \frac{\beta_1}{\sqrt[4]{\beta_2}})^2} \left( \frac{2}{1-\beta_2}(\sqrt{a_T} - \sqrt{a_0}) + \sum_{t=1}^{T} 2\sqrt{a_{t-1}} \right). \tag{53}$$

*Proof.* Due to the monotonicity of the $\frac{1}{\sqrt{a}}$ function, we can show that

$$\frac{(1-\beta_2)c_t^2}{\sqrt{a_t}} \leq \int_{a=a_t-(1-\beta_2)c_t^2}^{a_t} \frac{1}{\sqrt{a}} da = 2\sqrt{a_t} - 2\sqrt{a_t - (1-\beta_2)c_t^2}$$
$$= 2\sqrt{a_t} - 2\sqrt{\beta_2 a_{t-1}} \leq 2\sqrt{a_t} - 2\sqrt{a_{t-1}} + 2(1-\beta_2)\sqrt{a_{t-1}}. \tag{54}$$

Taking sum up from $t = 1$ to $T$, we obtain the following:

$$\sum_{t=1}^{T} \frac{(1-\beta_2)c_t^2}{\sqrt{a_t}} \leq 2\sqrt{a_T} - 2\sqrt{a_0} + 2(1-\beta_2)\sum_{t=1}^{T} \sqrt{a_{t-1}}. \tag{55}$$

We then derive the following bound:

$$\frac{b_t}{\sqrt[4]{a_t}} \leq (1-\beta_1) \sum_{i=1}^{t} \frac{\beta_1^{t-i}c_i}{\sqrt[4]{a_t}} \leq (1-\beta_1) \sum_{i=1}^{t} \left( \frac{\beta_1}{\sqrt[4]{\beta_2}} \right)^{t-i} \frac{c_i}{\sqrt[4]{a_i}}. \tag{56}$$

It then follows that

$$\frac{b_t^2}{\sqrt{a_t}} \leq (1-\beta_1)^2 \left( \sum_{i=1}^{t} \left( \frac{\beta_1}{\sqrt[4]{\beta_2}} \right)^{t-i} \frac{c_i}{\sqrt[4]{a_i}} \right)^2$$
$$\leq (1-\beta_1)^2 \left( \sum_{i=1}^{t} \left( \frac{\beta_1}{\sqrt[4]{\beta_2}} \right)^{t-i} \right) \left( \sum_{i=1}^{t} \left( \frac{\beta_1}{\sqrt[4]{\beta_2}} \right)^{t-i} \frac{c_i^2}{\sqrt{a_i}} \right)$$
$$\leq \frac{(1-\beta_1)^2}{1 - \frac{\beta_1}{\sqrt[4]{\beta_2}}} \left( \sum_{i=1}^{t} \left( \frac{\beta_1}{\sqrt[4]{\beta_2}} \right)^{t-i} \frac{c_i^2}{\sqrt{a_i}} \right). \tag{57}$$

Taking the sum from $t = 1$ to $T$, we can derive that

$$\sum_{t=1}^{T} \frac{b_t^2}{\sqrt{a_t}} \leq \frac{(1-\beta_1)^2}{(1 - \frac{\beta_1}{\sqrt[4]{\beta_2}})^2} \sum_{t=1}^{T} \frac{c_t^2}{\sqrt{a_t}}. \tag{58}$$

Plug equation 55 in equation 58, and we complete the proof. $\square$

## F Lemma 9 and Its Proof

Define $C_1 = 1 - \frac{\beta_1}{\sqrt{\beta_2}}, C_2 = \sqrt{1 - \frac{\beta_1^2}{\beta_2}}$, we have the following lemma:

**Lemma 9.** *For any $\alpha_0, \alpha_1, \alpha_3, \alpha_4 > 0$, $\beta_2 > 0.5$, and $0 < \beta_1^2 < \beta_2$, we have that*

$$\mathbb{E}[\langle \nabla f(\boldsymbol{u}_t), \boldsymbol{u}_t - \boldsymbol{u}_{t+1} \rangle | \mathcal{F}_t]$$
$$\geq \left( \frac{\eta(1-\beta_1)}{C_1} - \frac{\eta(1-\beta_1)}{C_1 C_2} \left( \frac{1}{2\alpha_0} + \frac{\alpha_0}{2\alpha_1} + \frac{\eta\alpha_0\sqrt{d}D_1 L_1(1-\beta_1)}{\sqrt{1-\beta_2}C_2} \right) \right.$$
$$\left. - \frac{\eta\beta_1(1-\beta_1)\sqrt{\zeta}}{2\alpha_3 C_1 C_2} - \frac{\eta L_1(1-C_1)^2}{2\alpha_4 C_1^2} - \frac{\eta L_1(1-C_1)}{2\alpha_4 C_1^2} \right) \times \sum_{i=1}^{d} \frac{(\partial_i f(\boldsymbol{x}_t))^2}{\sqrt{\beta_2 \boldsymbol{v}_{t-1,i} + \zeta}}$$

$$-\sum_{i=1}^{d}\frac{\eta(1-\beta_1)}{C_1 C_2}\left(\frac{\alpha_0 D_0}{2}\left(\frac{1}{\sqrt{\beta_2 \boldsymbol{v}_{t-1,i}}+\zeta}-\mathbb{E}\left[\frac{1}{\sqrt{\boldsymbol{v}_{t,i}}+\zeta}\bigg|\mathcal{F}_t\right]\right)\right.$$

$$+\frac{\alpha_0 D_1}{2}\mathbb{E}\left[\frac{(\partial_i f(\boldsymbol{x}_{t-1}))^2}{\sqrt{\beta_2 \boldsymbol{v}_{t-1,i}}+\zeta}-\frac{(\partial_i f(\boldsymbol{x}_t))^2}{\sqrt{\boldsymbol{v}_{t,i}}+\zeta}\bigg|\mathcal{F}_t\right]\bigg)$$

$$-\frac{\alpha_0\alpha_1 D_1^2 L_0^2\eta^3 d^2(1-\beta_1)^3}{2(1-\beta_2)C_1 C_2^3\sqrt{\zeta})}-\frac{\alpha_3\eta d\beta_1(1-\beta_1)(1-\beta_2)}{2C_1 C_2}$$

$$-\sum_{i=1}^{d}\frac{\eta^2((1-C_1)^2+0.5(1-C_1))\sqrt{d}L_0}{C_1^2}\frac{\boldsymbol{m}_{t-1,i}^2}{\boldsymbol{v}_{t-1,i}+\zeta}-\sum_{i=1}^{d}\frac{\eta^2 0.5(1-C_1)\sqrt{d}L_0}{C_1^2}\mathbb{E}\left[\frac{\boldsymbol{m}_{t,i}^2}{\boldsymbol{v}_{t,i}+\zeta}\bigg|\mathcal{F}_t\right]$$

$$-\sum_{i=1}^{d}\frac{\alpha_4\eta^3(1-\beta_1)^2(1-C_1)^2 dL_1}{2(1-\beta_2)C_1^2 C_2^2}\frac{\boldsymbol{m}_{t-1,i}^2}{\sqrt{\boldsymbol{v}_{t-1,i}}+\zeta}-\sum_{i=1}^{d}\frac{\alpha_4\eta^3(1-\beta_1)^2(1-C_1)dL_1}{2(1-\beta_2)C_1^2 C_2^2}\mathbb{E}\left[\frac{\boldsymbol{m}_{t,i}^2}{\sqrt{\boldsymbol{v}_{t,i}}+\zeta}\bigg|\mathcal{F}_t\right]. \quad (59)$$

*Proof.* Since $\boldsymbol{u}_t=\frac{\boldsymbol{x}_t-\frac{\beta_1}{\sqrt{\beta_2}}\boldsymbol{x}_{t-1}}{1-\frac{\beta_1}{\sqrt{\beta_2}}}$, we then have that

$$\boldsymbol{u}_{t+1}-\boldsymbol{u}_t=\frac{\boldsymbol{x}_{t+1}-\boldsymbol{x}_t}{1-\frac{\beta_1}{\sqrt{\beta_2}}}-\frac{\beta_1}{\sqrt{\beta_2}}\frac{\boldsymbol{x}_t-\boldsymbol{x}_{t-1}}{1-\frac{\beta_1}{\sqrt{\beta_2}}}$$

$$=\frac{-\eta\odot\frac{\boldsymbol{m}_t}{\sqrt{\boldsymbol{v}_t}+\zeta}}{1-\frac{\beta_1}{\sqrt{\beta_2}}}+\frac{\beta_1}{\sqrt{\beta_2}}\frac{\eta\odot\frac{\boldsymbol{m}_{t-1}}{\sqrt{\boldsymbol{v}_{t-1}}+\zeta}}{1-\frac{\beta_1}{\sqrt{\beta_2}}}$$

$$=\frac{-\eta\odot\frac{\boldsymbol{m}_t}{\sqrt{\beta_2\boldsymbol{v}_{t-1}}+\zeta}}{1-\frac{\beta_1}{\sqrt{\beta_2}}}+\frac{-\eta\odot\frac{\boldsymbol{m}_t}{\sqrt{\boldsymbol{v}_t}+\zeta}+\eta\odot\frac{\boldsymbol{m}_t}{\sqrt{\beta_2\boldsymbol{v}_{t-1}}+\zeta}}{1-\frac{\beta_1}{\sqrt{\beta_2}}}$$

$$+\frac{\eta\odot\frac{\beta_1\boldsymbol{m}_{t-1}}{\sqrt{\beta_2\boldsymbol{v}_{t-1}}+\zeta}}{1-\frac{\beta_1}{\sqrt{\beta_2}}}+\frac{\eta\odot\frac{\beta_1\boldsymbol{m}_{t-1}}{\sqrt{\beta_2\boldsymbol{v}_{t-1}+\beta_2\zeta}}-\eta\odot\frac{\beta_1\boldsymbol{m}_{t-1}}{\sqrt{\beta_2\boldsymbol{v}_{t-1}}+\zeta}}{1-\frac{\beta_1}{\sqrt{\beta_2}}}$$

$$=\frac{-\eta\odot\frac{(1-\beta_1)\boldsymbol{g}_t}{\sqrt{\beta_2\boldsymbol{v}_{t-1}}+\zeta}}{1-\frac{\beta_1}{\sqrt{\beta_2}}}+\frac{-\eta\odot\frac{\boldsymbol{m}_t}{\sqrt{\boldsymbol{v}_t}+\zeta}+\eta\odot\frac{\boldsymbol{m}_t}{\sqrt{\beta_2\boldsymbol{v}_{t-1}}+\zeta}}{1-\frac{\beta_1}{\sqrt{\beta_2}}}+\frac{\eta\odot\frac{\beta_1\boldsymbol{m}_{t-1}}{\sqrt{\beta_2\boldsymbol{v}_{t-1}+\beta_2\zeta}}-\eta\odot\frac{\beta_1\boldsymbol{m}_{t-1}}{\sqrt{\beta_2\boldsymbol{v}_{t-1}}+\zeta}}{1-\frac{\beta_1}{\sqrt{\beta_2}}}, \quad (60)$$

where the last equality is due to $\boldsymbol{m}_t=\beta_1\boldsymbol{m}_{t-1}+(1-\beta_1)\boldsymbol{g}_t$. For $(L_0, L_1)$-smooth objectives, from Lemma 1 in Crawshaw et al. (2022) we can get that

$$\underbrace{\mathbb{E}[\langle\nabla f(\boldsymbol{u}_t),\boldsymbol{u}_t-\boldsymbol{u}_{t+1}\rangle|\mathcal{F}_t]}_{\text{first-order}}\leq f(\boldsymbol{u}_t)-\mathbb{E}[f(\boldsymbol{u}_{t+1})|\mathcal{F}_t]+\underbrace{\sum_{i=1}^{d}\frac{L_0}{2\sqrt{d}}\mathbb{E}[\|\boldsymbol{u}_{t+1}-\boldsymbol{u}_t\|\|\boldsymbol{u}_{t+1,i}-\boldsymbol{u}_{t,i}\||\mathcal{F}_t]}_{\text{second-order}}$$

$$+\underbrace{\sum_{i=1}^{d}\frac{L_1|\partial_i f(\boldsymbol{u}_t)|}{2}\mathbb{E}[\|\boldsymbol{u}_{t+1}-\boldsymbol{u}_t\|\|\boldsymbol{u}_{t+1,i}-\boldsymbol{u}_{t,i}\||\mathcal{F}_t]}_{\text{additional term}}. \quad (61)$$

We then focus on the first-order term. Based on equation 60, we divide our first-order term into four parts:

$$\mathbb{E}[\langle\nabla f(\boldsymbol{u}_t),\boldsymbol{u}_t-\boldsymbol{u}_{t+1}\rangle|\mathcal{F}_t]=\mathbb{E}[\langle\nabla f(\boldsymbol{x}_t),\boldsymbol{u}_t-\boldsymbol{u}_{t+1}\rangle|\mathcal{F}_t]+\mathbb{E}\left[\langle\nabla f(\boldsymbol{u}_t)-\nabla f(\boldsymbol{x}_t),\boldsymbol{u}_t-\boldsymbol{u}_{t+1}\rangle|\mathcal{F}_t\right]$$

$$=\underbrace{\mathbb{E}\left[\left\langle\nabla f(\boldsymbol{x}_t),\frac{\eta\odot\frac{(1-\beta_1)\boldsymbol{g}_t}{\sqrt{\beta_2\boldsymbol{v}_{t-1}}+\zeta}}{1-\frac{\beta_1}{\sqrt{\beta_2}}}\right\rangle\bigg|\mathcal{F}_t\right]}_{\text{first-order main}}$$

$$- \underbrace{\mathbb{E}\left[\left\langle \nabla f(\boldsymbol{x}_t), \frac{-\eta \odot \frac{\boldsymbol{m}_t}{\sqrt{\boldsymbol{v}_t+\zeta}} + \eta \odot \frac{\boldsymbol{m}_t}{\sqrt{\beta_2 \boldsymbol{v}_{t-1}+\zeta}}}{1 - \frac{\beta_1}{\sqrt{\beta_2}}} \right\rangle \middle| \mathcal{F}_t\right]}_{\text{error 1}}$$

$$- \underbrace{\left\langle \nabla f(\boldsymbol{x}_t), \frac{\eta \odot \frac{\beta_1 \boldsymbol{m}_{t-1}}{\sqrt{\beta_2 \boldsymbol{v}_{t-1}+\beta_2\zeta}} - \eta \odot \frac{\beta_1 \boldsymbol{m}_{t-1}}{\sqrt{\beta_2 \boldsymbol{v}_{t-1}+\zeta}}}{1 - \frac{\beta_1}{\sqrt{\beta_2}}} \right\rangle}_{\text{error 2}}$$

$$- \underbrace{\mathbb{E}[\langle \nabla f(\boldsymbol{u}_t) - \nabla f(\boldsymbol{x}_t), \boldsymbol{u}_{t+1} - \boldsymbol{u}_t\rangle | \mathcal{F}_t]}_{\text{error 3}}. \tag{62}$$

The first-order main term is easy to bound. Since given $\mathcal{F}_t$, $\boldsymbol{g}_t$ is independent of $\boldsymbol{v}_{t-1}$ and we have that

$$\underbrace{\mathbb{E}\left[\left\langle \nabla f(\boldsymbol{x}_t), \frac{\eta \odot \frac{(1-\beta_1)\boldsymbol{g}_t}{\sqrt{\beta_2 \boldsymbol{v}_{t-1}+\zeta}}}{1 - \frac{\beta_1}{\sqrt{\beta_2}}} \right\rangle \middle| \mathcal{F}_t\right]}_{\text{first-order main}} = \sum_{i=1}^{d} \frac{\eta(1-\beta_1)}{1 - \frac{\beta_1}{\sqrt{\beta_2}}} \frac{(\partial_i f(\boldsymbol{x}_t))^2}{\sqrt{\beta_2 \boldsymbol{v}_{t-1,i} + \zeta}}. \tag{63}$$

We then focus on the error 1 term. Since $\boldsymbol{v}_t = \beta_2 \boldsymbol{v}_{t-1} + (1-\beta_2)\boldsymbol{g}_t \odot \boldsymbol{g}_t$, we have equation 15 and it follows that

$$\underbrace{\mathbb{E}\left[\left\langle \nabla f(\boldsymbol{x}_t), \frac{-\eta \odot \frac{\boldsymbol{m}_t}{\sqrt{\boldsymbol{v}_t+\zeta}} + \eta \odot \frac{\boldsymbol{m}_t}{\sqrt{\beta_2 \boldsymbol{v}_{t-1}+\zeta}}}{1 - \frac{\beta_1}{\sqrt{\beta_2}}} \right\rangle \middle| \mathcal{F}_t\right]}_{\text{error 1}}$$

$$\leq \sum_{i=1}^{d} \frac{\eta}{1 - \frac{\beta_1}{\sqrt{\beta_2}}} \mathbb{E}\left[|\partial_i f(\boldsymbol{x}_t)| \frac{(1-\beta_2)\boldsymbol{g}_{t,i}^2|\boldsymbol{m}_{t,i}|}{(\sqrt{\boldsymbol{v}_{t,i}+\zeta})(\sqrt{\beta_2 \boldsymbol{v}_{t-1,i}+\zeta})(\sqrt{\boldsymbol{v}_{t,i}+\zeta} + \sqrt{\beta_2 \boldsymbol{v}_{t-1,i}+\zeta})} \middle| \mathcal{F}_t\right]. \tag{64}$$

According to Lemma 6, we have $\frac{|\boldsymbol{m}_{t,i}|}{\sqrt{\boldsymbol{v}_{t,i}+\zeta}} \leq \frac{1-\beta_1}{\sqrt{1-\beta_2}\sqrt{1-\frac{\beta_1^2}{\beta_2}}}$, which demonstrates that

$$\text{error 1} \leq \sum_{i=1}^{d} \frac{\eta}{1 - \frac{\beta_1}{\sqrt{\beta_2}}} \frac{1-\beta_1}{\sqrt{1-\frac{\beta_1^2}{\beta_2}}} \frac{|\partial_i f(\boldsymbol{x}_t)|}{\sqrt{\beta_2 \boldsymbol{v}_{t-1,i}+\zeta}} \mathbb{E}\left[\left\|\frac{\sqrt{1-\beta_2}\boldsymbol{g}_{t,i}^2}{(\sqrt{\boldsymbol{v}_{t,i}+\zeta} + \sqrt{\beta_2 \boldsymbol{v}_{t-1,i}+\zeta})}\right\| \middle| \mathcal{F}_t\right]. \tag{65}$$

Similar to the proof in Appendix A, we can show that for any $\alpha_0 > 0$ and $i \in [d]$,

$$\frac{|\partial_i f(\boldsymbol{x}_t)|}{\sqrt{\beta_2 \boldsymbol{v}_{t-1,i}+\zeta}} \mathbb{E}\left[\left\|\frac{\sqrt{1-\beta_2}\boldsymbol{g}_{t,i}^2}{(\sqrt{\boldsymbol{v}_{t,i}+\zeta} + \sqrt{\beta_2 \boldsymbol{v}_{t-1,i}+\zeta})}\right\| \middle| \mathcal{F}_t\right]$$

$$\leq \frac{(\partial_i f(\boldsymbol{x}_t))^2}{2\alpha_0\sqrt{\beta_2 \boldsymbol{v}_{t-1,i}+\zeta}} + \frac{\alpha_0}{2\sqrt{\beta_2 \boldsymbol{v}_{t-1,i}+\zeta}} \left(\mathbb{E}\left[\frac{\sqrt{1-\beta_2}\boldsymbol{g}_{t,i}^2}{(\sqrt{\boldsymbol{v}_{t,i}+\zeta} + \sqrt{\beta_2 \boldsymbol{v}_{t-1,i}+\zeta})} \middle| \mathcal{F}_t\right]\right)^2. \tag{66}$$

For the last term, using Hölder's inequality, we have that

$$\frac{\alpha_0}{2\sqrt{\beta_2 \boldsymbol{v}_{t-1,i}+\zeta}} \left(\mathbb{E}\left[\frac{\sqrt{1-\beta_2}\boldsymbol{g}_{t,i}^2}{(\sqrt{\boldsymbol{v}_{t,i}+\zeta} + \sqrt{\beta_2 \boldsymbol{v}_{t-1,i}+\zeta})} \middle| \mathcal{F}_t\right]\right)^2$$

$$\leq \frac{(1-\beta_2)\alpha_0}{2\sqrt{\beta_2 \boldsymbol{v}_{t-1,i}+\zeta}} \mathbb{E}[\boldsymbol{g}_{t,i}^2|\mathcal{F}_t]\mathbb{E}\left[\frac{\boldsymbol{g}_{t,i}^2}{(\sqrt{\boldsymbol{v}_{t,i}+\zeta} + \sqrt{\beta_2 \boldsymbol{v}_{t-1,i}+\zeta})^2} \middle| \mathcal{F}_t\right]$$

$$\leq \frac{\alpha_0}{2}(D_0 + D_1(\partial_i f(\boldsymbol{x}_t))^2) \times \mathbb{E}\left[\frac{(1-\beta_2)\boldsymbol{g}_{t,i}^2}{\sqrt{\boldsymbol{v}_{t,i}+\zeta}\sqrt{\beta_2 \boldsymbol{v}_{t-1,i}+\zeta}(\sqrt{\boldsymbol{v}_{t,i}+\zeta} + \sqrt{\beta_2 \boldsymbol{v}_{t-1,i}+\zeta})} \middle| \mathcal{F}_t\right]$$

$$\le \frac{\alpha_0}{2}(D_0 + D_1(\partial_i f(\boldsymbol{x}_t))^2)\mathbb{E}\left[\frac{1}{\sqrt{\beta_2 \boldsymbol{v}_{t-1,i} + \zeta}} - \frac{1}{\sqrt{\boldsymbol{v}_{t,i} + \zeta}}\bigg| \mathcal{F}_t\right].$$

(67)

Similar to equation 20, we then show the sum of equation 67 from $t = 1$ to $T$ can be bounded. For $t > 1$ we have that

$$\frac{\alpha_0}{2}(D_0 + D_1(\partial_i f(\boldsymbol{x}_t))^2)\mathbb{E}\left[\frac{1}{\sqrt{\beta_2 \boldsymbol{v}_{t-1,i} + \zeta}} - \frac{1}{\sqrt{\boldsymbol{v}_{t,i} + \zeta}}\bigg| \mathcal{F}_t\right]$$

$$= \frac{\alpha_0 D_0}{2}\mathbb{E}\left[\frac{1}{\sqrt{\beta_2 \boldsymbol{v}_{t-1,i} + \zeta}} - \frac{1}{\sqrt{\boldsymbol{v}_{t,i} + \zeta}}\bigg| \mathcal{F}_t\right] + \frac{\alpha_0 D_1}{2}\mathbb{E}\left[\frac{(\partial_i f(\boldsymbol{x}_{t-1}))^2}{\sqrt{\beta_2 \boldsymbol{v}_{t-1,i} + \zeta}} - \frac{(\partial_i f(\boldsymbol{x}_t))^2}{\sqrt{\boldsymbol{v}_{t,i} + \zeta}}\bigg| \mathcal{F}_t\right]$$

$$+ \frac{\alpha_0 D_1}{2}\mathbb{E}\left[\frac{(\partial_i f(\boldsymbol{x}_t))^2 - (\partial_i f(\boldsymbol{x}_{t-1}))^2}{\sqrt{\beta_2 \boldsymbol{v}_{t-1,i} + \zeta}}\bigg| \mathcal{F}_t\right].$$

(68)

Note for the last term on RHS of equation 68, due to the different update of $\boldsymbol{x}_t$, there is a little difference from equation 21. For the last term and any $\alpha_1 > 0$ and $i \in [d]$, we have that

$$\frac{\alpha_0 D_1}{2}\mathbb{E}\left[\frac{(\partial_i f(\boldsymbol{x}_t))^2 - (\partial_i f(\boldsymbol{x}_{t-1}))^2}{\sqrt{\beta \boldsymbol{v}_{t-1,i} + \zeta}}\bigg| \mathcal{F}_t\right]$$

$$\le \frac{\alpha_0 D_1}{2}\mathbb{E}\left[\frac{2|\partial_i f(\boldsymbol{x}_t)||\partial_i f(\boldsymbol{x}_t) - \partial_i f(\boldsymbol{x}_{t-1})|}{\sqrt{\beta \boldsymbol{v}_{t-1,i} + \zeta}}\bigg| \mathcal{F}_t\right]$$

$$\le \alpha_0 D_1 \frac{|\partial_i f(\boldsymbol{x}_t)|(L_0 + L_1|\partial_i f(\boldsymbol{x}_t)|)\eta\left\|\frac{1}{\sqrt{\boldsymbol{v}_{t-1}+\zeta}} \odot \boldsymbol{m}_{t-1}\right\|}{\sqrt{\beta \boldsymbol{v}_{t-1,i} + \zeta}}$$

$$\le \frac{\alpha_0 D_1}{2\sqrt{\beta \boldsymbol{v}_{t-1,i} + \zeta}}\left(\frac{(\partial_i f(\boldsymbol{x}_t))^2}{\alpha_1 D_1} + \alpha_1 D_1 L_0^2 \eta^2 \left\|\frac{1}{\boldsymbol{v}_{t-1} + \zeta} \odot \boldsymbol{m}_{t-1}\right\|^2\right.$$

$$\left. + 2\eta L_1(\partial_i f(\boldsymbol{x}_t))^2\left\|\frac{1}{\boldsymbol{v}_{t-1} + \zeta} \odot \boldsymbol{m}_{t-1}\right\|\right)$$

$$\le \frac{\alpha_0 D_1}{2\sqrt{\beta \boldsymbol{v}_{t-1,i} + \zeta}}\left(\frac{(\partial_i f(\boldsymbol{x}_t))^2}{\alpha_1 D_1} + \alpha_1 D_1 L_0^2 \eta^2 \frac{d(1-\beta_1)^2}{(1-\beta_2)(1-\frac{\beta_1^2}{\beta_2})} + 2\eta L_1(\partial_i f(\boldsymbol{x}_t))^2\frac{\sqrt{d}(1-\beta_1)}{\sqrt{1-\beta_2}\sqrt{1-\frac{\beta_1^2}{\beta_2}}}\right), \quad (69)$$

where the first inequality is due to the fact that for any $a, b$, we have $a^2 - b^2 \le 2|a||a - b|$, the second inequality is by the $(L_0, L_1)$-smoothness assumption and update process of $x_t$, and the last inequality is due to Lemma 6. As a result, for $\alpha_0, \alpha_1 > 0$, if $t = 1$, it can be shown that

$$\text{error } 1 \le \sum_{i=1}^{d}\frac{\eta}{1 - \frac{\beta_1}{\sqrt{\beta_2}}}\frac{1 - \beta_1}{\sqrt{1 - \frac{\beta_1^2}{\beta_2}}}\left(\frac{(\partial_i f(\boldsymbol{x}_t))^2}{2\alpha_0\sqrt{\beta_2 \boldsymbol{v}_{t-1,i} + \zeta}} + \frac{\alpha_0 D_0}{2}\mathbb{E}\left[\frac{1}{\sqrt{\beta_2 \boldsymbol{v}_{t-1,i} + \zeta}} - \frac{1}{\sqrt{\boldsymbol{v}_{t,i} + \zeta}}\bigg| \mathcal{F}_t\right]\right.$$

$$\left. + \frac{\alpha_0 D_1}{2}\mathbb{E}\left[\frac{(\partial_i f(\boldsymbol{x}_t))^2}{\sqrt{\beta_2 \boldsymbol{v}_{t-1,i} + \zeta}} - \frac{(\partial_i f(\boldsymbol{x}_t))^2}{\sqrt{\boldsymbol{v}_{t,i} + \zeta}}\bigg| \mathcal{F}_t\right]\right),$$

(70)

and if $t > 1$ we have that

$$\text{error } 1 \le \sum_{i=1}^{d}\frac{\eta}{1 - \frac{\beta_1}{\sqrt{\beta_2}}}\frac{1 - \beta_1}{\sqrt{1 - \frac{\beta_1^2}{\beta_2}}}\left(\frac{(\partial_i f(\boldsymbol{x}_t))^2}{2\alpha_0\sqrt{\beta_2 \boldsymbol{v}_{t-1,i} + \zeta}} + \frac{\alpha_0 D_0}{2}\mathbb{E}\left[\frac{1}{\sqrt{\beta_2 \boldsymbol{v}_{t-1,i} + \zeta}} - \frac{1}{\sqrt{\boldsymbol{v}_{t,i} + \zeta}}\bigg| \mathcal{F}_t\right]\right.$$

$$\left. + \frac{\alpha_0 D_1}{2}\mathbb{E}\left[\frac{(\partial_i f(\boldsymbol{x}_{t-1}))^2}{\sqrt{\beta_2 \boldsymbol{v}_{t-1,i} + \zeta}} - \frac{(\partial_i f(\boldsymbol{x}_t))^2}{\sqrt{\boldsymbol{v}_{t,i} + \zeta}}\bigg| \mathcal{F}_t\right]\right]$$

$$+ \frac{\alpha_0 D_1}{2\sqrt{\beta \boldsymbol{v}_{t-1,i} + \zeta}} \left( \frac{(\partial_i f(\boldsymbol{x}_t))^2}{\alpha_1 D_1} + \alpha_1 D_1 L_0^2 \eta^2 \frac{d(1-\beta_1)^2}{(1-\beta_2)(1-\frac{\beta_1^2}{\beta_2})} \right.$$

$$\left. + 2\eta L_1 (\partial_i f(\boldsymbol{x}_t))^2 \frac{\sqrt{d}(1-\beta_1)}{\sqrt{1-\beta_2}\sqrt{1-\frac{\beta_1^2}{\beta_2}}} \right) \right). \tag{71}$$

Since we define $\boldsymbol{m}_0 = 0$, and $\boldsymbol{x}_0 = \boldsymbol{x}_1$, therefore this inequality holds for $t = 1$. Note many terms in the RHS of equation 71 can be reduced by telescoping, which will be demonstrated later. Now we move to the error 2 term. For any $\alpha_3 > 0, 0 < \beta_1^2 < \beta_2$ and $\beta_2 > 0.5$, we have that

$$\underbrace{\left\langle \nabla f(\boldsymbol{x}_t), \frac{\eta \odot \frac{\beta_1 \boldsymbol{m}_{t-1}}{\sqrt{\beta_2 \boldsymbol{v}_{t-1} + \beta_2 \zeta}} - \eta \odot \frac{\beta_1 \boldsymbol{m}_{t-1}}{\sqrt{\beta_2 \boldsymbol{v}_{t-1} + \zeta}}}{1 - \frac{\beta_1}{\sqrt{\beta_2}}} \right\rangle}_{\text{error 2}}$$

$$= \frac{\eta \beta_1}{1 - \frac{\beta_1}{\sqrt{\beta_2}}} \left\langle \nabla f(\boldsymbol{x}_t), \frac{(1-\beta_2)\zeta}{(\sqrt{\beta_2 \boldsymbol{v}_{t-1}} + \zeta) \odot (\sqrt{\beta_2 \boldsymbol{v}_{t-1}} + \beta_2 \zeta) \odot (\sqrt{\beta_2 \boldsymbol{v}_{t-1}} + \zeta + \sqrt{\beta_2 \boldsymbol{v}_{t-1}} + \beta_2 \zeta)} \odot \boldsymbol{m}_{t-1} \right\rangle$$

$$\leq \sum_{i=1}^d \frac{\eta \beta_1}{1 - \frac{\beta_1}{\sqrt{\beta_2}}} |\partial_i f(\boldsymbol{x}_t)| \frac{(1-\beta_2)\zeta |\boldsymbol{m}_{t-1,i}|}{(\sqrt{\beta_2 \boldsymbol{v}_{t-1,i}} + \zeta)(\sqrt{\beta_2 \boldsymbol{v}_{t-1,i}} + \beta_2 \zeta)(\sqrt{\beta_2 \boldsymbol{v}_{t-1,i}} + \zeta + \sqrt{\beta_2 \boldsymbol{v}_{t-1,i}} + \beta_2 \zeta)}$$

$$\leq \sum_{i=1}^d \frac{\eta \beta_1}{1 - \frac{\beta_1}{\sqrt{\beta_2}}} |\partial_i f(\boldsymbol{x}_t)| \frac{(1-\beta_2)\zeta}{(\sqrt{\beta_2 \boldsymbol{v}_{t-1,i}} + \zeta)\sqrt{\zeta}} \frac{1-\beta_1}{\sqrt{1-\beta_2}\sqrt{1-\frac{\beta_1^2}{\beta_2}}}$$

$$\leq \sum_{i=1}^d \frac{\eta \beta_1 (1-\beta_1)\sqrt{\zeta}}{(1 - \frac{\beta_1}{\sqrt{\beta_2}})\sqrt{1-\frac{\beta_1^2}{\beta_2}}} \left( \frac{(\partial_i f(\boldsymbol{x}_t))^2}{2\alpha_3 \sqrt{\beta_2 \boldsymbol{v}_{t-1,i}} + \zeta} + \frac{\alpha_3 (1-\beta_2)}{2\sqrt{\beta_2 \boldsymbol{v}_{t-1,i}} + \zeta} \right), \tag{72}$$

where the second inequality is due to Lemma 6 and $2\beta_2 > 1$.

For error 3, it can be bounded as follows

$$\langle \nabla f(\boldsymbol{u}_t) - \nabla f(\boldsymbol{x}_t), \boldsymbol{u}_t - \boldsymbol{u}_{t+1} \rangle$$

$$\leq \sum_{i=1}^d |\partial_i f(\boldsymbol{u}_t) - \partial_i f(\boldsymbol{x}_t)||\boldsymbol{u}_{t,i} - \boldsymbol{u}_{t+1,i}|$$

$$\leq \sum_{i=1}^d (L_0 + L_1|\partial_i f(\boldsymbol{x}_t)|)\|\boldsymbol{u}_t - \boldsymbol{x}_t\||\boldsymbol{u}_{t,i} - \boldsymbol{u}_{t+1,i}|$$

$$\leq \sum_{i=1}^d (L_0 + L_1|\partial_i f(\boldsymbol{x}_t)|) \frac{\frac{\beta_1}{\sqrt{\beta_2}}}{1 - \frac{\beta_1}{\sqrt{\beta_2}}} \|\boldsymbol{x}_t - \boldsymbol{x}_{t-1}\| \left| \frac{\boldsymbol{x}_{t+1,i} - \boldsymbol{x}_{t,i}}{1 - \frac{\beta_1}{\sqrt{\beta_2}}} - \frac{\beta_1}{\sqrt{\beta_2}} \frac{\boldsymbol{x}_{t,i} - \boldsymbol{x}_{t-1,i}}{1 - \frac{\beta_1}{\sqrt{\beta_2}}} \right|, \tag{73}$$

where the second inequality is due to Assumption 3. According to the update process of $\boldsymbol{x}_t$ and Lemma 7, it is easy to get that

$$\sum_{i=1}^d L_0 \frac{\frac{\beta_1}{\sqrt{\beta_2}}}{1 - \frac{\beta_1}{\sqrt{\beta_2}}} \|\boldsymbol{x}_t - \boldsymbol{x}_{t-1}\| \left| \frac{\boldsymbol{x}_{t+1,i} - \boldsymbol{x}_{t,i}}{1 - \frac{\beta_1}{\sqrt{\beta_2}}} - \frac{\beta_1}{\sqrt{\beta_2}} \frac{\boldsymbol{x}_{t,i} - \boldsymbol{x}_{t-1,i}}{1 - \frac{\beta_1}{\sqrt{\beta_2}}} \right|$$

$$\leq \sum_{i=1}^d L_0 \frac{\frac{\beta_1}{\sqrt{\beta_2}}}{1 - \frac{\beta_1}{\sqrt{\beta_2}}} \left( \frac{1}{1 - \frac{\beta_1}{\sqrt{\beta_2}}} (\|\boldsymbol{x}_t - \boldsymbol{x}_{t-1}\||\boldsymbol{x}_{t+1,i} - \boldsymbol{x}_{t,i}|) + \frac{\frac{\beta_1}{\sqrt{\beta_2}}}{1 - \frac{\beta_1}{\sqrt{\beta_2}}} (\|\boldsymbol{x}_t - \boldsymbol{x}_{t-1}\||\boldsymbol{x}_{t,i} - \boldsymbol{x}_{t-1,i}|) \right)$$

$$\leq \sum_{i=1}^d L_0 \frac{\frac{\beta_1}{\sqrt{\beta_2}}}{1 - \frac{\beta_1}{\sqrt{\beta_2}}} \left( \frac{1}{1 - \frac{\beta_1}{\sqrt{\beta_2}}} \left( \frac{\|\boldsymbol{x}_t - \boldsymbol{x}_{t-1}\|^2}{2\sqrt{d}} + \frac{\sqrt{d}|\boldsymbol{x}_{t+1,i} - \boldsymbol{x}_{t,i}|^2}{2} \right) \right)$$

$$+\frac{\frac{\beta_1}{\sqrt{\beta_2}}}{1-\frac{\beta_1}{\sqrt{\beta_2}}}\left(\frac{\|\boldsymbol{x}_t-\boldsymbol{x}_{t-1}\|^2}{2\sqrt{d}}+\frac{\sqrt{d}|\boldsymbol{x}_{t,i}-\boldsymbol{x}_{t-1,i}|^2}{2}\right)\right)$$

$$\leq L_0\eta^2\sum_{i=1}^{d}\left(\frac{\frac{\beta_1}{2\sqrt{\beta_2}}\sqrt{d}+\frac{\beta_1^2}{\beta_2}\sqrt{d}}{\left(1-\frac{\beta_1}{\sqrt{\beta_2}}\right)^2}\frac{|\boldsymbol{m}_{t-1,i}|^2}{\boldsymbol{v}_{t-1,i}+\zeta}+\frac{\frac{\beta_1}{2\sqrt{\beta_2}}\sqrt{d}}{\left(1-\frac{\beta_1}{\sqrt{\beta_2}}\right)^2}\frac{\boldsymbol{m}_{t,i}^2}{\boldsymbol{v}_{t,i}+\zeta}\right).\tag{74}$$

However, it is hard to bound the remaining $|\partial_i f(\boldsymbol{x}_t)|\|\boldsymbol{x}_t-\boldsymbol{x}_{t-1}\||\boldsymbol{u}_{t,i}-\boldsymbol{u}_{t+1,i}|$ term by directly applying Lemma 6, which will induce a $\mathcal{O}(|\partial_i f(\boldsymbol{x}_t)|\frac{\eta^2}{1-\beta_2})$ term and with our affine noise variance, we can only bound the $\frac{(\partial_i f(\boldsymbol{x}_t))^2}{\sqrt{\beta_2\boldsymbol{v}_{t-1,i}+\zeta}}$ term. It is worth noting that this challenging additional term is due to the $(L_0,L_1)$-smoothness, and methods in Wang et al. (2023a); Li et al. (2023) do not generalize to our case. To solve this challenge, we first bound the additional terms as follows:

$$\sum_{i=1}^{d}|\partial_i f(\boldsymbol{x}_t)|\|\boldsymbol{x}_t-\boldsymbol{x}_{t-1}\||\boldsymbol{x}_{t,i}-\boldsymbol{x}_{t-1,i}|$$

$$\leq\sum_{i=1}^{d}|\partial_i f(\boldsymbol{x}_t)|\|\boldsymbol{x}_t-\boldsymbol{x}_{t-1}\|\frac{\eta|\boldsymbol{m}_{t-1,i}|}{\sqrt{\boldsymbol{v}_{t-1,i}+\zeta}}$$

$$\leq\sum_{i=1}^{d}\frac{\eta(\partial_i f(\boldsymbol{x}_t))^2}{2\alpha_4\sqrt{\boldsymbol{v}_{t-1,i}+\zeta}}+\frac{\alpha_4\eta\boldsymbol{m}_{t-1,i}^2}{2\sqrt{\boldsymbol{v}_{t-1,i}+\zeta}}|\boldsymbol{x}_t-\boldsymbol{x}_{t-1}|^2$$

$$\leq\sum_{i=1}^{d}\frac{\eta(\partial_i f(\boldsymbol{x}_t))^2}{2\alpha_4\sqrt{\beta_2\boldsymbol{v}_{t-1,i}+\zeta}}+\sum_{i=1}^{d}\frac{\alpha_4\eta^3\boldsymbol{m}_{t-1,i}^2}{2\sqrt{\boldsymbol{v}_{t-1,i}+\zeta}}\frac{d(1-\beta_1)^2}{(1-\beta_2)(1-\frac{\beta_1^2}{\beta_2})},\tag{75}$$

for any $\alpha_4>0$, where the last inequality is due to Lemma 6. The motivation is to bound using $\frac{(\partial_i f(\boldsymbol{x}_t))^2}{\sqrt{\beta_2\boldsymbol{v}_{t-1,i}+\zeta}}$ and $\frac{\eta^3\boldsymbol{m}_{t-1,i}^2}{2\sqrt{\boldsymbol{v}_{t-1,i}+\zeta}}$, where the latter can be bounded by Lemma 8. Similarly, we have that

$$\sum_{i=1}^{d}|\partial_i f(\boldsymbol{x}_t)|\|\boldsymbol{x}_t-\boldsymbol{x}_{t-1}\||\boldsymbol{x}_{t+1,i}-\boldsymbol{x}_{t,i}|$$

$$\leq\sum_{i=1}^{d}\left[\frac{\eta(\partial_i f(\boldsymbol{x}_t))^2}{2\alpha_4\sqrt{\boldsymbol{v}_{t,i}+\zeta}}+\frac{\alpha_4\eta\boldsymbol{m}_{t,i}^2}{2\sqrt{\boldsymbol{v}_{t,i}+\zeta}}|\boldsymbol{x}_t-\boldsymbol{x}_{t-1}|^2\right]$$

$$\leq\sum_{i=1}^{d}\frac{\eta(\partial_i f(\boldsymbol{x}_t))^2}{2\alpha_4\sqrt{\beta_2\boldsymbol{v}_{t-1,i}+\zeta}}+\sum_{i=1}^{d}\frac{\alpha_4\eta^3\boldsymbol{m}_{t,i}^2}{2\sqrt{\boldsymbol{v}_{t,i}+\zeta}}\frac{d(1-\beta_1)^2}{(1-\beta_2)(1-\frac{\beta_1^2}{\beta_2})}.\tag{76}$$

Combine equation 62, equation 63, equation 71, equation 72, equation 73, equation 74, equation 75 and equation 76, and we then have that

$$\mathbb{E}[\langle\nabla f(\boldsymbol{u}_t),\boldsymbol{u}_{t+1}-\boldsymbol{u}_t\rangle|\mathcal{F}_t]$$

$$\leq-\sum_{i=1}^{d}\frac{\eta(1-\beta_1)}{1-\frac{\beta_1}{\sqrt{\beta_2}}}\frac{(\partial_i f(\boldsymbol{x}_t))^2}{\sqrt{\beta_2\boldsymbol{v}_{t-1,i}+\zeta}}$$

$$+\sum_{i=1}^{d}\frac{\eta}{1-\frac{\beta_1}{\sqrt{\beta_2}}}\frac{1-\beta_1}{\sqrt{1-\frac{\beta_1^2}{\beta_2}}}\left(\frac{(\partial_i f(\boldsymbol{x}_t))^2}{2\alpha_0\sqrt{\beta_2\boldsymbol{v}_{t-1,i}+\zeta}}+\frac{\alpha_0 D_0}{2}\mathbb{E}\left[\frac{1}{\sqrt{\beta_2\boldsymbol{v}_{t-1,i}+\zeta}}-\frac{1}{\sqrt{\boldsymbol{v}_{t,i}+\zeta}}\bigg|\mathcal{F}_t\right]\right)$$

$$+\frac{\alpha_0 D_1}{2}\mathbb{E}\left[\frac{(\partial_i f(\boldsymbol{x}_{t-1}))^2}{\sqrt{\beta_2\boldsymbol{v}_{t-1,i}+\zeta}}-\frac{(\partial_i f(\boldsymbol{x}_t))^2}{\sqrt{\boldsymbol{v}_{t,i}+\zeta}}\bigg|\mathcal{F}_t\right]$$

$$+ \frac{\alpha_0 D_1}{2\sqrt{\beta \boldsymbol{v}_{t-1,i} + \zeta}} \left( \frac{(\partial_i f(\boldsymbol{x}_t))^2}{\alpha_1 D_1} + \alpha_1 D_1 L_0^2 \eta^2 \frac{d(1-\beta_1)^2}{(1-\beta_2)(1-\frac{\beta_1^2}{\beta_2})} \right.$$

$$+ 2\eta L_1 (\partial_i f(\boldsymbol{x}_t))^2 \frac{\sqrt{d}(1-\beta_1)}{\sqrt{1-\beta_2}\sqrt{1-\frac{\beta_1^2}{\beta_2}}} \left. \right) \right)$$

$$+ \sum_{i=1}^{d} \frac{\eta\beta_1(1-\beta_1)\sqrt{\zeta}}{(1-\frac{\beta_1}{\sqrt{\beta_2}})\sqrt{1-\frac{\beta_1^2}{\beta_2}}} \left( \frac{(\partial_i f(\boldsymbol{x}_t))^2}{2\alpha_3\sqrt{\beta_2 \boldsymbol{v}_{t-1,i} + \zeta}} + \frac{\alpha_3(1-\beta_2)}{2\sqrt{\beta_2 \boldsymbol{v}_{t-1,i} + \zeta}} \right)$$

$$+ L_0\eta^2 \sum_{i=1}^{d} \frac{\frac{\beta_1}{2\sqrt{\beta_2}}\sqrt{d} + \frac{\beta_1^2}{\beta_2}\sqrt{d}}{\left(1-\frac{\beta_1}{\sqrt{\beta_2}}\right)^2} \frac{\boldsymbol{m}_{t-1,i}^2}{\boldsymbol{v}_{t-1,i}+\zeta} + \frac{\frac{\beta_1}{2\sqrt{\beta_2}}\sqrt{d}}{\left(1-\frac{\beta_1}{\sqrt{\beta_2}}\right)^2} \frac{\boldsymbol{m}_{t,i}^2}{\boldsymbol{v}_{t,i}+\zeta}$$

$$+ L_1 \frac{\frac{\beta_1^2}{\beta_2}}{\left(1-\frac{\beta_1}{\sqrt{\beta_2}}\right)^2} \left( \sum_{i=1}^{d} \frac{\eta(\partial_i f(\boldsymbol{x}_t))^2}{2\alpha_4\sqrt{\beta_2 \boldsymbol{v}_{t-1,i}+\zeta}} + \sum_{i=1}^{d} \frac{\alpha_4\eta^3 \boldsymbol{m}_{t-1,i}^2}{2\sqrt{\boldsymbol{v}_{t-1,i}+\zeta}} \frac{d(1-\beta_1)^2}{(1-\beta_2)(1-\frac{\beta_1^2}{\beta_2})} \right)$$

$$+ L_1 \frac{\frac{\beta_1}{\sqrt{\beta_2}}}{\left(1-\frac{\beta_1}{\sqrt{\beta_2}}\right)^2} \mathbb{E}\left[ \sum_{i=1}^{d} \frac{\eta(\partial_i f(\boldsymbol{x}_t))^2}{2\alpha_4\sqrt{\beta_2 \boldsymbol{v}_{t-1,i}+\zeta}} + \sum_{i=1}^{d} \frac{\alpha_4\eta^3 \boldsymbol{m}_{t,i}^2}{2\sqrt{\boldsymbol{v}_{t,i}+\zeta}} \frac{d(1-\beta_1)^2}{(1-\beta_2)(1-\frac{\beta_1^2}{\beta_2})} \right]. \tag{77}$$

Since $C_1 = 1 - \frac{\beta_1}{\sqrt{\beta_2}}$, and $C_2 = \sqrt{1-\frac{\beta_1^2}{\beta_2}}$, equation 77 can be further bounded as follows

$$\mathbb{E}[\langle \nabla f(\boldsymbol{u}_t), \boldsymbol{u}_{t+1} - \boldsymbol{u}_t \rangle | \mathcal{F}_t]$$

$$\leq \left[ -\frac{\eta(1-\beta_1)}{C_1} + \frac{\eta(1-\beta_1)}{C_1 C_2} \left( \frac{1}{2\alpha_0} + \frac{\alpha_0}{2\alpha_1} + \frac{\eta\alpha_0\sqrt{d}D_1 L_1(1-\beta_1)}{\sqrt{1-\beta_2}C_2} \right) + \frac{\eta\beta_1(1-\beta_1)\sqrt{\zeta}}{2\alpha_3 C_1 C_2} \right.$$

$$+ \left. \frac{\eta L_1(1-C_1)^2}{2\alpha_4 C_1^2} + \frac{\eta L_1(1-C_1)}{2\alpha_4 C_1^2} \right] \times \sum_{i=1}^{d} \frac{(\partial_i f(\boldsymbol{x}_t))^2}{\sqrt{\beta_2 \boldsymbol{v}_{t-1,i}+\zeta}}$$

$$+ \sum_{i=1}^{d} \frac{\eta(1-\beta_1)}{C_1 C_2} \left[ \frac{\alpha_0 D_0}{2} \left( \frac{1}{\sqrt{\beta_2 \boldsymbol{v}_{t-1,i}+\zeta}} - \mathbb{E}\left[ \frac{1}{\sqrt{\boldsymbol{v}_{t,i}+\zeta}} \middle| \mathcal{F}_t \right] \right) \right.$$

$$+ \left. \frac{\alpha_0 D_1}{2} \mathbb{E}\left[ \frac{(\partial_i \boldsymbol{f}(x_{t-1}))^2}{\sqrt{\beta_2 \boldsymbol{v}_{t-1,i}+\zeta}} - \frac{(\partial_i f(\boldsymbol{x}_t))^2}{\sqrt{\boldsymbol{v}_{t,i}+\zeta}} \middle| \mathcal{F}_t \right] \right] + \frac{\alpha_0 \alpha_1 D_1^2 L_0^2 \eta^3 d^2 (1-\beta_1)^3}{2(1-\beta_2)C_1 C_2^3 \sqrt{\zeta}} + \frac{\alpha_3 \eta d \beta_1 (1-\beta_1)(1-\beta_2)}{2C_1 C_2}$$

$$+ \sum_{i=1}^{d} \frac{\eta^2((1-C_1)^2 + 0.5(1-C_1))\sqrt{d}L_0}{C_1^2} \frac{\boldsymbol{m}_{t-1,i}^2}{\boldsymbol{v}_{t-1,i}+\zeta} + \sum_{i=1}^{d} \frac{\eta^2 0.5(1-C_1)\sqrt{d}L_0}{C_1^2} \mathbb{E}\left[ \frac{\boldsymbol{m}_{t,i}^2}{\boldsymbol{v}_{t,i}+\zeta} \middle| \mathcal{F}_t \right]$$

$$+ \sum_{i=1}^{d} \frac{\alpha_4 \eta^3 (1-\beta_1)^2 (1-C_1)^2 dL_1}{2(1-\beta_2)C_1^2 C_2^2} \frac{\boldsymbol{m}_{t-1,i}^2}{\sqrt{\boldsymbol{v}_{t-1,i}+\zeta}} + \sum_{i=1}^{d} \frac{\alpha_4 \eta^3 (1-\beta_1)^2 (1-C_1)dL_1}{2(1-\beta_2)C_1^2 C_2^2} \mathbb{E}\left[ \frac{\boldsymbol{m}_{t,i}^2}{\sqrt{\boldsymbol{v}_{t,i}+\zeta}} \middle| \mathcal{F}_t \right]. \tag{78}$$

This completes the proof. $\qquad \square$

## G   Proof of Lemma 4

For $(L_0, L_1)$-smooth objective functions, we have the following descent inequality (Lemma 1 in Crawshaw et al. (2022)):

$$\underbrace{\mathbb{E}[\langle \nabla f(\boldsymbol{u}_t), \boldsymbol{u}_t - \boldsymbol{u}_{t+1} \rangle | \mathcal{F}_t]}_{\text{first-order}} \leq f(\boldsymbol{u}_t) - \mathbb{E}[f(\boldsymbol{u}_{t+1}) | \mathcal{F}_t] + \underbrace{\sum_{i=1}^{d} \frac{L_0}{2\sqrt{d}} \mathbb{E}[\|\boldsymbol{u}_{t+1} - \boldsymbol{u}_t\| \|\boldsymbol{u}_{t+1,i} - \boldsymbol{u}_{t,i}\| | \mathcal{F}_t]}_{\text{second-order}}$$

$$+ \underbrace{\sum_{i=1}^{d} \frac{L_1 \|\partial_i f(\boldsymbol{u}_t)\|}{2} \mathbb{E}[\|\boldsymbol{u}_{t+1} - \boldsymbol{u}_t\| \|\boldsymbol{u}_{t+1,i} - \boldsymbol{u}_{t,i}\| | \mathcal{F}_t]}_{\text{additional term}}. \tag{79}$$

The first-order term is bounded by Lemma 9, we then only need to bound the remaining two terms. For the second-order term, based on the definition of $\boldsymbol{u}_t$ and update process of $\boldsymbol{x}_t$, we have that

$$\sum_{i=1}^{d} \frac{L_0}{2\sqrt{d}} \mathbb{E}[\|\boldsymbol{u}_{t+1} - \boldsymbol{u}_t\| \|\boldsymbol{u}_{t+1,i} - \boldsymbol{u}_{t,i}\| | \mathcal{F}_t]$$

$$\leq \sum_{i=1}^{d} \frac{L_0}{2\sqrt{d}} \mathbb{E}\left[ \frac{1}{2\sqrt{d}} \left\| \frac{\boldsymbol{x}_{t+1} - \boldsymbol{x}_t}{1 - \frac{\beta_1}{\sqrt{\beta_2}}} - \frac{\beta_1}{\sqrt{\beta_2}} \frac{\boldsymbol{x}_t - \boldsymbol{x}_{t-1}}{1 - \frac{\beta_1}{\sqrt{\beta_2}}} \right\|^2 + \frac{\sqrt{d}}{2} \left| \frac{\boldsymbol{x}_{t+1,i} - \boldsymbol{x}_{t,i}}{1 - \frac{\beta_1}{\sqrt{\beta_2}}} - \frac{\beta_1}{\sqrt{\beta_2}} \frac{\boldsymbol{x}_{t,i} - \boldsymbol{x}_{t-1,i}}{1 - \frac{\beta_1}{\sqrt{\beta_2}}} \right|^2 | \mathcal{F}_t \right]$$

$$\leq \sum_{i=1}^{d} \frac{L_0}{2\sqrt{d}} \mathbb{E}\left[ \frac{2\sqrt{d}}{(1 - \frac{\beta_1}{\sqrt{\beta_2}})^2} |\boldsymbol{x}_{t+1,i} - \boldsymbol{x}_{t,i}|^2 + \frac{2\sqrt{d} \frac{\beta_1^2}{\beta_2}}{(1 - \frac{\beta_1}{\sqrt{\beta_2}})^2} |\boldsymbol{x}_{t,i} - \boldsymbol{x}_{t-1,i}|^2 | \mathcal{F}_t \right]$$

$$= \sum_{i=1}^{d} \frac{L_0}{2\sqrt{d}} \left( \frac{2\sqrt{d}}{C_1^2} \mathbb{E}\left[ \frac{\eta^2 \boldsymbol{m}_{t,i}^2}{\boldsymbol{v}_{t,i} + \zeta} | \mathcal{F}_t \right] + \frac{2\sqrt{d}(1 - C_1)^2}{C_1^2} \frac{\eta^2 \boldsymbol{m}_{t-1,i}^2}{\boldsymbol{v}_{t-1,i} + \zeta} \right). \tag{80}$$

Now we focus on the additional term. According to the definition of $\boldsymbol{u}_t$ and update process of $\boldsymbol{x}_t$, for $\alpha_4 > 0$ we have that

$$\sum_{i=1}^{d} \frac{L_1 \|\partial_i f(\boldsymbol{u}_t)\|}{2} \mathbb{E}[\|\boldsymbol{u}_{t+1} - \boldsymbol{u}_t\| \|\boldsymbol{u}_{t+1,i} - \boldsymbol{u}_{t,i}\| | \mathcal{F}_t]$$

$$\leq \sum_{i=1}^{d} \frac{L_1}{2} (|\partial_i f(\boldsymbol{x}_t)| + (L_0 + L_1 |\partial_i f(\boldsymbol{x}_t)|) \|\boldsymbol{u}_t - \boldsymbol{x}_t\|) \mathbb{E}[\|\boldsymbol{u}_{t+1} - \boldsymbol{u}_t\| \|\boldsymbol{u}_{t+1,i} - \boldsymbol{u}_{t,i}\| | \mathcal{F}_t]$$

$$\leq \sum_{i=1}^{d} \frac{L_1}{2} \left( |\partial_i f(\boldsymbol{x}_t)| + L_0 \frac{\frac{\beta_1}{\sqrt{\beta_2}}}{1 - \frac{\beta_1}{\sqrt{\beta_2}}} \|\boldsymbol{x}_t - \boldsymbol{x}_{t-1}\| + L_1 |\partial_i f(\boldsymbol{x}_t)| \frac{\frac{\beta_1}{\sqrt{\beta_2}}}{1 - \frac{\beta_1}{\sqrt{\beta_2}}} \|\boldsymbol{x}_t - \boldsymbol{x}_{t-1}\| \right)$$

$$\times \mathbb{E}[\|\boldsymbol{u}_{t+1} - \boldsymbol{u}_t\| \|\boldsymbol{u}_{t+1,i} - \boldsymbol{u}_{t,i}\| | \mathcal{F}_t]$$

$$\leq \sum_{i=1}^{d} \frac{L_1}{2} \left( 1 + L_1 \frac{1 - C_1}{C_1} \eta \sqrt{d} \frac{1 - \beta_1}{\sqrt{1 - \beta_2} C_2} \right)$$

$$\times \left( \sqrt{d} \frac{2 - C_1}{C_1} \frac{\eta(1 - \beta_1)}{\sqrt{1 - \beta_2} C_2} |\partial_i f(\boldsymbol{x}_t)| \left( \frac{1}{C_1} \mathbb{E}\left[ \frac{\eta |\boldsymbol{m}_{t,i}|}{\sqrt{\boldsymbol{v}_{t,i} + \zeta}} | \mathcal{F}_t \right] + \frac{1 - C_1}{C_1} \frac{\eta |\boldsymbol{m}_{t-1,i}|}{\sqrt{\boldsymbol{v}_{t-1,i} + \zeta}} \right) \right)$$

$$+ \sum_{i=1}^{d} \frac{L_1 L_0}{2} \frac{1 - C_1}{C_1} \eta \sqrt{d} \frac{1 - \beta_1}{\sqrt{1 - \beta_2} C_2} \times \left( \frac{2\sqrt{d}}{C_1^2} \mathbb{E}\left[ \frac{\eta^2 \boldsymbol{m}_{t,i}^2}{\boldsymbol{v}_{t,i} + \zeta} | \mathcal{F}_t \right] + \frac{2\sqrt{d}(1 - C_1)^2}{C_1^2} \frac{\eta^2 \boldsymbol{m}_{t-1,i}^2}{\boldsymbol{v}_{t-1,i} + \zeta} \right)$$

$$\leq \sum_{i=1}^{d} \frac{L_1}{2} \left( 1 + L_1 \frac{1 - C_1}{C_1} \eta \sqrt{d} \frac{1 - \beta_1}{\sqrt{1 - \beta_2} C_2} \right)$$

$$\times \left( \frac{2\sqrt{d}}{C_1^2} \left( \frac{\eta}{2\alpha_4} \frac{(\partial_i f(\boldsymbol{x}_t))^2}{\sqrt{\beta_2 \boldsymbol{v}_{t-1,i} + \zeta}} + \frac{\alpha_4 \eta^3 (1 - \beta_1)^2}{2(1 - \beta_2) C_2^2} \mathbb{E}\left[ \frac{\boldsymbol{m}_{t,i}^2}{\sqrt{\boldsymbol{v}_{t,i} + \zeta}} | \mathcal{F}_t \right] \right) \right.$$

$$+ \frac{2\sqrt{d}(2 - C_1)^2}{C_1^2} \left( \frac{\eta}{2\alpha_4} \frac{(\partial_i f(\boldsymbol{x}_t))^2}{\sqrt{\beta_2 \boldsymbol{v}_{t-1,i} + \zeta}} + \frac{\alpha_4 \eta^3 (1 - \beta_1)^2}{2(1 - \beta_2) C_2^2} \frac{\boldsymbol{m}_{t-1,i}^2}{\sqrt{\boldsymbol{v}_{t-1,i} + \zeta}} \right) \right)$$

$$+ \sum_{i=1}^{d} \frac{L_1 L_0}{2} \frac{1 - C_1}{C_1} \eta \sqrt{d} \frac{1 - \beta_1}{\sqrt{1 - \beta_2} C_2} \left( \frac{2\sqrt{d}}{C_1^2} \mathbb{E}\left[ \frac{\eta^2 \boldsymbol{m}_{t,i}^2}{\boldsymbol{v}_{t,i} + \zeta} | \mathcal{F}_t \right] + \frac{2\sqrt{d}(1 - C_1)^2}{C_1^2} \frac{\eta^2 \boldsymbol{m}_{t-1,i}^2}{\boldsymbol{v}_{t-1,i} + \zeta} \right), \tag{81}$$

where the first inequality is due to the $(L_0, L_1)$-smoothness assumption, the third inequality is due to Lemma 6 and equation 80, and the last inequality is due to equation 75 and equation 76. Combine Lemma 9, equation 79, equation 80 and equation 81 together, and we have that

$$
\left( \frac{\eta(1-\beta_1)}{C_1} - \frac{\eta(1-\beta_1)}{C_1 C_2} \left( \frac{1}{2\alpha_0} + \frac{\alpha_0}{2\alpha_1} + \frac{\eta\alpha_0\sqrt{d}D_1 L_1(1-\beta_1)}{\sqrt{1-\beta_2}C_2} \right) - \frac{\eta\beta_1(1-\beta_1)\sqrt{\zeta}}{2\alpha_3 C_1 C_2} - \frac{\eta L_1(1-C_1)^2}{2\alpha_4 C_1^2} \right.
$$

$$
\left. - \frac{\eta L_1(1-C_1)}{2\alpha_4 C_1^2} - \frac{\sqrt{d}L_1}{2} \left( 1 + L_1 \frac{1-C_1}{C_1}\eta\sqrt{d}\frac{1-\beta_1}{\sqrt{1-\beta_2}C_2} \right) \left( \frac{\eta}{\alpha_4 C_1^2} + \frac{\eta(2-C_1)^2}{\alpha_4 C_1^2} \right) \right) \times \sum_{i=1}^{d} \frac{(\partial_i f(\boldsymbol{x}_t))^2}{\sqrt{\beta_2 \boldsymbol{v}_{t-1,i} + \zeta}}
$$

$$
\leq f(\boldsymbol{u}_t) - \mathbb{E}[f(\boldsymbol{u}_{t+1})|\mathcal{F}_t] + \sum_{i=1}^{d} \frac{\eta(1-\beta_1)}{C_1 C_2}
$$

$$
\times \left( \frac{\alpha_0 D_0}{2} \left( \frac{1}{\sqrt{\beta_2 \boldsymbol{v}_{t-1,i} + \zeta}} - \mathbb{E}\left[ \frac{1}{\sqrt{\boldsymbol{v}_{t,i} + \zeta}} \Big| \mathcal{F}_t \right] \right) + \frac{\alpha_0 D_1}{2}\mathbb{E}\left[ \frac{(\partial_i f(\boldsymbol{x}_{t-1}))^2}{\sqrt{\beta_2 \boldsymbol{v}_{t-1,i} + \zeta}} - \frac{(\partial_i f(\boldsymbol{x}_t))^2}{\sqrt{\boldsymbol{v}_{t,i} + \zeta}} \Big| \mathcal{F}_t \right] \right)
$$

$$
+ \frac{\alpha_0 \alpha_1 D_1^2 L_0^2 \eta^3 d^2(1-\beta_1)^3}{2(1-\beta_2)C_1 C_2^3 \sqrt{\zeta}} + \frac{\alpha_3 \eta d\beta_1(1-\beta_1)(1-\beta_2)}{2C_1 C_2}
$$

$$
+ \sum_{i=1}^{d} \left( \frac{\eta^2(2(1-C_1)^2 + 0.5(1-C_1))\sqrt{d}L_0}{C_1^2} + \frac{dL_1 L_0}{2}\frac{1-C_1}{C_1}\eta\frac{1-\beta_1}{\sqrt{1-\beta_2}C_2}\frac{2(1-C_1)^2}{C_1^2}\eta^2 \right) \frac{\boldsymbol{m}_{t-1,i}^2}{\boldsymbol{v}_{t-1,i} + \zeta}
$$

$$
+ \sum_{i=1}^{d} \left( \frac{\eta^2(0.5(1-C_1) + 1)\sqrt{d}L_0}{C_1^2} + \frac{dL_1 L_0}{2}\frac{1-C_1}{C_1}\eta\frac{1-\beta_1}{\sqrt{1-\beta_2}C_2}\frac{2}{C_1^2}\eta^2 \right) \mathbb{E}\left[ \frac{\boldsymbol{m}_{t,i}^2}{\boldsymbol{v}_{t,i} + \zeta} \Big| \mathcal{F}_t \right]
$$

$$
+ \sum_{i=1}^{d} \left( \frac{\alpha_4 \eta^3(1-\beta_1)^2(1-C_1)^2 dL_1}{2(1-\beta_2)C_1^2 C_2^2} \right.
$$

$$
\left. + \frac{L_1}{2}\left( 1 + L_1 \frac{1-C_1}{C_1}\eta\sqrt{d}\frac{1-\beta_1}{\sqrt{1-\beta_2}C_2} \right) \frac{2\sqrt{d}(2-C_1)^2}{C_1^2}\frac{\alpha_4 \eta^3(1-\beta_1)^2}{2(1-\beta_2)C_2^2} \right) \times \frac{\boldsymbol{m}_{t-1,i}^2}{\sqrt{\boldsymbol{v}_{t-1,i} + \zeta}}
$$

$$
+ \sum_{i=1}^{d} \left( \frac{\alpha_4 \eta^3(1-\beta_1)^2(1-C_1)dL_1}{2(1-\beta_2)C_1^2 C_2^2} + \frac{L_1}{2}\left( 1 + L_1\frac{1-C_1}{C_1}\eta\sqrt{d}\frac{1-\beta_1}{\sqrt{1-\beta_2}C_2} \right) \frac{2\sqrt{d}}{C_1^2}\frac{\alpha_4 \eta^3(1-\beta_1)^2}{2(1-\beta_2)C_2^2} \right)
$$

$$
\times \mathbb{E}\left[ \frac{\boldsymbol{m}_{t,i}^2}{\sqrt{\boldsymbol{v}_{t,i} + \zeta}} \Big| \mathcal{F}_t \right]. \tag{82}
$$

It is worth noting that (29) and (30) still hold for Adam since the update of $\boldsymbol{v}_t$ does not change. Specifically, for any $i \in [d]$ we have that

$$
\sum_{t=1}^{T} \mathbb{E}\left[ \frac{1}{\sqrt{\beta_2 \boldsymbol{v}_{t-1,i} + \zeta}} - \frac{1}{\sqrt{\boldsymbol{v}_{t,i} + \zeta}} \right] \leq \frac{1}{\sqrt{\zeta}} + T\frac{1-\sqrt{\beta_2}}{\sqrt{\zeta}}. \tag{83}
$$

Furthermore, since $\boldsymbol{x}_0 = \boldsymbol{x}_1$, we have that

$$
\mathbb{E}\left[ \frac{(\partial_i f(\boldsymbol{x}_0))^2}{\sqrt{\beta_2 \boldsymbol{v}_{0,i} + \zeta}} - \frac{(\partial_i f(\boldsymbol{x}_1))^2}{\sqrt{\boldsymbol{v}_{1,i} + \zeta}} \right] + \sum_{t=2}^{T} \mathbb{E}\left[ \frac{(\partial_i f(\boldsymbol{x}_{t-1}))^2}{\sqrt{\beta_2 \boldsymbol{v}_{t-1,i} + \zeta}} - \frac{(\partial_i f(\boldsymbol{x}_t))^2}{\sqrt{\boldsymbol{v}_{t,i} + \zeta}} \right]
$$

$$
\leq \frac{(\partial_i f(\boldsymbol{x}_1))^2}{\sqrt{\zeta}} + \sum_{t=1}^{T-1}(1-\beta_2)\mathbb{E}\left[ \frac{(\partial_i f(\boldsymbol{x}_t))^2}{\sqrt{\beta_2 \boldsymbol{v}_{t-1,i} + \zeta}} \right]. \tag{84}
$$

Taking expectations and telescoping (82) for $t = 1$ to $T$, and based on (83), (84), Lemma 7 and Lemma 8, we can show that

$$
\left( \frac{\eta(1-\beta_1)}{C_1} - \frac{\eta(1-\beta_1)}{C_1 C_2} \left( \frac{1}{2\alpha_0} + \frac{\alpha_0}{2\alpha_1} + \frac{\eta\alpha_0\sqrt{d}D_1 L_1(1-\beta_1)}{\sqrt{1-\beta_2}C_2} \right) - \frac{\eta\beta_1(1-\beta_1)\sqrt{\zeta}}{2\alpha_3 C_1 C_2} - \frac{\eta L_1(1-C_1)^2}{2\alpha_4 C_1^2} \right.
$$

$$-\frac{\eta L_1(1-C_1)}{2\alpha_4 C_1^2} - \frac{\sqrt{d}L_1}{2}\left(1 + L_1\frac{1-C_1}{C_1}\eta\sqrt{d}\frac{1-\beta_1}{\sqrt{1-\beta_2}C_2}\right)\left(\frac{\eta}{\alpha_4 C_1^2} + \frac{\eta(2-C_1)^2}{\alpha_4 C_1^2}\right)\right)$$

$$\times \sum_{t=1}^{T}\sum_{i=1}^{d}\frac{(\partial_i f(\boldsymbol{x}_t))^2}{\sqrt{\beta_2 \boldsymbol{v}_{t-1,i} + \zeta}}$$

$$\leq f(\boldsymbol{u}_1) - \mathbb{E}[f(\boldsymbol{u}_{T+1})|\mathcal{F}_t] + \sum_{i=1}^{d}\frac{\eta(1-\beta_1)}{C_1 C_2}$$

$$\times \left(\frac{\alpha_0 D_0}{2}\left(\frac{1}{\sqrt{\zeta}} + T\frac{1-\sqrt{\beta_2}}{\sqrt{\zeta}}\right) + \frac{\alpha_0 D_1}{2}\left(\frac{(\partial_i f(\boldsymbol{x}_1))^2}{\sqrt{\zeta}} + \sum_{t=1}^{T-1}(1-\beta_2)\mathbb{E}\left[\frac{(\partial_i f(\boldsymbol{x}_1))^2}{\sqrt{\beta_2 \boldsymbol{v}_{t-1,i} + \zeta}}\right]\right)\right)$$

$$+ T\frac{\alpha_0 \alpha_1 D_1^2 L_0^2 \eta^3 d^2(1-\beta_1)^3}{2(1-\beta_2)C_1 C_2^3 \sqrt{\zeta}} + T\frac{\alpha_3 \eta d\beta_1(1-\beta_1)(1-\beta_2)}{2C_1 C_2}$$

$$+ \left(\frac{\eta^2 2(1-C_1)^2\sqrt{d}L_0}{C_1^2} + \frac{\eta^2(2-C_1)\sqrt{d}L_0}{C_1^2} + \frac{dL_1 L_0}{2}\frac{1-C_1}{C_1}\eta\frac{1-\beta_1}{\sqrt{1-\beta_2}C_2}\frac{2(1-C_1)^2+2}{C_1^2}\eta^2\right)$$

$$\times \sum_{i=1}^{d}\mathbb{E}\left[\frac{(1-\beta_1)^2}{(1-\frac{\beta_1}{\sqrt{\beta_2}})^2(1-\beta_2)}\left(\ln\left(\frac{\boldsymbol{v}_{T,i}}{\boldsymbol{v}_{0,i}}\right) - T\ln(\beta_2)\right)\right]$$

$$+ \left(\frac{\alpha_4 \eta^3(1-\beta_1)^2 dL_1((1-C_1)+(1-C_1)^2)}{2(1-\beta_2)C_1^2 C_2^2}\right.$$

$$+ \frac{\sqrt{d}L_1}{2}\left(1 + L_1\frac{1-C_1}{C_1}\eta\sqrt{d}\frac{1-\beta_1}{\sqrt{1-\beta_2}C_2}\right)\frac{2+2(2-C_1)^2}{C_1^2}\frac{\alpha_4 \eta^3(1-\beta_1)^2}{2(1-\beta_2)C_2^2}\right)$$

$$\times \sum_{i=1}^{d}\mathbb{E}\left[\frac{(1-\beta_1)^2}{(1-\frac{\beta_1}{\sqrt[4]{\beta_2}})^2}\left(\frac{2}{1-\beta_2}(\sqrt{\boldsymbol{v}_{T,i}} - \sqrt{\boldsymbol{v}_{0,i}}) + \sum_{t=1}^{T}2\sqrt{\boldsymbol{v}_{t-1,i}}\right)\right]. \tag{85}$$

Moreover, for any $a > 0$ we have that $\ln(a) \leq a - 1$. We then have that

$$\frac{1}{T}\mathbb{E}[\ln(\boldsymbol{v}_{T,i})] = \frac{2}{T}\mathbb{E}[\ln(\sqrt{\boldsymbol{v}_{T,i}})] \leq \frac{2}{T}\mathbb{E}[\sqrt{\boldsymbol{v}_{T,i}}]$$

$$\leq \frac{2\sqrt{D_0 + \boldsymbol{v}_{0,i}}}{T} + \frac{2}{T}\sum_{i=0}^{T-1}\mathbb{E}\left[\sqrt{\beta_2^i(1-\beta_2)D_1}|\partial_i f(\boldsymbol{x}_{T-i})|\right]$$

$$\leq \frac{2\sqrt{D_0 + \boldsymbol{v}_{0,i}}}{T} + \frac{2}{T}\sum_{i=0}^{T-1}\mathbb{E}\left[\sqrt{(1-\beta_2)D_1}|\partial_i f(\boldsymbol{x}_{T-i})|\right], \tag{86}$$

where the second inequality is due to equation 38. In addition, for any $\beta_2 \geq 0.5$, we have that

$$-\ln(\beta_2) = \ln\left(\frac{1}{\beta_2}\right) \leq \frac{1}{\beta_2} - 1 \leq 2(1-\beta_2). \tag{87}$$

Combining equation 39, equation 85, equation 86 and equation 87, we then can show $\sum_{i=1}^{d}\sum_{t=1}^{T}\mathbb{E}\left[\frac{(\partial_i f(\boldsymbol{x}_t))^2}{\sqrt{\beta_2 \boldsymbol{v}_{t-1,i}+\zeta}}\right]$ is upper bounded by a function of $\sum_{t=1}^{T}\mathbb{E}\left[\|\nabla f(\boldsymbol{x}_t)\|\right]$. Specifically, we can get

$$\sum_{t=1}^{T}\sum_{i=1}^{d}\mathbb{E}\left[\frac{(\partial_i f(\boldsymbol{x}_t))^2}{\sqrt{\beta_2 \boldsymbol{v}_{t-1,i}+\zeta}}\right]\times\left(\frac{\eta(1-\beta_1)}{C_1} - \frac{\eta(1-\beta_1)}{C_1 C_2}\left(\frac{1}{2\alpha_0} + \frac{\alpha_0}{2\alpha_1} + \frac{\eta\alpha_0\sqrt{d}D_1 L_1(1-\beta_1)}{\sqrt{1-\beta_2}C_2}\right)\right.$$

$$-\frac{\eta\beta_1(1-\beta_1)\sqrt{\zeta}}{2\alpha_3 C_1 C_2} - \frac{\eta L_1(1-C_1)^2}{2\alpha_4 C_1^2} - \frac{\eta L_1(1-C_1)}{2\alpha_4 C_1^2}$$

$$-\frac{\sqrt{d}L_1}{2}\left(1+L_1\frac{1-C_1}{C_1}\eta\sqrt{d}\frac{1-\beta_1}{\sqrt{1-\beta_2}C_2}\right)\left(\frac{\eta}{\alpha_4C_1^2}+\frac{\eta(2-C_1)^2}{\alpha_4C_1^2}\right)-\frac{\eta(1-\beta_1)}{C_1C_2}\frac{\alpha_0D_1(1-\beta_2)}{2}\right)$$

$$\leq f(\boldsymbol{u}_1)-\mathbb{E}[f(\boldsymbol{u}_{T+1})|\mathcal{F}_t]+\frac{\eta\alpha_0dD_0(1-\beta_1)}{2C_1C_2\sqrt{\zeta}}+\frac{\eta\alpha_0D_1(1-\beta_1)\|\nabla f(\boldsymbol{x}_1)\|^2}{2C_1C_2\sqrt{\zeta}}$$

$$+T\frac{\eta d\alpha_0D_0(1-\beta_1)(1-\beta_2)}{2C_1C_2\sqrt{\zeta}}+T\frac{\alpha_0\alpha_1D_1^2L_0^2\eta^3d^2(1-\beta_1)^3}{2(1-\beta_2)C_1C_2^3\sqrt{\zeta}}+T\frac{\alpha_3\eta d\beta_1(1-\beta_1)(1-\beta_2)}{2C_1C_2}$$

$$+\left(\frac{\eta^2 2(1-C_1)^2\sqrt{d}L_0}{C_1^2}+\frac{\eta^2(2-C_1)\sqrt{d}L_0}{C_1^2}+\frac{dL_1L_0}{2}\frac{1-C_1}{C_1}\eta\frac{1-\beta_1}{\sqrt{1-\beta_2}C_2}\frac{2(1-C_1)^2+2}{C_1^2}\eta^2\right)$$

$$\times\frac{(1-\beta_1)^2}{(1-\frac{\beta_1}{\sqrt{\beta_2}})^2(1-\beta_2)}\sum_{i=1}^{d}\left(2\sqrt{D_0+\boldsymbol{v}_{0,i}}+\sum_{t=1}^{T}\mathbb{E}[2\sqrt{(1-\beta_2)D_1}|\partial_i f(\boldsymbol{x}_t)|]+2T(1-\beta_2)-\ln(\boldsymbol{v}_{0,i})\right)$$

$$+\left(\frac{\alpha_4\eta^3(1-\beta_1)^2dL_1((1-C_1)+(1-C_1)^2)}{2(1-\beta_2)C_1^2C_2^2}\right.$$

$$+\frac{\sqrt{d}L_1}{2}\left(1+L_1\frac{1-C_1}{C_1}\eta\sqrt{d}\frac{1-\beta_1}{\sqrt{1-\beta_2}C_2}\right)\frac{2+2(2-C_1)^2}{C_1^2}\frac{\alpha_4\eta^3(1-\beta_1)^2}{2(1-\beta_2)C_2^2}\right)$$

$$\times\sum_{i=1}^{d}\frac{(1-\beta_1)^2}{(1-\frac{\beta_1}{\sqrt[4]{\beta_2}})^2}\left(\frac{2}{1-\beta_2}\left(\sqrt{D_0+\boldsymbol{v}_{0,i}}+\sum_{t=1}^{T}\mathbb{E}[\sqrt{(1-\beta_2)D_1}|\partial_i f(\boldsymbol{x}_t)|]\right)\right.$$

$$\left.+2T\sqrt{D_0+\boldsymbol{v}_{0,i}}+\frac{4\sqrt{D_1}}{\sqrt{1-\beta_2}}\sum_{t=1}^{T}\mathbb{E}[|\partial_i f(\boldsymbol{x}_t)|]\right). \tag{88}$$

If we have $\alpha_0\geq\frac{21}{2C_2},\alpha_1\geq\frac{21\alpha_0}{2C_2},\alpha_3\geq\frac{7\beta_1\sqrt{\zeta}}{2C_2},\alpha_4\geq\frac{14L_1\sqrt{d}(2-C_1)^2}{C_1(1-\beta_1)}$, for $\eta\leq\min\left(\frac{C_2^2\sqrt{1-\beta_2}}{21\alpha_0\sqrt{d}D_1L_1(1-\beta_1)},\frac{C_1C_2\sqrt{1-\beta_2}}{\sqrt{d}L_1(1-C_1)(1-\beta_1)}\right), 1-\beta_2\leq(\frac{2C_2}{7\alpha_0D_1})$, then it can be shown that

$$\frac{\eta(1-\beta_1)}{C_1}-\frac{\eta(1-\beta_1)}{C_1C_2}\left(\frac{1}{2\alpha_0}+\frac{\alpha_0}{2\alpha_1}+\frac{\eta\alpha_0\sqrt{d}D_1L_1(1-\beta_1)}{\sqrt{1-\beta_2}C_2}\right)-\frac{\eta\beta_1(1-\beta_1)\sqrt{\zeta}}{2\alpha_3C_1C_2}$$

$$-\frac{\eta L_1(1-C_1)^2}{2\alpha_4C_1^2}-\frac{\eta L_1(1-C_1)}{2\alpha_4C_1^2}-\frac{\eta(1-\beta_1)}{C_1C_2}\frac{\alpha_0D_1(1-\beta_2)}{2}$$

$$-\frac{\sqrt{d}L_1}{2}\left(1+L_1\frac{1-C_1}{C_1}\eta\sqrt{d}\frac{1-\beta_1}{\sqrt{1-\beta_2}C_2}\right)\left(\frac{\eta}{\alpha_4C_1^2}+\frac{\eta(2-C_1)^2}{\alpha_4C_1^2}\right)\geq\frac{\eta(1-\beta_1)}{7C_1}. \tag{89}$$

Let $\Delta'=f(\boldsymbol{u}_1)-\inf_{\boldsymbol{x}}f(\boldsymbol{x})+\frac{\eta\alpha_0dD_0(1-\beta_1)}{2C_1C_2\sqrt{\zeta}}+\frac{\eta\alpha_0D_1(1-\beta_1)\|\nabla f(\boldsymbol{x}_1)\|^2}{2C_1C_2\sqrt{\zeta}}, C_3=\left(\frac{2(1-C_1)^2\sqrt{d}L_0}{C_1^2}+\frac{(2-C_1)\sqrt{d}L_0}{C_1^2}+\sqrt{d}L_0\frac{(1-C_1)^2+1}{C_1^2}\right), C_4=\left(\frac{\alpha_4(1-\beta_1)^2dL_1((1-C_1)+(1-C_1)^2)}{2C_1^2C_2^2}+\sqrt{d}L_1\frac{2+2(2-C_1)^2}{C_1^2}\frac{\alpha_4(1-\beta_1)^2}{2C_2^2}\right)$, we then have that

$$\frac{\eta(1-\beta_1)}{7C_1}\sum_{i=1}^{d}\sum_{t=1}^{T}\mathbb{E}\left[\frac{(\partial_i f(\boldsymbol{x}_t))^2}{\sqrt{\beta_2\boldsymbol{v}_{t-1,i}+\zeta}}\right]$$

$$\leq\Delta'+T\frac{\eta d\alpha_0D_0(1-\beta_1)(1-\beta_2)}{2C_1C_2\sqrt{\zeta}}+T\frac{\alpha_0\alpha_1D_1^2L_0^2\eta^3d^2(1-\beta_1)^3}{2(1-\beta_2)C_1C_2^3\sqrt{\zeta}}+T\frac{\alpha_3\eta d\beta_1(1-\beta_1)(1-\beta_2)}{2C_1C_2}$$

$$+\frac{\eta^2C_3(1-\beta_1)^2}{C_1^2(1-\beta_2)}\sum_{i=1}^{d}\left(2\sqrt{D_0+\boldsymbol{v}_{0,i}}+\sum_{t=1}^{T}\mathbb{E}[2\sqrt{(1-\beta_2)D_1}|\partial_i f(\boldsymbol{x}_t)|]+2T(1-\beta_2)-\ln(\boldsymbol{v}_{0,i})\right)$$

$$+\frac{\eta^3C_4}{(1-\beta_2)}\frac{(1-\beta_1)^2}{(1-\frac{\beta_1}{\sqrt[4]{\beta_2}})^2}\times\sum_{i=1}^{d}\left(\frac{2}{1-\beta_2}\left(\sqrt{D_0+\boldsymbol{v}_{0,i}}+\sum_{t=1}^{T}\mathbb{E}[\sqrt{(1-\beta_2)D_1}|\partial_i f(\boldsymbol{x}_t)|]\right)\right.$$

$$+ 2T\sqrt{D_0 + \boldsymbol{v}_{0,i}} + \frac{4\sqrt{D_1}}{\sqrt{1 - \beta_2}} \sum_{t=1}^{T} \mathbb{E}[|\partial_i f(\boldsymbol{x}_t)|]\bigg). \tag{90}$$

By rearranging the items in equation 90, it further follows that

$$\frac{1}{T} \sum_{t=1}^{T} \mathbb{E}\left[ \frac{\|\nabla f(\boldsymbol{x}_t)\|^2}{\sqrt{\beta_2 \|\boldsymbol{v}_{t-1}\|} + \zeta} \right]$$

$$\leq \frac{1}{T} \sum_{i}^{d} \sum_{t=1}^{T} \mathbb{E}\left[ \frac{(\partial_i f(\boldsymbol{x}_t))^2}{\sqrt{\beta_2 \boldsymbol{v}_{t-1,i}} + \zeta} \right]$$

$$\leq \frac{1}{T} \left( \frac{7C_1 \Delta'}{\eta(1 - \beta_1)} + \frac{7\eta C_3 (1 - \beta_1)}{C_1 (1 - \beta_2)} (2d\sqrt{D_0 + \|\boldsymbol{v}_0\|} - \sum_{i=1}^{d} \ln(\boldsymbol{v}_{0,i})) + \frac{14\eta^2 C_1 C_4 (1 - \beta_1) d}{(1 - \beta_2)^2 (1 - \frac{\beta_1}{\sqrt[4]{\beta_2}})^2} \sqrt{D_0 + \|\boldsymbol{v}_0\|} \right)$$

$$+ \frac{7\alpha_0 d D_0 (1 - \beta_2)}{2C_2 \sqrt{\zeta}} + \frac{7\alpha_0 \alpha_1 D_1^2 L_0^2 \eta^2 d^2 (1 - \beta_1)^2}{2(1 - \beta_2) C_2^3 \sqrt{\zeta}} + \frac{7\alpha_3 \beta_1 (1 - \beta_2) d}{2C_2} + \frac{14\eta d C_3 (1 - \beta_1)}{C_1}$$

$$+ \frac{14\eta^2 C_1 C_4 (1 - \beta_1) d \sqrt{D_0 + \|\boldsymbol{v}_0\|}}{(1 - \beta_2)(1 - \frac{\beta_1}{\sqrt[4]{\beta_2}})^2}$$

$$+ \left( \frac{14\eta C_3 (1 - \beta_1) \sqrt{D_1}}{C_1 \sqrt{1 - \beta_2}} + \frac{42\eta^2 C_1 C_4 (1 - \beta_1) \sqrt{D_1}}{(1 - \beta_2)^{1.5} (1 - \frac{\beta_1}{\sqrt[4]{\beta_2}})^2} \right) \left( \frac{\sum_{t=1}^{T} \sqrt{d} \mathbb{E}[\|\nabla f(\boldsymbol{x}_t)\|]}{T} \right). \tag{91}$$

For $\eta \leq C_5(1 - \beta_2)$, (where $C_5 > 0$ and will be introduced in Appendix H), $C_6 = \min\left( \frac{C_2 \sqrt{\zeta}}{21\alpha_0 d D_0}, \frac{C_2^3 \sqrt{\zeta}}{21\alpha_0 \alpha_1 D_1^2 L_0^2 (1 - \beta_1)^2 d^2 C_5^2}, \frac{C_2}{21\alpha_3 d \beta_1}, \frac{C_1}{84 C_3 C_5 d (1 - \beta_1)}, \frac{(1 - \frac{\beta_1}{\sqrt[4]{\beta_2}})^2}{84 C_1 C_4 C_5^2 (1 - \beta_1) d \sqrt{D_0 + \|\boldsymbol{v}_0\|}}, \frac{C_1^2}{784 C_3^2 C_5^2 (1 - \beta_1)^2 d D_1}, \right.$

$\left. \frac{(1 - \frac{\beta_1}{\sqrt[4]{\beta_2}})^4}{7056 C_1^2 C_4^2 C_5^4 d D_1} \right)$, $1 - \beta_2 \leq C_6 \epsilon^2$, and $T \geq \max\left( \frac{126 C_1 \Delta'}{\eta(1 - \beta_1)\epsilon^2}, \frac{126 C_3 C_5 (1 - \beta_1)}{C_1} (2d\sqrt{D_0 + \|\boldsymbol{v}_0\|} - \right.$

$\left. \sum_{i=1}^{d} \ln(\boldsymbol{v}_{0,i}))\epsilon^{-2}, \frac{252 C_5^2 C_1 C_4 (1 - \beta_1) d}{(1 - \frac{\beta_1}{\sqrt[4]{\beta_2}})^2} \sqrt{D_0 + \|\boldsymbol{v}_0\|}\epsilon^{-2} \right)$, equation 91 can be further written as

$$\frac{1}{T} \sum_{t=1}^{T} \mathbb{E}\left[ \frac{\|\nabla f(\boldsymbol{x}_t)\|^2}{\sqrt{\beta_2 \|\boldsymbol{v}_{t-1}\|} + \zeta} \right]$$

$$\leq \epsilon^2 + \left( \frac{14\eta C_3 (1 - \beta_1) \sqrt{d D_1}}{C_1 \sqrt{1 - \beta_2}} + \frac{42\eta^2 C_1 C_4 (1 - \beta_1) \sqrt{d D_1}}{(1 - \beta_2)^{1.5} (1 - \frac{\beta_1}{\sqrt[4]{\beta_2}})^2} \right) \left( \frac{\sum_{t=1}^{T} \mathbb{E}[\|\nabla f(\boldsymbol{x}_t)\|]}{T} \right). \tag{92}$$

Since we have that $\eta \leq C_5(1 - \beta_2)$ and $1 - \beta_2 \leq C_6 \epsilon^2$, thus we have that

$$\frac{1}{T} \sum_{t=1}^{T} \mathbb{E}\left[ \frac{\|\nabla f(\boldsymbol{x}_t)\|^2}{\sqrt{\beta_2 \|\boldsymbol{v}_{t-1}\|} + \zeta} \right] \leq \epsilon^2 + \epsilon \left( \frac{\sum_{t=1}^{T} \mathbb{E}[\|\nabla f(\boldsymbol{x}_t)\|]}{T} \right). \tag{93}$$

This completes the proof.

## H  Formal Version of Theorem 2 and Its Proof

For $\alpha_0 \geq \frac{21}{2C_2}, \alpha_1 \geq \frac{21\alpha_0}{2C_2}, \alpha_3 \geq \frac{7\beta_1 \sqrt{\zeta}}{2C_2}, \alpha_4 \geq \frac{14 L_1 \sqrt{d}(2 - C_1)^2}{C_1(1 - \beta_1)}, C_1, C_2$ defined in Appendix F, define

$$\Delta' = f(\boldsymbol{u}_1) - \inf_{\boldsymbol{x}} f(\boldsymbol{x}) + \frac{\eta\alpha_0 d D_0 (1 - \beta_1)}{2C_1 C_2 \sqrt{\zeta}} + \frac{\eta\alpha_0 D_1 (1 - \beta_1)\|\nabla f(\boldsymbol{x}_1)\|^2}{2C_1 C_2 \sqrt{\zeta}},$$

$$C_3 = \left( \frac{2(1 - C_1)^2 \sqrt{d} L_0}{C_1^2} + \frac{(2 - C_1) \sqrt{d} L_0}{C_1^2} + \sqrt{d} L_0 \frac{(1 - C_1)^2 + 1}{C_1^2} \right),$$

$$C_4 = \left( \frac{\alpha_4 (1 - \beta_1)^2 d L_1 ((1 - C_1) + (1 - C_1)^2)}{2 C_1^2 C_2^2} + \sqrt{d} L_1 \frac{2 + 2(2 - C_1)^2}{C_1^2} \frac{\alpha_4 (1 - \beta_1)^2}{2 C_2^2} \right),$$

$$C_5 = \min \left( \frac{C_1}{112 C_3 (1 - \beta_1) d D_1}, \frac{1 - \frac{\beta_1}{\sqrt[4]{\beta_2}}}{168 D_1 C_1 C_4 (1 - \beta_1) d} \right),$$

$$C_6 = \min \left( \frac{C_2 \sqrt{\zeta}}{21 \alpha_0 d D_0}, \frac{C_2^3 \sqrt{\zeta}}{21 \alpha_0 \alpha_1 D_1^2 L_0^2 (1 - \beta_1)^2 d^2 C_5^2}, \frac{C_2}{21 \alpha_3 d \beta_1}, \right.$$

$$\left. \frac{C_1}{84 C_3 C_5 d (1 - \beta_1)}, \frac{(1 - \frac{\beta_1}{\sqrt[4]{\beta_2}})^2}{84 C_1 C_4 C_5^2 (1 - \beta_1) d \sqrt{D_0 + \|v_0\|}}, \frac{C_1^2}{784 C_3^2 C_5^2 (1 - \beta_1)^2 d D_1}, \frac{(1 - \frac{\beta_1}{\sqrt[4]{\beta_2}})^4}{7056 C_1^2 C_4^2 C_5^4 d D_1} \right),$$

$$\Lambda_4 = \min \left( \frac{C_2^2}{21 \alpha_0 \sqrt{d} D_1 L_1 (1 - \beta_1)}, \frac{C_1 C_2}{\sqrt{d} L_1 (1 - C_1)(1 - \beta_1)} \right),$$

$$\Lambda_5 = \left( \frac{126 C_3 C_5 (1 - \beta_1)}{C_1} (2d \sqrt{D_0 + \|v_0\|} - \sum_{i=1}^{d} \ln(v_{0,i})) \right),$$

$$\Lambda_6 = \left( \frac{252 C_5^2 C_1 C_4 (1 - \beta_1) d}{(1 - \frac{\beta_1}{\sqrt[4]{\beta_2}})^2} \sqrt{D_0 + \|v_0\|} \right).$$

We then have the following theorem:

**Theorem 4.** *Let Assumptions 1, 2 and 3 hold. Let $1 - \beta_2 = \min \left( \frac{2 C_2}{7 \alpha_0 D_1}, C_6 \epsilon^2 \right) = \mathcal{O}(\epsilon^2)$, $0 < \beta_1 \leq \sqrt{\beta_2} < 1$, $\eta \leq \min \left( \Lambda_4 \sqrt{1 - \beta_2}, C_5 (1 - \beta_2) \right) = \mathcal{O}(\epsilon^2)$, and $T \geq \max \left( \frac{126 C_1 \Delta'}{\eta (1 - \beta_1) \epsilon^2}, \Lambda_5 \epsilon^{-2}, \Lambda_6 \epsilon^{-2} \right) = \mathcal{O}(\epsilon^{-4})$. For small $\epsilon$ such that $\epsilon \leq \frac{\sqrt{2 C_2}}{\sqrt{7 \alpha_0 C_6 D_1}}$, we have that*

$$\frac{1}{T} \sum_{t=1}^{T} \mathbb{E}[\|\nabla f(x_t)\|] \leq \left( 2c + \sqrt{2c} + \frac{4 \sqrt{d D_1}}{\sqrt{C_6}} \right) \epsilon. \tag{94}$$

*Proof.* The proof follows similarly as the one in Appendix D. According to equation 92 in the proof of Corollary 4, we have that

$$\frac{1}{T} \sum_{t=1}^{T} \mathbb{E} \left[ \frac{\|\nabla f(x_t)\|^2}{\sqrt{\beta_2 \|v_{t-1}\| + \zeta}} \right]$$

$$\leq \epsilon^2 + \left( \frac{14 \eta C_3 (1 - \beta_1) \sqrt{d D_1}}{C_1 \sqrt{1 - \beta_2}} + \frac{42 \eta^2 C_1 C_4 (1 - \beta_1) \sqrt{d D_1}}{(1 - \beta_2)^{1.5} (1 - \frac{\beta_1}{\sqrt[4]{\beta_2}})^2} \right) \left( \frac{\sum_{t=1}^{T} \mathbb{E}[\|\nabla f(x_t)\|]}{T} \right). \tag{95}$$

According to Lemma 3, we have that

$$\left( \frac{1}{T} \sum_{t=1}^{T} \mathbb{E}[\sqrt{\beta_2 \|v_{t-1}\| + \zeta}] \right)$$

$$\leq c + \frac{2 \sqrt{d D_1}}{\sqrt{(1 - \beta_2)}} \frac{\sum_{t=1}^{T} \mathbb{E}[\|\nabla f(x_t)\|]}{T}. \tag{96}$$

Define $e = \frac{1}{T} \sum_{t=1}^{T} \mathbb{E}[\|\nabla f(x_t)\|]$. By Hölder's inequality, we can show that

$$\left( \frac{1}{T} \sum_{t=1}^{T} \mathbb{E}[\|\nabla f(x_t)\|] \right)^2 \leq \left( \frac{1}{T} \sum_{t=1}^{T} \mathbb{E} \left[ \frac{\|\nabla f(x_t)\|^2}{\sqrt{\beta_2 \|v_{t-1}\| + \zeta}} \right] \right) \left( \frac{1}{T} \sum_{t=1}^{T} \mathbb{E}[\sqrt{\beta_2 \|v_{t-1}\| + \zeta}] \right). \tag{97}$$

By Lemma 3 and Corollary 4, equation 97 can be further written as

$$
e^2 \le \left( \epsilon^2 + \left( \frac{14\eta C_3(1-\beta_1)\sqrt{dD_1}}{C_1\sqrt{1-\beta_2}} + \frac{42\eta^2 C_1 C_4(1-\beta_1)\sqrt{dD_1}}{(1-\beta_2)^{1.5}(1-\frac{\beta_1}{\sqrt[4]{\beta_2}})^2} \right) e \right) \left( c + \frac{2\sqrt{dD_1}}{\sqrt{1-\beta_2}} e \right)
$$

$$
\le c\epsilon^2 + ce\epsilon + \frac{2\sqrt{dD_1}}{\sqrt{C_6}\epsilon} e\epsilon^2 + \frac{e^2}{2}, \tag{98}
$$

where the second inequality is due to the fact that $\left( \frac{14\eta C_3(1-\beta_1)\sqrt{dD_1}}{C_1\sqrt{1-\beta_2}} + \frac{42\eta^2 C_1 C_4(1-\beta_1)\sqrt{dD_1}}{(1-\beta_2)^{1.5}(1-\frac{\beta_1}{\sqrt[4]{\beta_2}})^2} \right) \le \epsilon$ and

$\left( \frac{14\eta C_3(1-\beta_1)\sqrt{dD_1}}{C_1\sqrt{1-\beta_2}} + \frac{42\eta^2 C_1 C_4(1-\beta_1)\sqrt{dD_1}}{(1-\beta_2)^{1.5}(1-\frac{\beta_1}{\sqrt[4]{\beta_2}})^2} \right) \frac{2\sqrt{dD_1}}{\sqrt{1-\beta_2}} \le \frac{1}{2}$ if $\eta \le C_5(1-\beta_2)$, $1-\beta_2 = \min\left( \frac{2C_2}{7\alpha_0 D_1}, C_6\epsilon^2 \right) = C_6\epsilon^2$

and $C_5 = \min\left( \frac{C_1}{112 C_3(1-\beta_1)dD_1}, \frac{1-\frac{\beta_1}{\sqrt[4]{\beta_2}}}{168 D_1 C_1 C_4(1-\beta_1)d} \right)$.

Thus, we have that

$$
\frac{1}{T} \sum_{t=1}^{T} \mathbb{E}[\|\nabla f(x_t)\|] = e \le \left( 2c + \sqrt{2c} + \frac{4\sqrt{dD_1}}{\sqrt{C_6}} \right) \epsilon,
$$

which completes the proof. $\qquad\square$

## I   Experiments

In this section, we provide numerical experiments to verify the coordinate-wise generalized smoothness and affine noise variance conditions. We follow the same setting of the LSTM language model (Zhang et al., 2019) for the Penn Treebank (PTB) (Mikolov et al., 2010) dataset. The model is a 3-layer LSTM language model with hidden size of 1150 and embedding size of 400. The training details follow Merity et al. (2017).

Given $\boldsymbol{x_t}$ and $\boldsymbol{x_{t+1}}$, we estimate the coordinate-wise smoothness by

$$
L_{t,i} = \max_{\gamma \in \{\delta_1, \delta_2, ...., \delta_N\}} \frac{|\partial_i f(\boldsymbol{x_t} + \gamma(\boldsymbol{x_{t+1}} - \boldsymbol{x_t})) - \partial_i f(\boldsymbol{x_t})|}{\gamma \|\boldsymbol{x_t} - \boldsymbol{x_{t+11}}\|}, \tag{99}
$$

where $\{\delta_1, \delta_2, ...., \delta_N\}$ denotes for the sample locations. We then show the training results for coordinate-wise smoothness vs. absolute gradient value in Fig. 3. In Fig. 4, we plot the coordinate-wise gradient standard deviation vs. absolute gradient value.

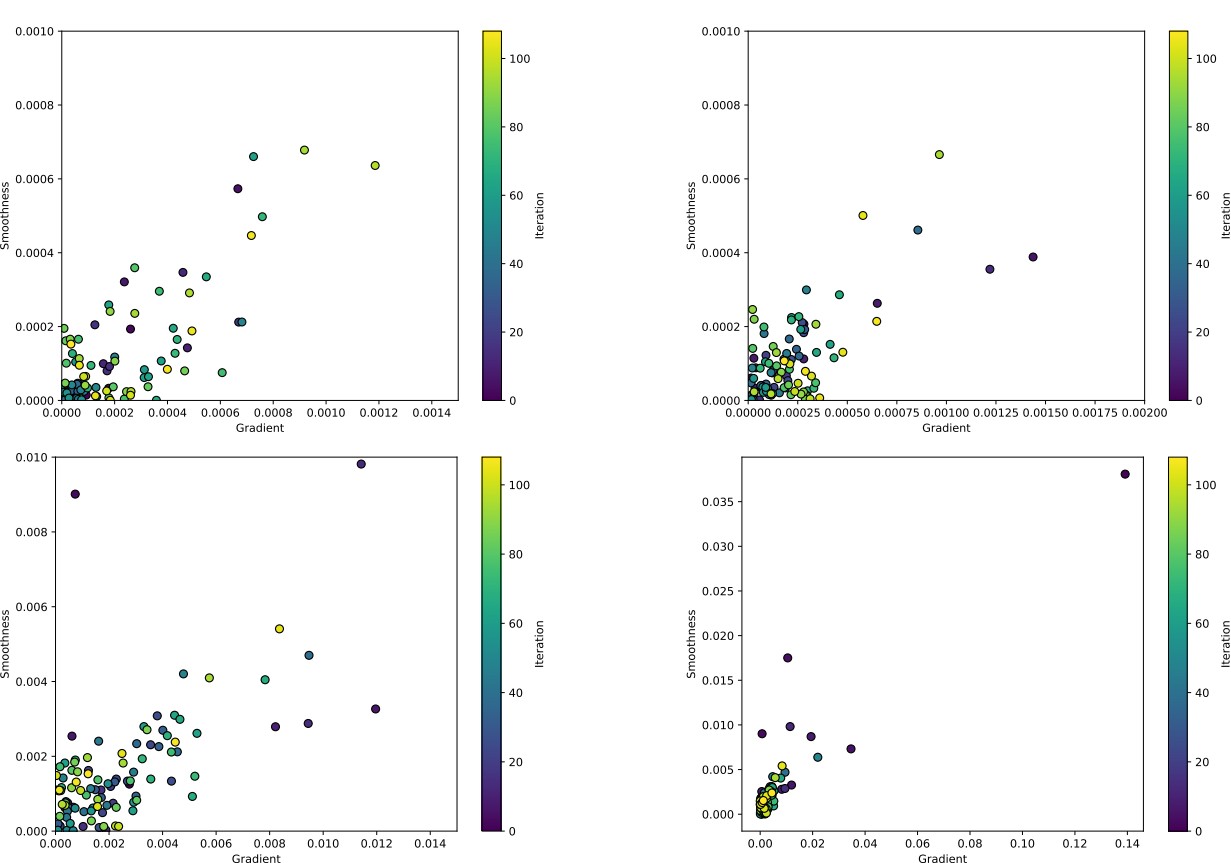

Figure 3: Coordinate-wise smoothness vs. absolute gradient value on LSTM language model for the PTB datatset. Each figure presents one randomly selected coordinate.

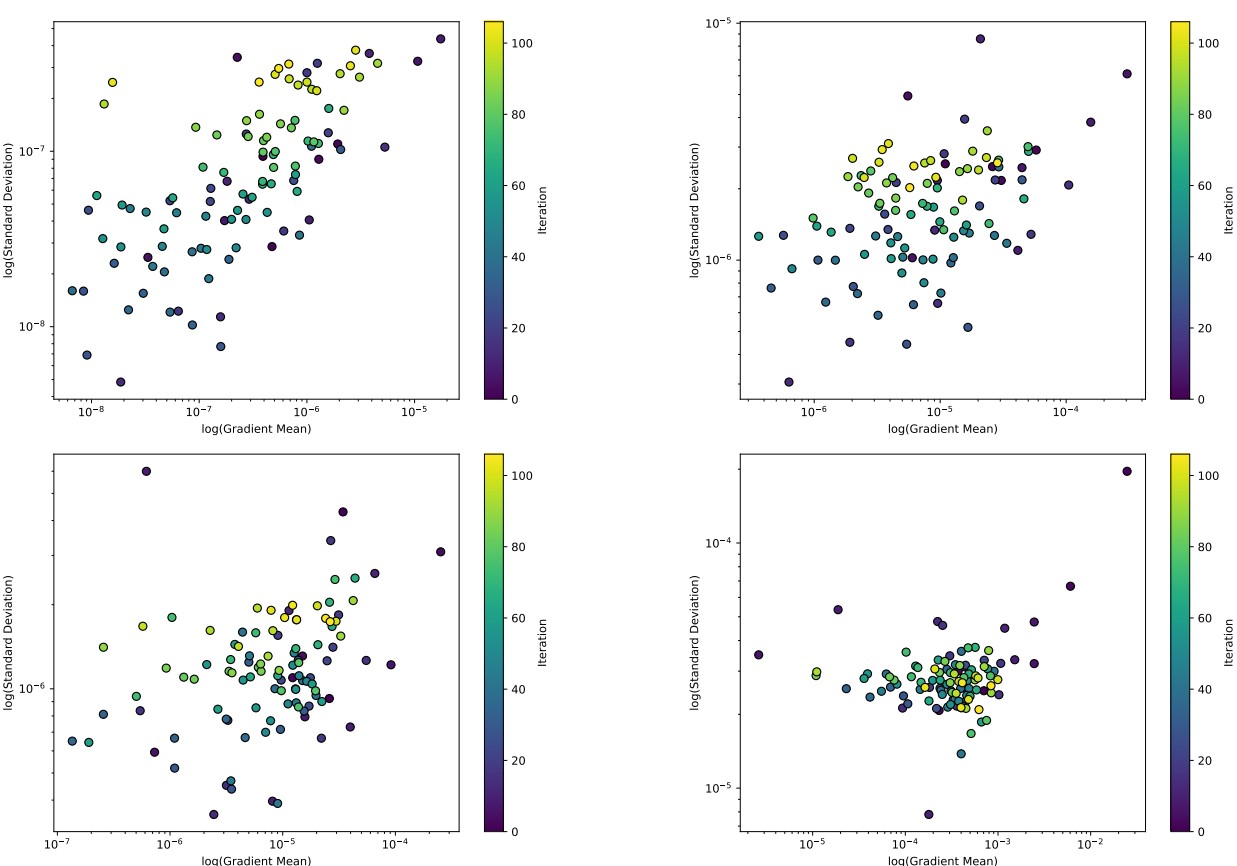

Figure 4: Coordinate-wise gradient standard deviation vs. absolute gradient value on LSTM language model for the PTB datatset. Each figure presents one randomly selected coordinate.

