# OpenReview forum: "Convergence Guarantees for RMSProp and Adam in Generalized-smooth Non-convex Optimization with Affine Noise Variance"
_TMLR — Accepted by TMLR_

### Review · Reviewer_fhNs · 2024-10-29

**Summary Of Contributions:**

The authors establish the convergence of both RMSProp and Adam to a $\epsilon$-stationary point with iteration complexity of $\mathcal{O}(\epsilon^{-4})$ for functions verifying coordinate-wise generalized $(L_0,L_1)$-smoothness assumption and under affine noise variance condition. Their approach focuses specifically on neural network training.

**Audience:**

Yes

**Broader Impact Concerns:**

I have no ethical concerns for this paper.

**Claims And Evidence:**

Yes

**Requested Changes:**

My only request is that the authors highlight examples where the combination of the two assumptions is particularly relevant, in order to reinforce the interest of this particular framework. That said, I think the proposed article is of sufficient interest to the community to be published in TMLR.

Minor comments :
- Page 12 : In Figure 1, you could specify what the parameter $\lambda$ corresponds to.
- Page 12 : "The only thing inspired by Wang et al. (2023a) in this paper to set this $\beta = \frac{\beta_1}{\sqrt{\beta_2}}$." -> The verb is missing.
- Page 16 : "RKS" should be RHS.
- Page 17 : $\frac{\vert g_{t,i}\vert}{\sqrt{v_{t,i}}+\zeta}$ should be $\frac{\vert g_{t,i}\vert}{\sqrt{v_{t,i}+\zeta}}$
- Page 19 : In equation (27), I may be wrong, but shouldn't we have a factor $\frac{1}{4}$ instead of $\frac{1}{2}$ for the last two terms in the last inequality?
- Page 22 : "Give" -> Given.
- Page 27 : "haveequation" -> have equation.
- Page 29 : Last line : "$2\beta_2\geq 1$" -> $2\beta_2> 1$.

**Strengths And Weaknesses:**

**Strengths :** The paper shows a tight bound for convergence analysis to an $\epsilon$-stationary point for RMSProp and Adam under relaxed assumptions with respect of what is done in the literature, which the authors effectively situate within the literature. The technical content is robust; the proofs appear correct, to the best of my knowledge. Understanding of the main results through the well-presented key ideas in the main text (with detailed evidence available in the appendix) is particularly appreciated.

**Weaknesses :** The assumption that $f$ is differentiable is common in convergence analyses for neural network training, but is rarely the case in practice. In particular, it is said the chosen assumptions "best describe the training of some neural networks". A clarification on neural networks that fall within the framework of coordinated generalized smoothness $(L_0,L_1)$ and affine noise variance condition could be added; along with a note on the neural networks (or the type of functions) that verify these assumptions (the more relaxed one), but not those otherwise considered in the literature.

Please note that my expertise lies outside the direct focus of this paper, therefore I am not familiar with the literature around RMSProp and Adam, so I may be missing references and insights.

---

> ### Author Response · Authors · 2024-12-14
>
> **Weakness:**
> Thanks for the question.
> The neural network is not globally differentiable but almost differentiable everywhere. The statement of "best describe the training of some neural networks" is for the coordinate-wise generalized smoothness and affine noise variance conditions in this paper, compared with the widely-used standard $L$-smooth and bounded gradient variance conditions.
>
> **Clarification for Generalized Smoothness:**  The generalized smoothness is first observed on the observation of training for LSTM language models and ResNet20 in Zhang et al., (2019). The 3-layer LSTM language model with hidden size of 1150 and embedding size of 400 and ResNet20  (He et al., 2016) model are studied. Later, Crawshaw et al., (2022) observes the coordinate-wise generalized smoothness condition on a 2-layer Transformer Encoder on the Wikitext-2 dataset and a 6-layer Transformer on the WMT’16 Multimodal Machine Translation Task German-English dataset. In Appendix I, we train the LSTM language model with the same setting as Zhang et al., (2019) and show the coordinate-wise generalized smoothness condition.
>
> **Clarification for Affine Noise Variance:**  The affine noise variance condition is generalized from the weak growth condition (Vaswani et al. 2019). The weak growth condition $\mathbb E[\|\boldsymbol g_{t}\|^2|\mathcal F_t]\le D_1 \|\nabla f(\boldsymbol x_{t})\|^2$ is a special case of the affine noise variance condition and is motivated by overparameterized neural networks since the gradient for every sample can obtain $0$ for these overparameterized neural networks.  In Appendix I, we train the LSTM language model with the same setting as Zhang et al., (2019) and show the coordinate-wise affine noise variance condition.
>
> **Requested Change:** Thanks for the question. As mentioned in the previous answer for **Weakness**, the generalized smoothness is widely observed in neural networks such as LSTM, transformers, and ResNet models. The affine noise variance condition is also reasonable for neural networks. Moreover, we show the coordinate-wise generalized smoothness and coordinate-wise affine noise variance conditions in the training of LSTM. Thus, we believe our assumptions are reasonable and practical.
>
> **Minor comments:** Thanks for the question. We have fixed the typos, and for Page 19, there should be $\frac{1}{4}$ and we further bound it   $\frac{1}{2}$.
>
> Reference:
>
> Jingzhao Zhang, Tianxing He, Suvrit Sra, and Ali Jadbabaie. Why gradient clipping accelerates training: A
> theoretical justification for adaptivity. arXiv preprint arXiv:1905.11881, 2019.
>
> Michael Crawshaw, Mingrui Liu, Francesco Orabona, Wei Zhang, and Zhenxun Zhuang. Robustness to
> unbounded smoothness of generalized signsgd. Advances in Neural Information Processing Systems, 35:
> 9955–9968, 2022.
>
> Sharan Vaswani, Francis Bach, and Mark Schmidt. Fast and faster convergence of SGD for overparameterized models and an accelerated perceptron. In International Conference on Artificial Intelligence
> and Atatistics, pp. 1195–1204. PMLR, 2019.
>
> He, Kaiming, Xiangyu Zhang, Shaoqing Ren, and Jian Sun. "Deep residual learning for image recognition." In Proceedings of the IEEE conference on computer vision and pattern recognition, pp. 770-778. 2016.

---

> > ### Comment · Reviewer_fhNs · 2024-12-16
> > **Response to revised version**
> >
> > Thank you for your detailed responses on the relevance of your assumptions and the additional experiments, which I find convincing. I am satisfied with the revision. (There is a minor typo "cooridinate" in the captions of Figures 3 and 4.)

---

> > > ### Author Response · Authors · 2024-12-16
> > >
> > > Dear Reviewer fhNs,
> > >
> > > Thanks for your response. We have fixed the typo in the revision. Please let us know if you have more questions.
> > >
> > > Best,
> > > the authors

---

### Review · Reviewer_p2PR · 2024-11-04

**Summary Of Contributions:**

This paper analyzes the convergence behavior of RMSProp and Adam as two well-known primary optimizers for different variations of neural networks (Convolutional Neural Networks, LSTM, Transformers) under more realistic assumptions of coordinate-wise Lipschitz continuity (smoothness) and coordinate-wise affine noise variance of the gradient. While there are several recent papers showing the convergence of these optimizers under each one of these two assumptions combined with stronger assumptions, this is the first paper that have both of these more general assumptions compared to the previous works. The main challenge of showing the convergence of these algorithms under these conditions is that due to the dependence of the step-size to the gradient size, bounding the first-order surrogate error is not straightforward. To address this issue, the authors slightly change the update rule for the step-size of Adam. They numerically show that by applying this change, the performance of the optimizer in the test phase is not drastically affected. By changing the update rule, they bounded the first-order surrogate error, and therefore, given the rich theory for proving the convergence of Adam and RMSProp, the remaining steps are straightforward.

**Audience:**

Yes

**Broader Impact Concerns:**

No Concern.

**Claims And Evidence:**

Yes

**Requested Changes:**

Suggestions:

1. There are several issues in proving theorems.

- Lemma 1 should be used as an upper-bound. I am not sure why we seek to lower-bound the first-order.b term in Lemma 1. Therefore, the inequality must be flipped in Lemma 1 and its formal version (Lemma 5). The explanation in the beginning of Page 8 must be revised accordingly. By following the proof of Lemma 5, we can observe that the term is upper-bounded but the Lemma statement is written as a lower-bound. Also, the authors use Lemma 1 in the proof of Lemma 2 correctly (as an upper-bound). I also suggest to explain more in Lemma 2’s proof how you use Lemma 1.

- In the proof of Lemma 2, the authors just mentioned “Based on Lemma 1, we can bound the first-order.b term. We then have that.” Please explain how Lemma 1 can be applied and what will be the result. The current version is confusing and needs efforts for the readers to understand the proof

2. The change in the Adam algorithm is for the proof convenience. To justify it, either there should be an intuitive explanation (why does the original version have that form? And, how this change conceptually change the algorithm). Also, it will be nice to validate the test error for another type of neural networks such as transformers. Because, the CIFAR10 dataset is not a very challenging one.

3. The paper does suggest the limitation of the bounded gradient and $(L_0, L_1)$ smoothness even for the linear regression and mentions the coordinate-wise version is more general. Is it possible to validate it for a simple multi-layer neural network or a small convolutional neural network to motivate the assumption (as the main contribution of the paper) for practitioners?

**Strengths And Weaknesses:**

Strengths:

1. Major theorems and lemmas are generally correct and the paper is technically sound.
2. The paper demonstrates the limitation of the previous works in the literature showing that most convergence theorems are based on the assumptions that are not hold for widely used neural networks.
3. The paper shows the minor change to the algorithm does not drastically affect the performance of the convolutional neural network trained on CIFAR10 dataset.
4. The paper explains the main challenges of using coordinate-wise $(L_0, L_1)$ smoothness instead of bounding gradient and $(L_0, L_1)$ smoothness.

Weaknesses (the suggestions for them is listed in the next section)
1) There is an error in the presentation of Lemma 1 (and its extended version Lemma 5). Also, the explanation of its usage in the beginning of Page 8 is not clear (and needs to be revised).

2) It is not clear how Lemma 1 is applied in the proof of Lemma 2.

3) The change in the Adam algorithm has no motivation except the convenience in the proof. It is expected to discuss how it changes the behavior of the algorithm. While it is numerically shown that it does not change the performance drastically, it can be further explored for other neural network architectures.

---

> ### Author Response · Authors · 2024-12-14
>
> **Weakness 1:** See requested changes 1.
>
> **Weakness 2:** See requested changes 1.
>
> **Weakness 3:** See requested changes 3.
>
> **Requested Changes 1a:**
>
> Thanks for this question. In our stage I, we want to provide the upper bound $\mathbb E\left[\frac{ \|\nabla f(\boldsymbol x_t)\|^2}{\sqrt{{\beta_2 \|\boldsymbol v_{t-1}} \|+ \zeta}}\right]$, which can be obtained by the upper bound of First Order.a in eq (4). Based on eq (2), we can provide an upper bound on First order term, which is the sum of First Order.a  and First Order.b. Thus by providing a lower bound on First Order.b and an upper bound on First Order term, we can have an upper bound on First Order.a. In eq (23) of Lemma 5, we show an upper bound in eq (23), which is the opposite number of First Order.b. Thus we can have the lower bound on First Order.b.
>
> For the proof in Lemma 2, in eq (24) it is the opposite number of First Order.b and we have fixed this typo.
>
> **Requested Changes 1b:**
>
> Thanks for pointing this problem out. We have added the following explanations:
>
> **In Lemma 1, we provide a lower bound on the First Order.b term. By plugging in Lemma 1 to eq (24), we have the following inequality...**
>
> **Requested Changes 2:**
>
> Thanks for this question.
> Our modified update $\boldsymbol x_{t+1} \gets\boldsymbol  x_{t} - \eta  \frac{1}{\sqrt{{\boldsymbol v_t} + \zeta} } \odot \boldsymbol m_t$ is the same as the original Adam update $\boldsymbol x_{t+1} \gets\boldsymbol  x_{t} - \eta  \frac{1}{\sqrt{{\boldsymbol v_t}} + \lambda } \odot \boldsymbol m_t$ if $\zeta=\lambda=0$.  In both the original and our modified Adam, $\lambda$ and $\zeta$ are used to avoid large updates when $\sqrt{{\boldsymbol v_t}} $ is small and $\frac{\boldsymbol m_t}{\sqrt{{\boldsymbol v_t}} }$ is large. We have added the test error for vision-transformers on CIFAR10. More details can be found in Fig. 2.
>
> **Requested Changes 3:**
>
> Thanks for this question. We have added the numerical results for LSTM model to motivate our coordinate-wise assumptions. Details can be found in Appendix I.

---

> > ### Author Response · Authors · 2025-01-24
> > **Follow-Up on Response Feedback**
> >
> > Dear Reviewer p2PR,
> >
> > Hope this message finds you well. It has been some time since we submitted our responses, and we have not yet received your feedback.
> >
> > We kindly ask if you could confirm whether our responses have adequately addressed your concerns. Please let us know if further clarification or additional information is needed—we would be happy to provide it.
> >
> > Thank you for your time and attention.
> >
> > Best regards,
> >
> > The authors

---

> > > ### Comment · Reviewer_p2PR · 2025-01-26
> > > **Thanks for the revision**
> > >
> > > Thanks for addressing the mentioned concerns properly. I request no further changes.

---

### Review · Reviewer_McMB · 2024-11-19

**Summary Of Contributions:**

This paper studies the convergence properties of RMSProp and Adam in non-convex optimization settings. RMSProp and Adam are analyzed under relaxed assumptions: coordinate-wise generalized smoothness and affine noise variance. The paper addresses challenges like the dependence of adaptive updates, unbounded gradient estimates, and Lipschitz constants, and establishes a convergence to an $\epsilon$-stationary point with an iteration complexity of $O(\epsilon^{-4})$, which matches existing analysis under similar assumptions.

**Audience:**

Yes

**Broader Impact Concerns:**

The paper studies the theoretical convergence of widely used optimizers. It will give us more theoretical understanding of adaptive optimizers and will not lead to other impact concerns as far as I see.

**Claims And Evidence:**

Yes

**Requested Changes:**

Please correct or restate the claims in an more accurate way, which are mentioned in the weakness part.

**Strengths And Weaknesses:**

Strengths

**Technical Depth**: The authors develop intricate proofs to overcome significant technical challenges, such as handling unbounded gradient estimates and mismatched terms in the descent lemma.

**Comprehensive Coverage**: Both RMSProp (without momentum) and Adam (with momentum) are analyzed, and their coordinate-wise version is analyzed, which is more detailed comparing to existing work.

**Clear Structure and Proof**: The manuscript is well-structured, and the proof is presented in a manner that is easy to understand, which facilitates the reader's comprehension of the convergence analysis.

Weaknesses

**Inaccurate Claim**

The paper's claim that the result matches the lower bound in Arjevani et al. (2023) is inaccurate because this paper is developed under the assumptions of relaxed smoothness of the objective while Arjevani et al. (2023) derives the lower bound under the assumptions of  global L-smooth condition of the objective.

The paper claims that it improves logarithmic factors in iteration complexity than Wang et al. 2023a. It is not an accurate claim as the logarithmic factors are addressed in  Section 5 Theorem 2 of  Wang et al., 2023a.

Moreover, the paper's techniques are not much of innovations. Given the result of Li et al., 2023, the contribution of this paper is the expansion of the noise assumption to coordinate-wise affine noise. This extension is not technically hard given existing works have already considered affine noise assumptions.

---

> ### Author Response · Authors · 2024-12-13
>
> **Inaccurate Claim:**
>
> **The paper's claim that... condition of the objective:**
>
> Thanks for the question. The lower bound in Arjevani et al. (2023) is a minimax bound, which is the supremum over a set
> $\mathcal F(\Delta,L):=\{F:\mathbb R^d\to \mathbb R, s.t. F(0)-\inf_xF(x)\le \Delta,
>  \|\nabla F(x)-\nabla F(y) \|\le L\|x-y\| \text{for all} x,  y\}.$ For our generalized smooth setting, the lower bound is the supremum over a set
> $\mathcal F(\Delta,L_0,L_1):=\{F:\mathbb R^d\to \mathbb R, s.t. F(0)-\inf_xF(x)\le \Delta,
>  \|\nabla F(x)-\nabla F(y) \|\le (L_0+L_1\|\nabla F(x)\|\|x-y\| \text{for all} x,  y\}. $  Since $\mathcal F(\Delta,L)$ is a subset of $\mathcal F(\Delta,L_0,L_1)$, we have the new lower bound $\ge \mathcal O(\epsilon^{-4})$. Moreover, we show the complexity for our method $= \mathcal O(\epsilon^{-4})$. As a result, the lower bound $= \mathcal O(\epsilon^{-4})$.
>
> **The paper claims that it ...of Wang et al., (2023a):**
>
> Thanks for the reminder. We have corrected the claims.
>
> **Moreover, the paper's...noise assumptions:**
>
> Thanks for the question.
>
> **Comparation with Li et al., (2023):** The analysis of Li et al., (2023) is fundamentally different compared with this paper. Li et al., (2023) introduces a stopping time, and shows that under their almost-surely bounded gradient noise/sub-Gaussian norm gradient noise assumption, the gradients and local smoothness are bounded along their optimization trajectory with high probability. Thus this problem reduces to the  $L$-smoothness. However, they need stronger assumptions on noise and the convergence with high probability is weaker than our convergence in expectation.  Our proof does not bound the gradient norm and local smoothness. Instead, in our stage I, we introduce a surrogate and bound $\mathbb E[{\frac{\|\nabla f(x_t)\|^2}{\sqrt{\beta_2\|v_{t-1}\|+\zeta}}}]$, where $\|v_{t-1}\|$ is potentially unbounded. In our stage II we then bound $\mathbb E[\sqrt{\beta_2\|v_{t-1}\|+\zeta}]$. We then get the convergence of $\mathbb E[\|\nabla f(x_t)\|]$ by the H\"older's inequality. Our method does not require the gradient norm in the training trajectory bounded. In other words, due to our different analyses, we do not require the almost surely bounded gradient
> noise/sub-Gaussian norm gradient noise assumptions, and we show the convergence in expectation instead of with high probability.
>
> **Innovations:** The contribution of this paper is not only limited to the expansion of the noise assumption for Adam, but also includes the study of coordinate-wise generalized smoothness conditions for Adam. Except for our concurrent work (Wang et al. 2024), only (Li et al., 2023) in the existing studies on Adam studied the generalized smoothness condition. However, they only show the convergence with high probability and the method can not be extended to coordinate-wise affine noise variance conditions.
>
> The other studies on affine noise variance conditions are limited to simpler algorithms such as Normalized momentum (Jin et al., 2021) and Spider (Chen et al., 2023), where the adaptive stepsize uses a sample normalized term to control the uniformly scaled gradient step.
> In Adam, the second moment is used to estimate the gradient variance and normalize the update, where the update steps vary. Thus, the above analysis cannot be extended to Adam directly.
>
>
> Under our coordinate-wise generalized smoothness and affine noise variance conditions, there are two challenges: the surrogate error is harder to bound and the adaptive stepsize can be close to $0$. By applying the descent lemma, we can demonstrate that the squared gradient norm multiplied by stepsize is bounded by a function of the gradient norm.  We then show the inverse of the adaptive stepsize can be bounded by a function of gradient norm. Finally, based on the H\"{o}lder's inequality, we prove the upper bound on the gradient norm.

---

> > ### Author Response · Authors · 2024-12-13
> >
> > Reference:
> >
> > Yossi Arjevani, Yair Carmon, John C Duchi, Dylan J Foster, Nathan Srebro, and Blake Woodworth. Lower
> > bounds for non-convex stochastic optimization. Mathematical Programming, 199(1-2):165–214, 2023.
> >
> > Bohan Wang, Jingwen Fu, Huishuai Zhang, Nanning Zheng, and Wei Chen. Closing the gap between the
> > upper bound and the lower bound of Adam’s iteration complexity. arXiv preprint arXiv:2310.17998,
> > 2023a.
> >
> > Haochuan Li, Ali Jadbabaie, and Alexander Rakhlin. Convergence of Adam under relaxed assumptions.
> > arXiv preprint arXiv:2304.13972, 2023.
> >
> > Bohan Wang, Huishuai Zhang, Qi Meng, Ruoyu Sun, Zhi-Ming Ma, and Wei Chen. On the convergence of adam under non-uniform smoothness: Separability from sgdm and beyond. arXiv preprint
> > arXiv:2403.15146, 2024.
> >
> > Jikai Jin, Bohang Zhang, Haiyang Wang, and Liwei Wang. Non-convex distributionally robust optimization:
> > Non-asymptotic analysis. Advances in Neural Information Processing Systems, 34:2771–2782, 2021.
> >
> > Ziyi Chen, Yi Zhou, Yingbin Liang, and Zhaosong Lu. Generalized-smooth nonconvex optimization is as
> > efficient as smooth nonconvex optimization. arXiv preprint arXiv:2303.02854, 2023.

---

> > ### Comment · Reviewer_McMB · 2024-12-25
> > **Thanks for the feedback**
> >
> > Thanks for the extensive explanation. The author's feedback resolved my concerns. I have no further questions and I lean towards the acceptatnce of the paper.

---

### Decision · Action_Editor_8ckL · 2025-01-29

**Recommendation:** Accept as is

**Comment:**

All three reviewers are positive enough and any of their concerns were addressed in the rebuttal.  The only outstanding issue is, to quote from a reviewer's final decision,

"If the authors can suggest several practical scenarios in which the assumptions work beyond the established literature, the paper can gain significantly more attention."

I agree with that. I know the paper already talks about the assumptions, but being even more clear or with more examples can only help. So if the authors wish to add more discussion/examples/citations here, they may do so in the final camera-ready version (but it's not a strict requirement, so up to their judgment).

**Audience:**

This is a technical analysis, so that restricts the audience some. But these algorithms are so well-known that I find there is still enough audience in the TMLR readership to make this paper relevant.

**Claims And Evidence:**

The paper analyzes Adam and RMSProp (or slight variations) under different assumptions than prior literature, and the main claim is an iteration complexity of $$\mathcal{O}(\epsilon^{-4})$$. The evidence for this claim is from mathematical proofs.  I don't expect reviewers to look over every single line of a proof, but they did take a look and didn't find any significant issues. I find it likely that the proofs are correct.

The paper almost makes some claims about the suitability of their assumptions (or the unsuitability of assumptions from prior literature), referencing citations about modern neural net architectures. These serve only as motivation so they do not need to be scrutinized to the same level. The reviewers and myself find these plausible.

Overall, we don't see any issues with accuracy in the paper.